# Assessing Non-Ideal Instrumental Effects in High-Resolution FTIR Spectroscopy: Instrument Performance Characterization

Gezahegn Sufa Daba[1,2] and Gizaw Mengistu Tsidu[3]

[1]Department of Physics, Addis Ababa University, P.O. Box 1176, Addis Ababa, Ethiopia
[2]Department of Physics, Debre Birhan University, P.O. Box 405, Debre Birhan, Ethiopia
[3]Department of Earth and Environmental Sciences, Botswana International University of Science and Technology (BIUST), Private Mail Bag 16, Palapye, Botswana

**Correspondence:** Gezahegn Sufa Daba (gezahegnsufaa@gmail.com)

**Abstract.** This study investigates the impact of non-ideal instrumental effects on the performance of high-resolution Fourier Transform Infrared (FTIR) spectrometers, with a focus on the Bruker FTS 120M. Key non-idealities, including retroreflector misalignments, baseline drift, and spectral channeling, were systematically analyzed using advanced diagnostic tools such as ALIGN60 and LINEFIT. The nominal configuration exhibited significant anomalies, notably modulation efficiency (ME) deviations of up to +10.9%, phase error (PE) variability of $2.11 \times 10^{-2}$ radians, and spectral channeling frequencies such as a persistent 2.9044 $cm^{-1}$, along with emerging frequencies around 0.24 $cm^{-1}$ attributed to retroreflector wear and $CaF_2$ beamsplitter degradation. A pronounced anomaly at 40.672 $cm^{-1}$, likely induced by environmental factors such as external vibrations or mechanical instability, was also identified. Implementation of a modified configuration effectively addressed these issues, reducing PE variability to $0.042 \times 10^{-2}$ radians, aligning ME within the NDACC-acceptable threshold of 1.1, and achieving substantial improvements in the instrument line shape (ILS), including sharper peaks, narrower full-width at half maximum (FWHM), and reduced sidelobe asymmetry. Analysis of HBr transmission spectra revealed improved fitting of the P(2) line, characterized by lower residuals and enhanced spectral quality. Simulated Haidinger fringes near zero path difference (ZPD) highlighted alignment degradation patterns, underscoring the necessity for precise optical adjustments. Temporal trends showed an increase in ILS peak height of >14% associated with the instrument upgrade, together with significant mean absolute error (MAE) reductions achieved by the modified configuration. In addition, a targeted retrieval case study demonstrates that explicit propagation of the empirically characterized instrumental response into the forward model reduces spectral residuals and retrieval uncertainties and increases the retrieved total column by approximately 6–7 % relative to the nominal configuration. Overall, this study provides a robust framework for diagnosing and correcting instrumental artifacts, ensuring the accuracy, reproducibility, and long-term stability of FTIR measurements essential for atmospheric trace gas retrievals.

## 1 Introduction

Fourier Transform Infrared (FTIR) spectroscopy has become a cornerstone technique in atmospheric remote sensing, providing high-resolution, non-invasive measurements of trace gases that are crucial for understanding air quality, climate change, atmospheric composition, and atmospheric dynamics (e.g. Bacsik et al. (2004); Yamanouchi et al. (2021)). Its ability to simul-

taneously detect and quantify multiple gases with high precision in a single measurement, combined with its robustness and versatility, makes it essential for ground-based monitoring of atmospheric composition and chemistry (Zhou et al., 2022; Zhu et al., 2024). This capability enables FTIR to capture detailed profiles of pollutants like CO, $C_2H_6$, HCN, and $C_2H_2$, supporting in-depth analysis of pollutant sources, transport pathways, and transformations in the atmosphere. As a result, FTIR instruments, such as the Bruker IFS 120M, are widely adopted by prominent international networks such as the Network for the Detection of Atmospheric Composition Changes Infrared Working Group (NDACC-IRWG) and the Total Column Observing Network (TCOON) (Wunch et al., 2011; Mazière et al., 2017; Yamanouchi et al., 2023; García et al., 2021). These networks rely on the capabilities of FTIR spectroscopy to deliver consistent, long-term, and high-quality data, which are essential for monitoring changes in atmospheric composition, tracking greenhouse gas trends, and assessing the broader impacts on air quality and climate.

FTIR spectrometry operates by splitting an infrared beam into two paths, one reflecting off a fixed mirror and the other off a moving mirror, before recombining to form an interference pattern that is Fourier-transformed into a spectrum to provide molecular absorption information (Salzer, 2008). Accurate retrievals of atmospheric constituents depend critically on the instrument line shape (ILS), which characterizes the spectral resolution and fidelity (Roche et al., 2021). The ILS impacts spectral accuracy by determining resolution and line shape, with deviations in modulation efficiency (ME) or phase error (PE) leading to discrepancies in spectral fitting (Hase et al., 1999; Hase, 2012; Sun et al., 2018b). Imperfections in the ILS lead to errors in determining total column and partial column amounts of trace gases, ultimately causing potential misinterpretations of atmospheric composition (e.g. Sun et al., 2017; Chesnokova et al., 2019; Langerock et al., 2024). Under nominal conditions, ME approaches unity, and PE remains minimal at zero path difference (ZPD), gradually diminishing with increasing optical path difference (OPD) (Garc'ia et al., 2022). Misalignments such as field stop decentering, retroreflector shear, or scanner bar distortions introduce errors in ME and PE, affecting the measured spectra (e.g. Sun et al., 2017, 2018b). For instance, cosine bending misalignment causes overmodulation, particularly at maximum OPD, which distorts the ILS by artificially broadening or narrowing spectral lines and resulting in inaccuracies in gas concentration retrievals (e.g. Hase, 2012; Sun et al., 2017, 2018b). The present study builds directly on the ILS retrieval framework developed by Hase (2012), which introduced the extended LINEFIT-based parameterization of ME and PE in the interferogram domain. While Hase (2012) focused on improving ILS monitoring using optimized gas cell measurements, this study extends the framework to long-term operational data, with a focus on diagnosing and quantifying non-ideal instrumental effects.

Despite the recognized sensitivity of FTIR retrievals to ILS deviations, the NDACC community largely relies on nominal ILS assumptions for trace gas retrievals, often overlooking the impact of misalignments and spectroscopic artifacts on spectral accuracy (e.g. Liu et al., 2021b; García et al., 2021; Smale et al., 2023). Although theoretical studies have investigated the effects of instrumental misalignments on column retrievals of NDACC standard gases through simulations, empirical validation using real-world instrumental setups remains limited (e.g. Sun et al., 2018b). Additionally, spectroscopic artifacts such as baseline drift, spectral channeling, and DC offsets , which are common in field operations, are frequently underreported or uncorrected in routine analyses (e.g. Abrams et al., 1994; Salomaa and Kauppinen, 1998; Blumenstock et al., 2021; Yamanouchi et al., 2023). The absence of standardized protocols for identifying and correcting these artifacts introduces systematic biases into

long-term data records, undermining the reliability and reproducibility of FTIR-based atmospheric measurements. These gaps in addressing practical ILS imperfections and spectroscopic artifacts highlight a critical research need: to move beyond theoretical simulations toward real-world diagnostics and corrections. Without systematic evaluation and mitigation of instrumental non-idealities, the precision and comparability of trace gas retrievals across global networks remain compromised, limiting the broader utility of FTIR observations for atmospheric science. To demonstrate why accurate instrument characterization is essential for atmospheric FTIR applications, this study explicitly integrates a targeted retrieval case study that quantifies how non-ideal instrumental effects propagate into spectral residuals, retrieval uncertainties, and total column estimates. By establishing a direct causal link between empirically characterized instrumental behavior and atmospheric retrieval outcomes within a unified analysis, the work shows that instrument diagnostics and retrieval accuracy cannot be treated independently.

The objective of this study is to rigorously assess the influence of non-ideal instrumental effects on the performance of high-resolution FTIR spectrometers, focusing specifically on the Bruker FTS 120M. Key non-idealities under investigation include retroreflector misalignments, baseline drift, and spectral channeling, which can significantly compromise the accuracy of atmospheric trace gas retrievals. Using advanced diagnostic tools, such as ALIGN60 and LINEFIT, this study systematically diagnoses and corrects mechanical and optical misalignments under operational conditions. To demonstrate the practical relevance of the instrument characterization for atmospheric FTIR applications, a targeted single-day $C_6H_6$ retrieval is conducted using the PROFFIT retrieval algorithm, with the empirically characterized instrumental response explicitly incorporated into the forward model. The retrieval employs NDACC-recommended microwindows and serves solely to quantify retrieval sensitivity to non-ideal instrumental effects through analysis of spectral residuals, retrieval uncertainties, and total column differences. Furthermore, this research addresses existing gaps in the operational protocols of prominent global monitoring networks like the NDACC and the TCCON. By emphasizing the need to integrate real-world instrumental deviations into data analysis frameworks, the study advocates for more robust methodologies that account for practical limitations. Temporal degradation trends, particularly changes in ME and PE, are analyzed to highlight the importance of routine performance evaluations and maintenance schedules for long-term operational stability. By addressing these overlooked challenges, the study ultimately aims to establish standardized methodologies for diagnosing , quantifying, and correcting instrumental artifacts, ensuring that high-resolution FTIR spectrometers, like the Bruker FTS 120M, deliver reliable, reproducible, and precise measurements over time. By addressing these challenges, the research contributes to strengthening the operational frameworks of international networks such as NDACC and TCCON, ultimately advancing global efforts in atmospheric monitoring and climate research.

The paper is structured as follows: Section 2 provides the theoretical and technical background. Section 3 outlines the methodology, followed by the presentation of data analysis results in Section 4. A detailed discussion is presented in Section 5, and the conclusions are summarized in Section 6.

## 2 Theoretical and Technical Background

### 2.1 The Ideal FTIR Spectrometer

The Michelson interferometer is a fundamental component of most FTIR spectrometers enabling precise modulation of the OPD between its fixed and movable mirrors or retroreflectors for high-resolution spectral measurements (e.g. Zeng et al. (2022); Zhou et al. (2024)). An ideal FTIR spectrometer, such as the one shown in Fig. 1, is characterized by precise and optimized components that ensure high accuracy and reproducibility in spectral measurements. Its retroreflector must be perfectly aligned to maintain a linear OPD, while the beamsplitter requires uniform transmission and reflection properties across all wavelengths to minimize artifacts. Additionally, a high-sensitivity detector and finely tuned amplifiers enhance signal quality, collectively ensuring the reliability of spectral data. Together, these elements produce artifact-free interferograms and high-resolution spectra, achieving a modulation efficiency of unity and eliminating phase errors essential for detailed analysis. When a monochromatic plane wave enters the interferometer, it splits into two beams traveling different paths. Their superposition at the detector forms the DC-corrected interferogram, expressed as

$$I(x) \quad = \quad I(x_0)\cos(2\pi\nu x) \tag{1}$$

where $I(x_0)$ is the intensity at zero path difference, and $\nu$ is the wavenumber (e.g. Sahoo et al., 2023). At ZPD, the corner cube mirrors are equidistant from the beamsplitter, causing all incident light to interfere constructively at the detector; as the scan progresses and OPD increases, the interference pattern alternates between constructive and destructive interference based on the source radiation's wavenumber.

### 2.2 Nominal Instrument characteristic

In practice, an FTIR spectrometer's spectral response deviates from the idealized assumption due to imperfections in components like the beamsplitter, detector, amplifier, and cube-corner retroreflectors (CCRs), (e.g. yuan Yue et al., 2017; Zhang et al., 2018). These components introduce wavenumber-dependent non-uniformities that affect modulation efficiency, which quantifies the ability to encode spectral information accurately (e.g. Hanssen et al., 2022). Realistic interferograms incorporate modulation loss and phase delay:

$$I = \int_0^\infty S_0(\nu)m(x,\nu)\cos[2\pi\nu x + \phi(x,\nu)]\,d\nu \tag{2}$$

where $S_0(\nu)$ is the true spectrum, $m(x,\nu)$ is the modulation depth indicating signal quality, and $\phi(x,\nu)$ is the phase error (e.g. Carnio et al., 2023). These parameters capture the cumulative effects of non-idealities, such as reduced modulation depth ($m(x,\nu)$) and phase distortions ($\phi(x,\nu)$), particularly at higher wavenumbers.

#### 2.2.1 Wavenumber-Dependent Instrument Response

Beamsplitters, such as those made from $CaF_2$ (e.g. Fig 2(b)), exhibit dispersion that reduces modulation efficiency (e.g. Letz et al., 2002, 2003). Applying thin germanium coatings ($n = 4.0$) can minimize chirping and enhance performance (e.g. Gill

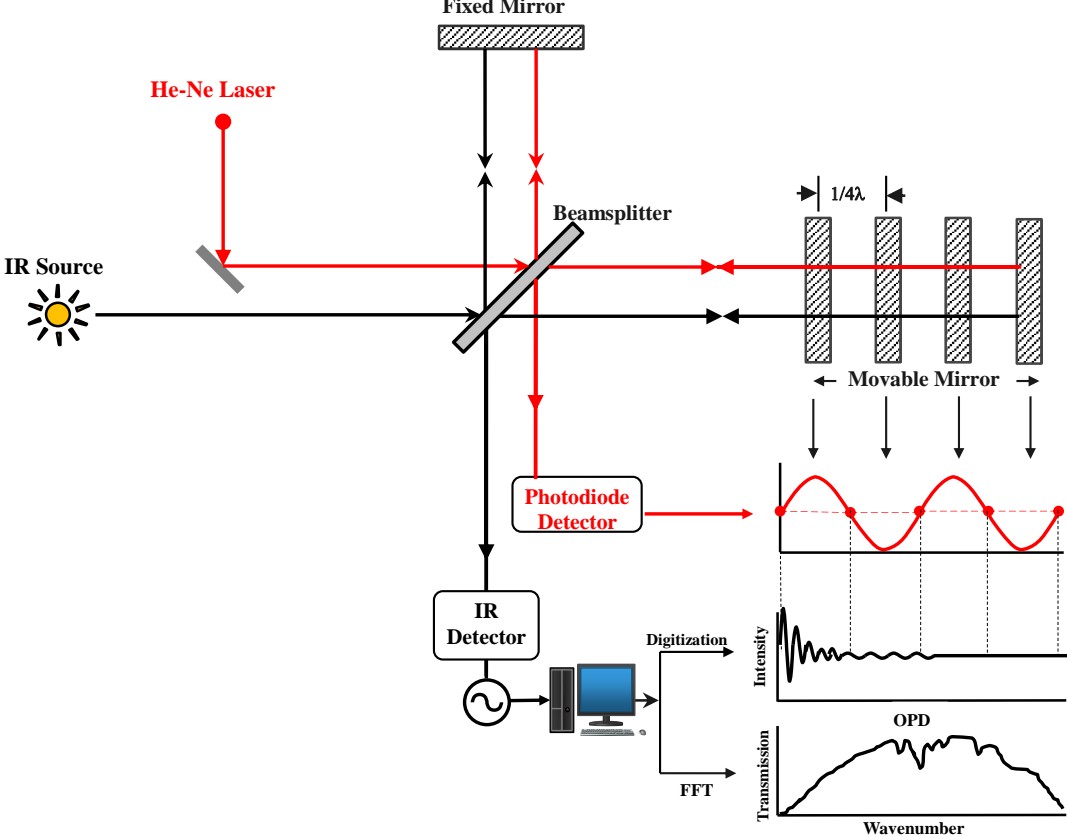

**Figure 1.** Schematic diagram of an ideal FTIR spectrometer illustrating the working principle, including key components such as the light source, beamsplitter, interferometer, sample chamber, and detector.

et al., 2022). The efficiency of a beamsplitter, assuming no absorption or polarization effects, is given by:

$$m_{\mathrm{bs}}(\nu) = 4 \cdot R_{\mathrm{bs}}(\nu) \cdot T_{\mathrm{bs}}(\nu) \tag{3}$$

where $R_{\mathrm{bs}}$ and $T_{\mathrm{bs}}$ represent reflectance and transmittance. While ideal beamsplitters achieve $m_{\mathrm{bs}} = 100\%$, practical imperfections, such as surface irregularities and wavelength-dependent absorption, reduce this value. For example, germanium coatings at a 45° angle of incidence yield an average efficiency exceeding 86% but remain theoretical for perfect uniformity (e.g. Yenisoy and Tüzemen, 2020).

The liquid nitrogen-cooled InSb detector exhibits peak performance near 4000 cm$^{-1}$, with modulation efficiency ($m_{\mathrm{det}}(\nu)$) expressed as:

$$m_{\mathrm{det}}(\nu) = \frac{R_{\mathrm{det}}(\nu)}{R_{\mathrm{det,\ max}}} \tag{4}$$

where $R_{\mathrm{det}}(\nu)$ and $R_{\mathrm{det,\ max}}$ are the responsivity at a specific wavenumber and peak responsivity, respectively. The efficiency decreases from $\sim 1$ near 4000 cm$^{-1}$ to 0.7–0.8 at 1800 cm$^{-1}$, necessitating careful calibration (e.g. Shi et al., 2019). Amplifiers

further contribute wavenumber-dependent gain variations due to the properties of resistors and transistors (e.g. Shi et al., 2019).

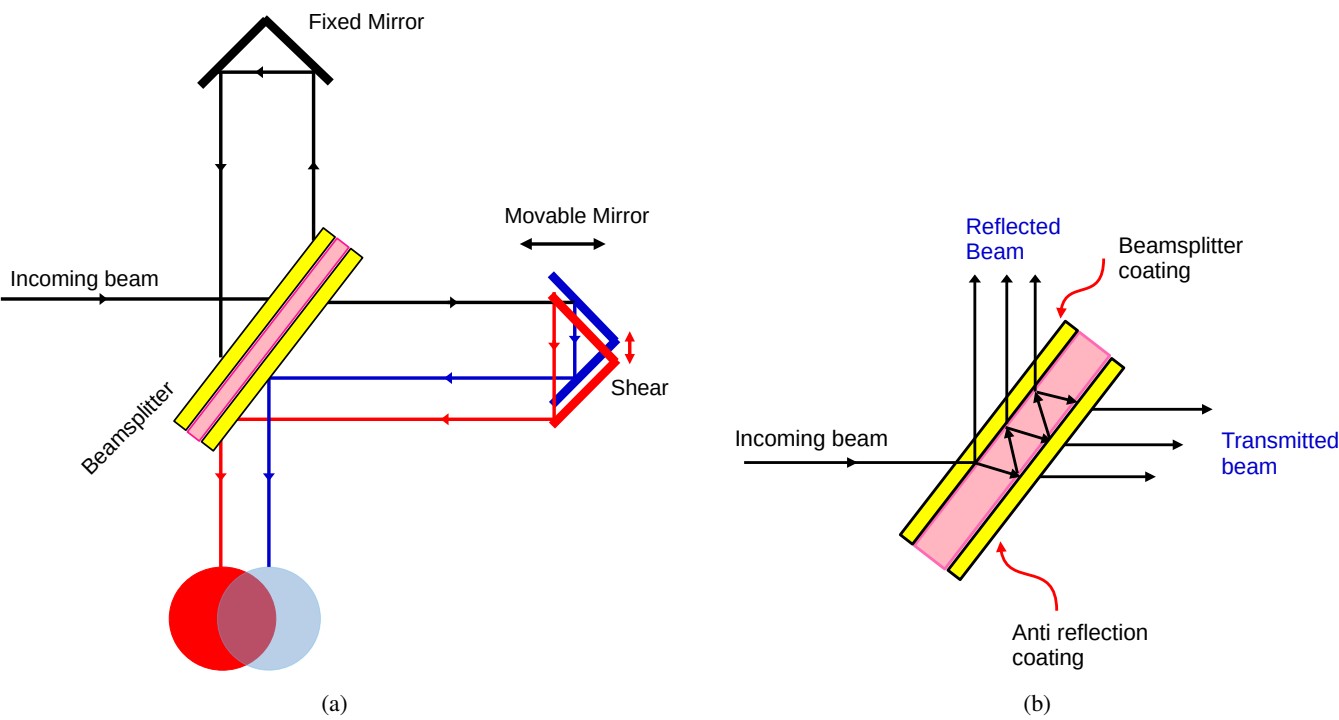

**Figure 2.** (a) Schematic of the interferometer showing fixed and movable retroreflector mirrors, where one movable mirror is laterally sheared, and light from both mirrors is focused on the detector. (b) Beamsplitter illustrating the partial reflection and transmission of incident light, with the coatings labeled for better visualization.

The CCRs used in the Bruker 120M spectrometer, coated with aluminum, silver, or gold, provide reflectivities exceeding 98% with minimal wavenumber dependence in the mid-IR range, ensuring relatively uniform performance (e.g. Liu et al., 2019, 2022). Despite these optimizations, component-specific imperfections underscore the need for precise calibration to 135   ensure high-quality spectral measurements.

### 2.2.2   Self-Apodization

In typical FTIR spectrometers, it's often assumed that the wave phases of a plane wave (a perfectly collimated beam) do not exhibit spatial dependence. However, in real-world scenarios, this assumption is challenged by finite-sized sources, such as extended sources like the sun, which cannot be perfectly collimated due to their finite size (e.g. Wawrzyniuk, 2021). This 140   introduces spatial dependence to the wave phases across the beam, affecting the modulation function (e.g. Fulton et al., 2015). Therefore, the total interferogram accounts for the finite divergence angle ($\alpha$) using a Taylor series expansion of the cosine

function around $\alpha = 0$ and truncating the series to the first two terms (e.g. Ridder et al., 2014; Boone and Bernath, 2019):

$$I(x) = I_0 m_{\text{self}}(\nu, x) cos\left[2\pi\nu x\left(1 - \frac{\Omega}{4\pi}\right)\right] \tag{5}$$

where $\Omega$ is the solid angle of the circular aperture at the focus of a collimating mirror, $\Omega = \pi\alpha^2$ with $\alpha$ representing the angular
radius of the field-of-view, and $m_{\text{self}}(\nu, x)$ is the modulation function due to self apodization

$$m_{\text{self}}(\nu, x) = sinc\left(\frac{\Omega\nu x}{2}\right) \tag{6}$$

Hence, there are two effects on the interferogram/spectrum caused by the finite entrance aperture of a non-ideal interferometer (e.g. Gero et al., 2018): the scale/shift change in OPD and wavenumber which can be expressed as

$$\nu' = \nu\left(1 - \frac{\Omega}{4\pi}\right) \quad \text{and} \quad x' = x\left(1 - \frac{\Omega}{4\pi}\right) \tag{7}$$

A perfectly-aligned interferometer will perfectly center the Haidinger fringes on the field stop at all OPDs (e.g. Sun et al., 2018b). The modulation at ZPD is therefore expected to be unity with assumption that our spectrometer is only subjected to unavoidable instrument characteristics (e.g. García et al., 2022). As the mirror moves away from the ZPD, it experiences modulation loss for the fact that tilt are getting worse for the longer OPDs because of the interferometer's FOV (e.g. García et al., 2022; Liu et al., 2021a). For example, a schematic figure (Fig. 3) illustrates the modulation efficiency of a nominally ideal
instrument in blue, providing a visual comparison against the theoretical and real instrument performance. For an interferometer with a cone of radiation of wavenumber $\nu$ and solid angle $\Omega$ incident upon perfectly aligned mirrors, the interferometer is expected to have tolerable tilt.

## 2.3 Fabry-Pérot Etalons

The Fabry–Pérot effect, or spectral channeling, occurs due to the interference of light waves undergoing multiple reflections
and refractions within parallel optical surfaces (e.g. Ball, 2012; Konevskikh et al., 2015). In FTIR spectrometers, the beam-splitter (e.g. Fig 2(b)) is the primary source of this effect (e.g. Blumenstock et al., 2021). Its layered structure and substrate create Fabry–Pérot cavities that generate standing wave patterns, leading to periodic oscillations in the observed spectrum and reducing the accuracy of spectral analysis (e.g. Blumenstock et al., 2021). When light strikes the beamsplitter, it undergoes partial reflection and refraction at each interface. The transmitted and reflected beams, with decreasing amplitudes due to par-
tial reflection and absorption, are superimposed at the detector. The interference pattern is governed by the optical path length difference ($\Delta L$) between successive transmitted beams, given by (e.g. Hecht, 2016):

$$\Delta L = 2nd\cos\theta, \tag{8}$$

where $n$ is the refractive index, $d$ the thickness of the cavity, and $\theta$ the angle of incidence. The phase shift ($\Delta\phi$) associated with this path difference is:

$$\Delta\phi = 4\pi\nu nd\cos\theta, \tag{9}$$

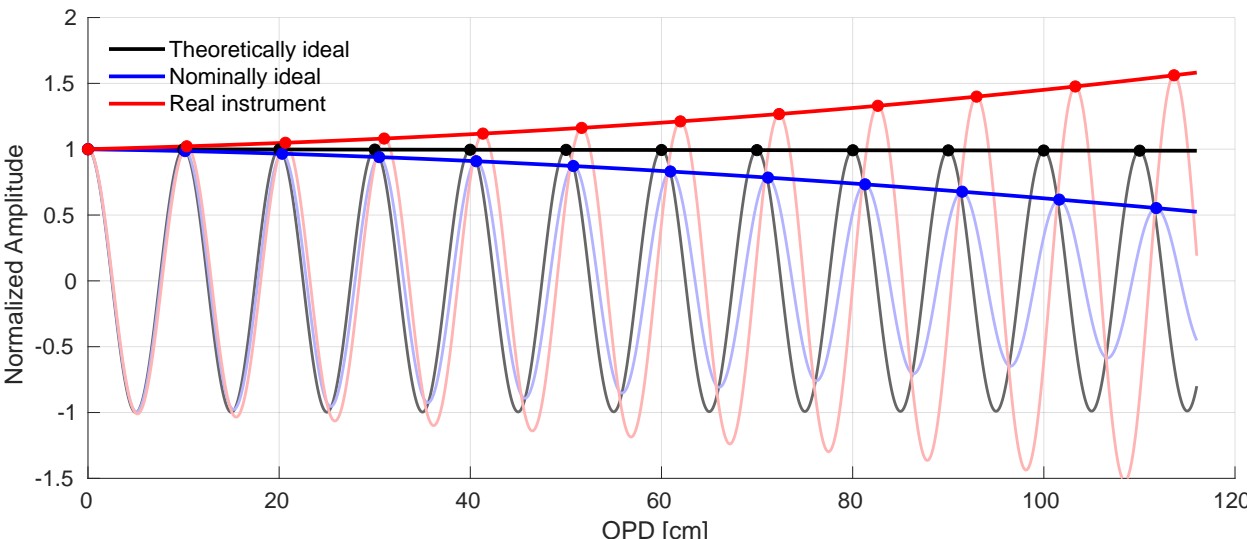

**Figure 3.** Schematic illustration of the deviation between the ideal interferogram and the measured interferogram, with identification of the characteristics described by the modulation function. The theoretically ideal curve represents the analytic interferogram with ME = 1, zero PE, no self-apodization, and no wavenumber dependence. The nominally ideal curve represents a realistic-but-idealized response including only self-apodization from the finite field-of-view, assuming perfect alignment and no other non-idealities.

where $\nu$ is the wavenumber of light. Constructive interference occurs when: $\Delta\phi = m \cdot 2\pi$  (for integer $m$), and destructive interference occurs when: $\Delta\phi = \left(m + \frac{1}{2}\right) \cdot 2\pi$. These phase conditions define the periodicity of the spectral oscillations. The free spectral range (FSR), representing the spacing between constructive interference maxima, is:

$$\nu_{\text{FSR}} = \frac{1}{\Delta L} = \frac{1}{2nd\cos\theta}. \tag{10}$$

The intensity of the observed spectrum is affected by these periodic oscillations, which can be modeled as:

$$I(\nu) = I_0(\nu) + A\cos(2\pi f_c \nu + \phi), \tag{11}$$

where $I(\nu)$ is the observed intensity, $I_0(\nu)$ is the true spectrum, $A$ is the amplitude of the channeling oscillations, $f_c$ is the channeling frequency related to the FSR ($f_c = \nu_{\text{FSR}}$), and $\phi$ is the phase offset. The harmonic oscillations distort the baseline of the spectrum and can overlap with absorption features, making it challenging to accurately retrieve spectral parameters such

as line strength, width, and position. These distortions are particularly problematic for weak absorption lines, where the artifact may mimic or obscure real spectral features (e.g. Hecht, 2016; Blumenstock et al., 2021).

### 2.4 Retroreflector Misalignment and Overmodulation

Bruker interferometers, such as the 120M model, rely on cube-corner retroreflectors to ensure precise and reliable measurements. These retroreflectors consist of three mutually perpendicular reflective surfaces forming the corner of a cube, which

enables them to reflect incident beams back parallel to their original paths, regardless of the direction of incidence (e.g. Brown et al., 2014). This design minimizes the impact of mirror tilt on the interferometer's performance, maintaining accurate retroreflection even when minor misalignments occur. However, while the cube-corner retroreflector is immune to tilting, it remains sensitive to lateral displacement, where any positional shift can disrupt the beam alignment.

Lateral displacement, or shear, becomes a critical factor influencing overmodulation in the interferometer (e.g. Frey et al., 190 2019). This misalignment introduces an additional OPD that varies with the propagation angle, significantly affecting the modulation depth at the interferometer's center burst (e.g. Martino and Hagopian, 1998; Henault et al., 1999). In the Bruker 120M spectrometer, shear arises from the misalignment of the retroreflector, leading to a lateral shift of the reflected beam in one arm of the interferometer (e.g. Sun et al., 2018b). This shift disrupts phase coherence and reduces modulation depth in the interferogram, particularly at ZPD, due to wavefront aberrations, corner cube retroreflector misalignment, and lateral 195 displacement of optical components (e.g. Yue et al., 2017; Sun et al., 2018b).

To maintain consistency in spectral analysis, the modulation at ZPD is normalized to unity, ensuring proper area-normalization of the ILS (Hase, 2012). While this normalization provides a standard reference, it can introduce an apparent increase in modulation amplitude as a function of OPD in the presence of shear misalignment (e.g. Hase, 2012; Sun et al., 2018b), as evidenced in Fig. 3. This apparent increase, however, does not imply a true enhancement of modulation efficiency; rather, it reflects 200 distortions caused by the lateral displacement.

The lateral displacement ($\Delta x$) introduced by the misaligned retroreflector can be expressed as:

$$\Delta x = \varepsilon(x)\sin\alpha \tag{12}$$

where $\varepsilon(x)$ is the lateral shift of the retroreflector apex relative to the optical axis, and $\alpha$ represents the acceptable angle of tilt. This displacement introduces phase incoherence between recombining wavefronts, which, for a range of incident angles, 205 reduces the modulation depth. Studies (e.g. Murty (1960); Kauppinen et al. (2004); Genest and Tremblay (2005)) have extensively analyzed the effects of such lateral shifts on modulation depth. For a circular aperture, these shifts produce a diffraction effect represented by the jinc function:

$$m\left[\nu_0, \varepsilon(x)\right] = \frac{2J_1\left[2\pi\nu_0\varepsilon(x)\alpha\right]}{2\pi\nu_0\varepsilon(x)\alpha} \approx 1 - \frac{\left[2\pi\nu_0\varepsilon(x)\alpha\right]^2}{8} \tag{13}$$

Here, $J_1$ is the first-order Bessel function, and the term $2J_1(x)/x$ represents the diffraction pattern of the circular aperture. As 210 OPD increases, modulation depth decreases due to the worsening effects of shear, emphasizing the need for precise alignment of the retroreflector. Understanding and addressing these misalignments are essential for ensuring accurate spectral measurements and reducing overmodulation artifacts in FTIR spectroscopy.

## 2.5 Combined Effects of Instrumental Artifacts on the Interferogram

The combined effects of nominal instrument response, misalignment, spectroscopic artifacts, and self-apodization can sig-215 nificantly distort the interferogram in FTIR spectroscopy (e.g. Gero et al., 2018). While the nominal response represents the ideal performance of components, real-world conditions often introduce misalignments that degrade modulation efficiency

and induce phase errors. Spectroscopic artifacts further disrupt signal coherence, while self-apodization imposes wavenumber-and OPD-dependent modulation loss due to the finite FOV. These interconnected effects necessitate a comprehensive model that accounts for their cumulative impact, enabling precise corrections and ensuring reliable spectral retrievals. Therefore, by accounting for the instrumental non-idealities, the general expression of the interferogram would take the form:

$$I(\nu_0, x) \quad = \quad \frac{1}{2} \times m_{\text{tot}}(\nu_0) \times I(\nu_0) \cos\left[\Delta\phi + \phi_{\text{tot}}(\nu_0)\right] \tag{14}$$

where $m_{\text{tot}}(\nu_0)$ is the total modulation efficiency combining nominal response, misalignment, spectroscopic artifacts, and self-apodization.

## 3 Methodology

### 3.1 Introduction to the Experimental Setup

#### 3.1.1 Historical Perspectives

The Bruker FTS 120M spectrometer, installed in Addis Ababa, is dedicated to measuring atmospheric gases in the mid-infrared (MIR) region and is the only known high-resolution instrument within the NDACC-IRWG network located in the tropical African region. It traces its origins to its predecessor, the Bruker FTS 125HR, originally based in Kiruna, Sweden, where it specialized in near-infrared (NIR) spectroscopy. Following a significant upgrade to enhance its capabilities and shift its focus to MIR spectroscopy, the instrument was renamed the FTS 120M and relocated to Addis Ababa to take advantage of the high-altitude location for improved atmospheric research. In 2009, the Karlsruhe Institute of Technology (KIT) collaborated with Addis Ababa University (AAU) to install the spectrometer at the College of Natural Sciences (9.01° N, 38.76° E, and 2443 m above sea level). Measurements using the instrument began in May 2009, and since then, research studies have been conducted by KIT-affiliated researchers as well as researchers and graduate students from the university (e.g. Takele Kenea et al., 2013; Tsidu et al., 2014; Hase et al., 2015; Yirdaw Berhe et al., 2020; Berhe et al., 2020).

#### 3.1.2 Overview of the Components

The Bruker FTS 120M spectrometer in Addis Ababa is equipped with advanced components designed to optimize its performance for high-resolution atmospheric measurements. Central to its operation is a helium-neon (HeNe) laser, which ensures precise alignment and stability of the scanning mirror, critical for producing accurate spectral data. The spectrometer utilizes a $CaF_2$ beamsplitter, covering the spectral range of 1200-15000 cm$^{-1}$, paired with a liquid-nitrogen-cooled indium antimonide (InSb) detector to capture interferograms in the MIR region. Although the system also supports a KBr beamsplitter for the 450-4800 cm$^{-1}$ range and a Mercury Cadmium Telluride (MCT) detector, these components have not been utilized at this site. The system operates under a vacuum maintained at pressures below 1 hPa by a dedicated pump, minimizing atmospheric interference and enhancing stability. Significant maintenance in 2012 by the KIT addressed an InSb detector failure and extended the optical path length to improve spectral resolution. Manufactured by Bruker Optics in Germany, the FTS 120M

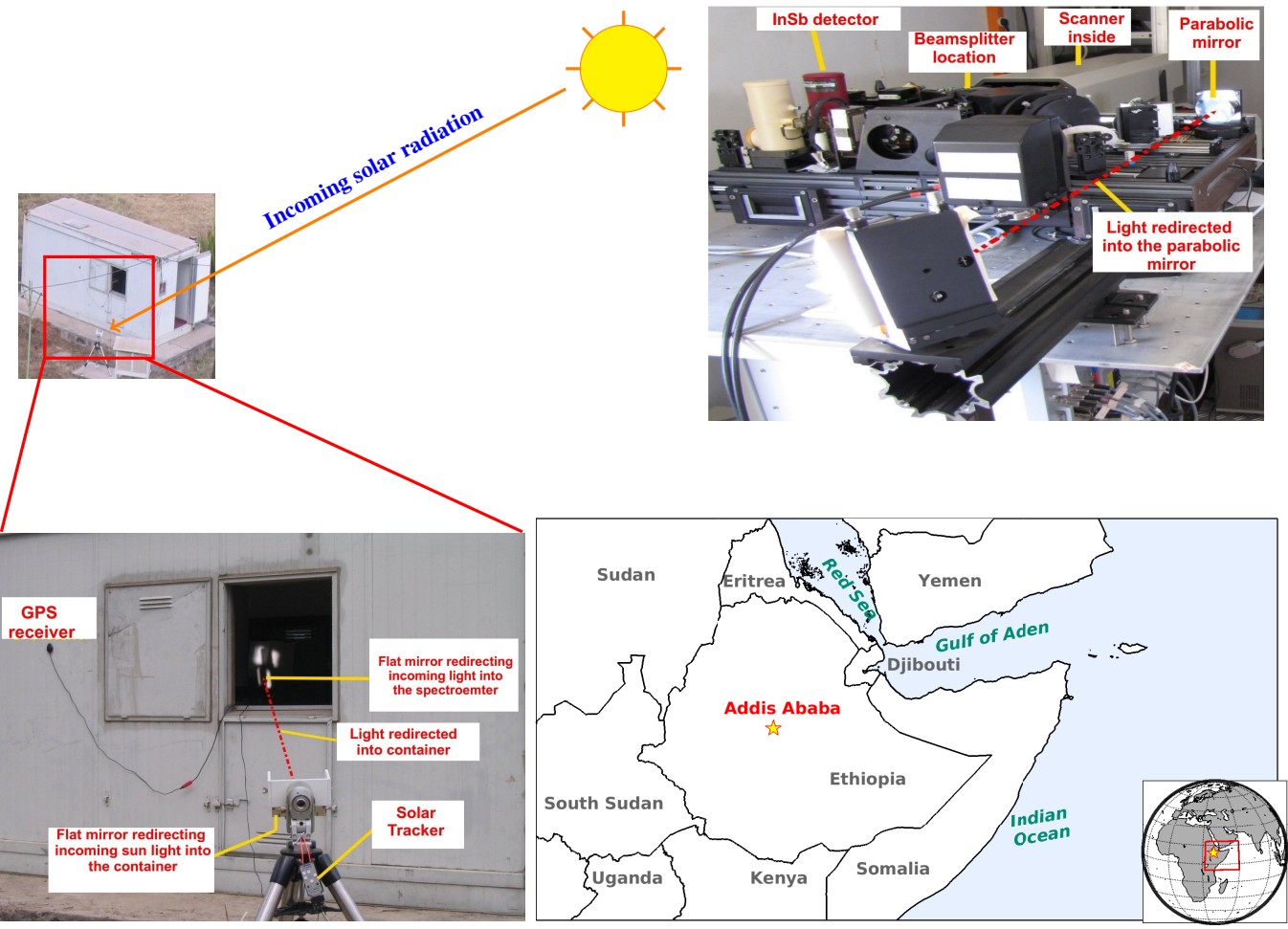

**Figure 4.** Deployment and configuration of the Bruker 120M spectrometer in Addis Ababa. The instrument is installed at Addis Ababa University, College of Natural Sciences (red star on map). The lower-left panel shows the container setup with a GPS antenna for time synchronization and a solar tracker redirecting sunlight into the spectrometer. The top-right panel displays key components of the instrument, including the InSb detector, beamsplitter, scanner, and parabolic mirror. The sun and arrows illustrate the path of incoming solar radiation.

offers exceptional sensitivity and resolution, making it an integral tool for atmospheric research and a key instrument in global networks such as NDACC and TCCON.

| Filter number | Wavenumber range ($cm^{-1}$) |
|:---:|:---:|
| 1 | 3950 - 4900 |
| 2 | 2900 - 3950 |
| 3 | 2400 - 3320 |
| 4 | 1900 - 2750 |
| 5 | 1730 - 2250 |

**Table 1.** NDACC narrow bandpass filters used in Bruker 120M at Addis Ababa

## 3.2 Measurements Conditions

The operation of the Bruker FTS 120M spectrometer is carefully managed under controlled environmental conditions to ensure high performance and precise data acquisition. The spectrometer room is maintained within a temperature range of 18°C to 35°C, with temperature fluctuations strictly limited to no more than 1°C per hour and 2°C per day (Bruker Optik GmbH, 2009). These limits are critical to prevent thermal gradients within the instrument, which could lead to mechanical distortions in the optical components, adversely affecting alignment and spectral resolution. Relative humidity is kept below 80% to 255 protect hygroscopic elements, such as the KBr beamsplitter, optical filters, and sample cell windows, from moisture-induced degradation. To avoid condensation effects, the room temperature is maintained slightly above the ambient level. Additionally, the InSb detector operates in a vacuum environment with pressures below 1 hPa, which minimizes thermal noise and enhances sensitivity. The vacuum pump is required periodically to maintain these low-pressure conditions, particularly after maintenance activities or when the system indicates pressure levels exceeding the operational threshold, ensuring the detector continues to 260 function with optimal performance. Together, these environmental and operational controls are vital for maintaining instrument stability and enabling high-resolution measurements essential for precise atmospheric trace gas retrievals.

## 3.3 ILS Characterization

### 3.3.1 Background Signal Characterization

Background signal measurements were conducted with a high-precision configuration optimized for MIR spectral analysis.
The system employed a $CaF_2$ beamsplitter and a liquid nitrogen-cooled indium antimonide (LN-InSb) detector with a 30° FOV. The aperture was set to 0.85 mm, and the source was configured for MIR emissions. A focal length of 220 mm was used to focus the optical path. Signal processing included no high-pass filter, a low-pass filter with a cutoff at 20 Hz, and optical filters configured as Filter 3 for both primary settings, with an open configuration for additional filtering. A preamplifier gain of 1 was applied, and acquisition was performed in single-sided, forward-backward mode, with 100 scans (50 forward and

50 backward) completed over $\sim$ 68 minutes. The interferogram was obtained at a resolution of 0.00775 cm$^{-1}$, phase corrected using the Mertz algorithm (e.g. Chen et al., 2024), and stored in the OPUS software as full bidirectional data (e.g. Lingling et al., 2017), preserving both forward and backward scans (e.g. Fig. 5(a)). The forward and backward interferograms were transformed and phase corrected separately, with the resulting spectra combined using a mean function (e.g. Fig. 5(b)). The zero crossings of the HeNe laser, used to sample the interferogram, were calculated based on the Nyquist theorem (e.g. Zabit

and Bernal, 2023), which ensures adequate sampling by setting the maximum detectable frequency ($\nu_{max}$) to half the sampling rate. Using the relationship:

$$\Delta x = \frac{1}{2(\nu_{max} - \nu_{min})} \tag{15}$$

where $\Delta x$ is the sampling interval of the laser, and $\nu_{min}$ is the minimum frequency (set to 0 in this case). With $\nu_{max}$ derived from the spectral range, the sampled positions ($x_{HeNe\_sampl}$) were calculated as:

$$x_{HeNe\_sampl} = [0, \Delta x, 2\Delta x, \ldots, (N-1)\Delta x] \tag{16}$$

where $N$ is the total number of interferogram data points. This approach ensured accurate mapping of the interferogram to spectral data. The apodization function used was Boxcar, and the system was calibrated with a He-Ne laser at $\sim$15,800 cm$^{-1}$.

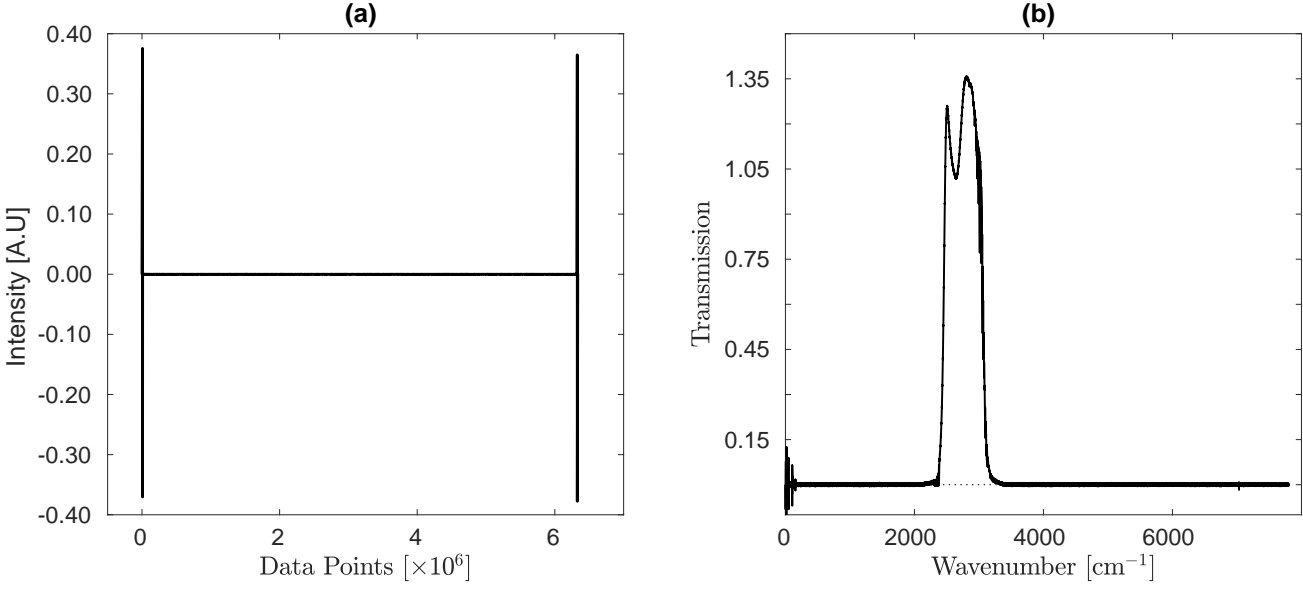

**Figure 5.** (a) Single-sided, bi-directional interferogram from the measurement dated 091209 and (b) its corresponding Fourier transform. The interferogram was obtained at a resolution of 0.00775cm$^{-1}$ using 100 scans. Phase correction and spectrum calculation were performed using the Mertz method. Forward and backward scans were individually transformed, phase-corrected, and combined using a mean function.

### 3.3.2 Low-Pressure Gas Cell Measurements

The NDACC-IRWG enforces rigorous standards for ILS characterization and correction to ensure consistent, accurate, and reliable spectroscopic measurements across its global network of FTIR instruments. The ILS, which defines the instrument's response to a monochromatic light source, is regularly evaluated through established methods such as low-pressure HBr gas cell measurements and He-Ne laser-based interferometry to assess key parameters, including modulation efficiency, phase errors, and line broadening (e.g. Hase et al., 1999; Hase, 2012). The LINEFIT implementation and ILS parameterization used in this study follow the formalism introduced by Hase (2012), including the normalization of ME at ZPD and the retrieval of modulation amplitude and phase at discrete OPD positions. The present study does not modify the underlying ILS retrieval scheme, but applies it systematically to a long-term dataset in order to evaluate temporal variability, maintenance-induced changes, and the interaction of ILS distortions with other spectroscopic artifacts. These evaluations ensure compliance with predefined thresholds to minimize spectral distortions and retrieval inaccuracies. Identified deviations are addressed through corrections integrated into retrieval algorithms and routine hardware maintenance, including optical realignments and field stop optimizations. Environmental stabilization measures, such as thermal control and vibration isolation, further improve fluctuations (e.g. García et al., 2022).

The calibration of the Bruker IFS 120M spectrometer relies on low-pressure HBr gas cells, which provide narrow Doppler-broadened absorption lines that are highly sensitive to optical misalignments and imperfections. These features, combined with the closely spaced absorption lines ($\sim$ 20 cm$^{-1}$ apart), (Gupta et al.) and the presence of the isotopologues H$^{79}$Br and H$^{81}$Br, double the number of calibration lines, enabling detailed spectral analysis. Measurements employed similar instrumental settings as those used for background signal acquisition, ensuring consistency in aperture and optical configurations, and focused on the spectral range of 2408.0-2530.0 cm$^{-1}$. A 2 cm sealed glass cell filled with HBr at 1.5 mbar, in accordance with NDACC-IRWG standards, was used to identify instrumental artifacts, including baseline offsets, phase errors, and spectral channeling, as well as to evaluate modulation efficiency and validate the spectrometer's response. The transmittance spectrum was calculated using OPUS software by dividing the sample spectrum by the background, effectively minimizing artifacts such as water vapor absorption. The resulting spectrum revealed distinct P-branch (P(5) to P(8)) and R-branch (R(4) to R(7)) lines dominated by Doppler broadening, with excellent agreement between measured and simulated data, as shown in Fig. 6(a). Residuals between the measured and calculated spectra provided valuable insights into misalignment and spectroscopic artifacts, highlighting the importance of regular HBr cell measurements and residual analysis using LINEFIT 14.5 software to ensure instrument accuracy and reliable spectroscopic data, as noted by (Hase et al., 1999; Hase, 2012).

### 3.3.3 Wavenumber Calibration and Data Harmonization

Accurate alignment between the measured and calculated wavenumber scales is essential to ensure that interferogram sampling correctly maps into the spectral domain for ILS retrievals. In high-resolution Bruker FTIR spectrometers, the sampling grid is defined by the internal He–Ne reference laser; therefore, any change in the laser frequency or scanning geometry alters the wavenumber scale and must be explicitly corrected. In 2011, the Bruker IFS 120M underwent major maintenance, including

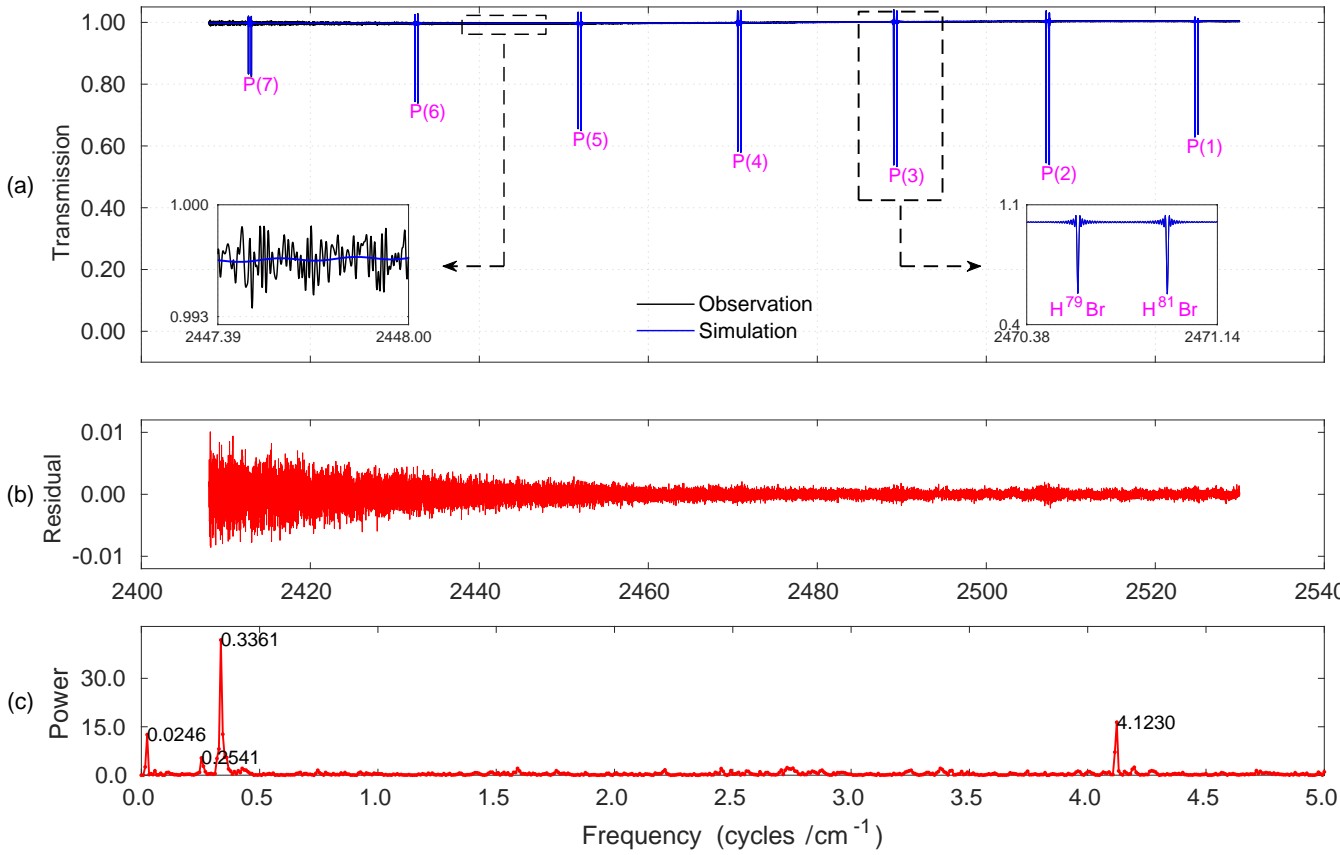

**Figure 6.** HBr transmission spectra acquired via gas cell measurement. (a) Comparison between observed and simulated spectra in the region 2400-2540 cm$^{-1}$, illustrating P-branch transitions. Insets display isotopic splitting (right) and low-amplitude channeling patterns (left). (b) Residuals between experiment and simulation. (c) FFT of the residual showing dominant frequencies linked to channeling artifacts arising from instrumental imperfections. Data acquired on December 14$^{\text{th}}$, 2012.

replacement of the He–Ne reference laser and an increase in the MOPD from 100 cm to 116.13 cm. These changes improved the effective resolution from ∼0.009 cm$^{-1}$ to ∼0.00775 cm$^{-1}$ and shifted the laser reference wavenumber from 15798.088 cm$^{-1}$ to 15798.0604 cm$^{-1}$. Together, these modifications introduced small but systematic distortions in the spectral sampling grid between pre- and post-maintenance datasets. For the purpose of comparing the HBr spectra—both over the full spectral range and specifically around the P(2) line—we reconstructed the measured spectra from their interferograms using identical FFT-processing parameters. We applied a zero-filling factor of 2, a phase-resolution parameter of 4, boxcar apodization, Mertz phase correction, automatic ZPD detection, a spectral window of 0–7800 cm$^{-1}$, and clipped the effective MOPD to 100 cm to match the pre-2011 resolution. In addition, all spectra were internally resampled by a factor of four inside LINEFIT to increase the sampling density around narrow spectral features. For all subsequent analyses, we followed a similar procedure while keeping the original spectral resolution: post-maintenance spectra were interpolated onto the pre-maintenance grid to ensure

consistent sampling density while retaining the improved resolution provided by the upgraded instrument. After harmonizing the sampling points, transmittance spectra were calculated from the uniformly sampled datasets. The multiplicative distortion introduced by the laser replacement was corrected using the stretching factor

$$s = \frac{15798.088}{15798.0604} \approx 1.00000175, \tag{17}$$

and all wavenumber-scale corrections were applied using OPUS. This ensures that the complete dataset resides on a physically consistent and harmonized wavenumber scale, enabling reliable and unbiased comparison of ILS characteristics across the full multi-year record. Consequently, the measured spectra entering LINEFIT already lie on a common, physically correct wavenumber axis, leaving LINEFIT to apply only the small, appropriate fine-alignment shift relative to HITRAN. In addition, LINEFIT was configured to read the HBr isotopologues ($H^{79}Br$ and $H^{81}Br$) separately from HITRAN rather than treating the

line as a single composite transition. This isotopic treatment provides a more accurate representation of the HBr absorption structure and prevents spurious s-shaped residuals that could otherwise arise from an inappropriate line model.

### 3.4 True gas cell column amount

Ideally, it was assumed that the temperature of the room was identical to that of the HBr cell, and that the pressure within the HBr cell remained constant throughout the experiment. Thus, the amount of the gas sample contained inside the cell can be

derived from the consideration of the ideal gas law and initial cell parameters (e.g. Tenny and Cooper, 2017):

$$\text{cell column} = 7.243 \times 10^{24} \left( \frac{P\ell}{T} \right) = 7.39 \times 10^{20} \frac{\text{molecules}}{m^2} \tag{18}$$

Where P and T are the pressure in millibar and the temperature in Kelvin of the HBr gas confined inside the cell, respectively and $\ell$ is the thickness of the HBr cell.

However, uncertainty in the calculation of the actual gas concentration in the cell arises from the temperature of the gas

cell and imperfections encountered in the intsrument optics. Though the temperature of the room is a constantly monitored quantity, it may not be representative of the actual mean temperature of the absorbing HBr (e.g. Alberti et al., 2022). The temperature of the room and the temperature within the cell may differ slightly due to several factors. For example, the cell's environment, influenced by the presence of the HBr gas and its interaction with the cell material, could lead to slight variations in temperature. The effective temperature of the gas is therefore determined by least-squares fit of the retrieved line strengths.

Moreover, HBr cells are known to experience slow leakage over time, which results in a gradual decrease in both concentration and pressure (Lutsch, 2019). Consequently, it becomes essential to accurately determine the HBr column amount and pressure to ensure the reliability of experimental data.

LINEFIT has been effectively utilized to retrieve the total pressure, partial pressure, and cell column amount under specified conditions, as summarized in Table 2. Initially, the temperature, pressure, and total pressure from nominal instrument setup

and ideal-gas estimates were used as starting parameters for the LINEFIT run. The process began by fitting the temperatures of the two isotopes of HBr gases within the cell, continuously updating the cell column as the temperature fluctuated until the temperature differences between the isotopes stabilized. Subsequently, with the fitted temperature constant, both the partial and

total pressures were varied, and the cell column was updated to reflect these new pressures until the column ratio between the isotopes approached 1:1. For enhanced accuracy, further refinement was conducted by iterating with the total pressure, ensuring it remained greater than the partial pressure. The cell pressure was repeatedly updated based on the retrieved concentration until it converged with a defined threshold of 0.001 mbar, ensuring precise and reliable results. Throughout this procedure, all instrumental parameters, including the wavenumber scale and ILS, were held fixed, such that the fitted thermodynamic parameters could not compensate for potential spectral misalignment. This sequential fitting of HBr cell temperature, pressure, and column amount with fixed instrumental parameters follows typical NDACC practice in ILS characterization using low-pressure HBr gas cells (e.g. Yamanouchi et al., 2023).

| Dates [YYMMDD] | Scanner Temperature [Kelvin] | Partial Pressure [millibar] | Total Pressure [millibar] | Temperature [Kelvin] | Column [$\times 10^{20} \frac{molecules}{m^2}$] |
|---|---|---|---|---|---|
| 091208 | 297.65 | 1.50 | 1.51 | 298.6 | 7.33 |
| 100128 | 299.25 | 1.40 | 1.50 | 301.7 | 7.20 |
| 110117 | 294.95 | 1.50 | 1.60 | 298.7 | 7.76 |
| 121214 | 296.35 | 1.50 | 1.62 | 296.8 | 7.91 |
| 130218 | 298.05 | 1.50 | 1.58 | 300.3 | 7.62 |
| 150317 | 292.75 | 1.40 | 1.50 | 295.6 | 7.35 |
| 161021 | 291.35 | 1.40 | 1.49 | 293.1 | 7.36 |

**Table 2.** LINEFIT retrieval of cell temperature, cell column amount, partial pressure and total pressure of the HBr istopes.

## 3.5 LINEFIT14.5

Retrieval of the ILS is a critical step in evaluating the performance of FTIR spectrometers. In this study, the LINEFIT software, developed by Hase et al. (1999)., was employed to retrieve the ILS parameters. LINEFIT is a flexible tool that supports both simplified parameterization for near-ideal instruments and detailed descriptions for significantly non-ideal conditions (e.g. Alberti et al., 2022). It retrieves the ILS by characterizing two primary distortions: ME, which quantifies the attenuation of modulated radiation, and PE, which represents asymmetry in the line shape caused by phase shifts in the optical components. These distortions are influenced by instrumental factors, including baseline drift, spectral channeling, and spectral shifts, which arise from optical misalignments and calibration errors. LINEFIT provides two parameterization schemes: a simple parameter set, using a linear decline in ME and a constant PE, and an extended parameter set, which employs 40 parameters to capture variations in ME and PE across the interferogram with smoothness constraints ensuring physical realism. Input data included absorption lines selected from microwindows with spectroscopic properties derived from the HITRAN (high-resolution transmission molecular absorption) database (Gordon et al., 2022), which provides reference line parameters for atmospheric gases. Gas cell parameters, such as pressure, temperature, and column density, were carefully controlled to achieve Doppler-limited

linewidths and precise spectral profiles. Through iterative adjustments, LINEFIT minimizes residuals between the measured
and simulated spectra, offering a comprehensive characterization of both ideal and non-ideal instrument behavior. In addition
to retrieving the ILS parameters, LINEFIT also provides averaging kernels that quantify the sensitivity of the retrieval to per-
turbations in the fitted ILS functions. These averaging kernels were used in this study to assess how the apodization and phase
parameters influence the response of the instrument across OPD. After the retrieval had converged, the relevant sub-matrices
corresponding to the ILS parameters were isolated and visualized as two-dimensional kernels in OPD space. This approach
ensures that the averaging kernels directly reflect the parameter sensitivities as derived from the LINEFIT inversion, rather than
from independent forward simulations.

### 3.6 ALIGN60

ALIGN60 is a model included in the LINEFIT package that calculates the instrument ILS and Haidinger fringes based on
misalignment parameters, such as retroreflector displacement or aperture shifts, to evaluate and correct interferometer perfor-
mance. The model has been used in previous studies (e.g. Sun et al., 2018a, b; Yin et al., 2021) and its simulation results have
been found to be reliable for evaluating and correcting interferometer performance. According to Sun et al. (2018b), the optical
misalignments on high-resolution FTIR spectrometers can be modelled as:

$$\varepsilon(x) = C_0 + C_1 \cdot \mathrm{x} + C_2 \cdot \cos\left(\frac{2\pi \cdot \mathrm{x}}{\mathrm{T}}\right) + C_3 \cdot \sin\left(\frac{2\pi \cdot \mathrm{x}}{\mathrm{T}}\right) \tag{19}$$

where $C_0$ represents the constant shear offset, $C_1$ describes the linear shear dependence, and $C_2$ and $C_3$ capture periodic
components arising from mechanical bending or oscillations in the retroreflector system. By incorporating both constant and
periodic shear contributions, ALIGN60 provides a comprehensive simulation of their effects on interferometric measurements.
Additionally, it models decentering of the field stop, boundary sharpness, and deformation of the field stop image, further
influencing fringe patterns. By generating sketches of laser fringes at specific OPD positions, ALIGN60 allows for the visual-
ization of phase distortions and their influence on Haidinger fringes. These simulations are critical for diagnosing mechanical
misalignments, optimizing instrument performance, and improving the accuracy of atmospheric trace gas retrievals.

### 3.7 Simulating Nominal Haidinger Fringe

The determination of the Haidinger fringe period and its simulation involved a systematic approach to assess its impact on
instrument performance. The period of the shear was calculated by analyzing the phase error data from LINEFIT under the
nominal configuration. First, the phase error was subjected to a FFT to identify the dominant periodic components. The am-
plitude spectrum resulting from the FFT was sorted in descending order, and the frequency corresponding to the maximum
amplitude was identified as the dominant shear frequency. The period of the shear was then determined as the reciprocal of this
frequency, enabling precise characterization of the periodic misalignment.

The simulation of the ILS using ALIGN60 was performed to replicate the effects of the Haidinger fringe and validate the
findings against LINEFIT results. Visual inspection of the modulation efficiency pattern from the Bruker 120M spectrometer
at our site revealed similarities to the cosine bending misalignment category described in Sun et al. (2018b). Based on this

observation, a periodically varying cosine bending misalignment was assumed, with the initial amplitude set according to the values reported in prior studies (e.g. Sun et al., 2018b). ALIGN60 was configured to simulate the ILS at grid points matching those used in LINEFIT to ensure consistency.

To quantify the agreement between the ILS simulated by ALIGN60 and LINEFIT, the following objective function was used:

$$\text{Objective Function} = \sqrt{\text{mean}\left[(\text{ILS}_{\text{ALIGN60}} - \text{ILS}_{\text{LINEFIT}})^2\right]} \tag{20}$$

The amplitude of the misalignment was iteratively tuned to minimize this objective function. Convergence was achieved when the RMSE between ALIGN60 and LINEFIT fell below 0.1. Once the optimized misalignment parameters were established, the final ILS simulation was performed using ALIGN60, providing a robust characterization of the Haidinger fringe effects on the spectrometer's performance.

## 3.8 ILS Retrieval Algorithm Configuration

The simulation of HBr spectra was performed using two configurations: a nominal configuration and a modified configuration. In this study, the nominal configuration represents an idealized instrument response that includes only self-apodization effects arising from the finite field of view, assuming unity ME and zero PE. No LINEFIT-derived ILS parameters, wavenumber shifts, or spectral channeling corrections are applied in the nominal configuration. The modified configuration, in contrast, explicitly incorporates the ILS retrieved using LINEFIT, including deviations in ME and PE from the nominal response, together with additional corrections for baseline distortions and spectral channeling, thereby representing realistic instrument behavior. The modified setup introduced deviations, starting with the nominal baseline as an initial guess while initially excluding channeling. For baseline correction, a sixth-degree polynomial was chosen to model complex variations without overfitting, expressed as:

$$y_i = a_0 + \sum_{j=1}^{k} \left(a_j \nu_i^j + e_j\right) \tag{21}$$

where $y_i$ is the baseline-corrected transmittance spectrum, $\nu_i$ is the wavenumber, $a_0$ is the offset, $a_1$ the slope, $a_2$ the curvature, and higher-degree terms ($a_3$ to $a_6$) account for additional baseline distortions. The coefficients were determined using least-squares regression, and the corrected baseline was iteratively subtracted to obtain distortion-free spectra.

In the analysis of transmittance spectra, the observed intensity, which is often contaminated by spectral channeling, is effectively handled by LINEFIT through the decomposition of the channeling component into its sine and cosine projections. The channeling term, represented as $A\cos(2\pi f_c \nu + \phi)$, is decomposed into $A\cos\phi\cos(2\pi f_c \nu) + A\sin\phi\sin(2\pi f_c \nu)$. This allows the observed intensity to be expressed in summation form as:

$$I(\nu) = I_0(\nu) + \sum_{i=1}^{N} \left[A_{\cos,i}\cos(2\pi f_i \nu) + A_{\sin,i}\sin(2\pi f_i \nu)\right], \tag{22}$$

where $N$ is the total number of channeling frequencies considered, and $A_{\cos,i}$ and $A_{\sin,i}$ are the amplitude projections along the cosine and sine components for the $i$-th channeling frequency $f_i$. These projections are related to the amplitude $A_i$ and phase

$\phi_i$ of the $i$-th channeling component by the equations:

$$A_{\cos,i} = A_i \cdot \cos(\phi_i), \quad A_{\sin,i} = A_i \cdot \sin(\phi_i), \tag{23}$$

and the overall amplitude $A_i$ and phase $\phi_i$ are reconstructed as:

$$A_i = \sqrt{A_{\cos,i}^2 + A_{\sin,i}^2}, \quad \phi_i = \tan^{-1}\left(\frac{A_{\sin,i}}{A_{\cos,i}}\right). \tag{24}$$

The amplitude and frequency of the channeling spectra were identified using a FFT applied to the residuals between the calculated and observed HBr spectra, transforming the data from the wavenumber domain to the frequency domain as shown in the Fig. 6(b). Initially, an ideal instrument state was assumed in the LINEFIT software, setting the amplitude and frequency of channeling spectra to zero. The FFT results provided initial estimates for these parameters, which LINEFIT iteratively refined to accurately determine their values. Once the baseline fitting process converged, the channeling parameters were

further adjusted, resulting in an ILS profile that reflected the spectrometer's non-ideal characteristics. LINEFIT also eliminated the channeling spectra, ensuring these artifacts did not interfere with the analysis of the transmittance spectra. The general workflow, encompassing the entire experimental setup, data processing, and iterative analysis, is illustrated in the Fig. 7.

### 3.9   Performance Evaluation Metrics

   To compare the performance of the nominal and modified configurations, several metrics were used to evaluate the quality

of the spectral simulations. These metrics focus on the residuals, which represent the difference between the observed and simulated spectra, and provide a quantitative assessment of the simulation's accuracy. The following equations define the metrics used in this study:

Maximum Residual:      $\text{Max Residual} = \max(|r_i|)$

Residual Range:      $\text{Residual Range} = \max(r_i) - \min(r_i)$

Standard Deviation:      $\sigma = \sqrt{\frac{1}{N-1}\sum_{i=1}^{N}(r_i - \bar{r})^2}$

Mean Absolute Error:      $\text{MAE} = \frac{1}{N}\sum_{i=1}^{N}|r_i|$

Root Mean Square:      $\text{RMS} = \sqrt{\frac{1}{N}\sum_{i=1}^{N}r_i^2}$

Correlation Coefficient:      Pearson correlation coefficient

   These metrics were applied systematically to quantify the differences between observed and simulated spectra. By focusing

on residuals, the analysis captured the variations introduced by instrumental non-idealities and highlighted the improvements achieved in the modified configuration. This evaluation forms the basis for understanding the effectiveness of corrections applied to the spectrometer's setup.

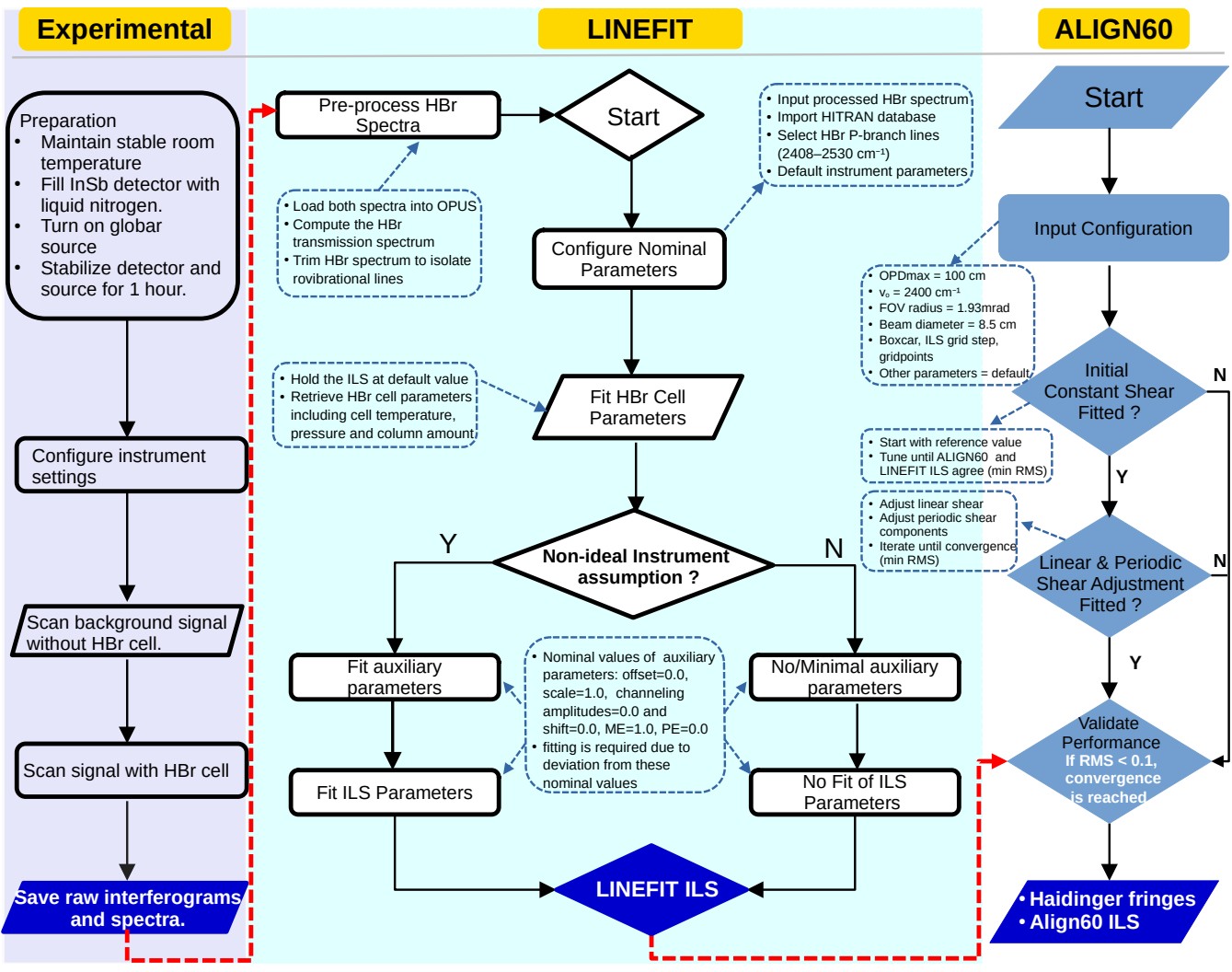

**Figure 7.** Workflow for Instrumental Characterization: integrating experimental setup, data Processing with LINEFIT and ALIGN60, and iterative optimization for ILS and Haidinger Fringe Simulations.

The signal-to-noise ratio (S/N) measures the strength of a signal relative to background noise, offering an essential metric for assessing spectral data quality. For the P(2) line in the HBr transmittance spectrum, the S/N was calculated by:

$$\text{S/N} = \frac{S}{\sigma_{\text{noise}}}$$

where $S$ represents the mean of the fitted signal in the region of interest, and $\sigma_{\text{noise}}$ denotes the RMS of the residuals, obtained after subtracting a linear fit from the measured spectrum over the same interval. This approach provides a more accurate estimate of random noise by first accounting for the underlying signal trend.

Asymmetry of the ILS is an important characteristic for evaluating deviations from an ideal symmetric profile. For a line shape with sidelobes, the asymmetry can be assessed by comparing the heights of the first sidelobes on either side of the central peak. We calculated the asymmetry as the magnitude of the height difference between the two first sidelobes, normalized to the height of the central peak. Mathematically, the asymmetry is given by (e.g. García et al., 2022):

$$A_{\text{ILS}} = \frac{|H_{\text{left}} - H_{\text{right}}|}{H_{\text{peak}}}$$

where $H_{\text{left}}$ is the height of the first sidelobe on the left side of the central peak, $H_{\text{right}}$ is height of the first sidelobe on the right side of the central peak and $H_{\text{peak}}$ is the height of the central peak. Both the sidelobe heights used in the asymmetry calculation and the amplitudes of the successive interferogram maxima were determined after applying Fourier interpolation onto a finer OPD and wavenumber grid, ensuring that sampling effects are consistently minimized and do not bias any of the diagnostic metrics.

The Full Width at Half Maximum (FWHM) of the ILS was calculated by determining the positions where the ILS crossed half of its maximum intensity. Linear interpolation was used between adjacent data points to estimate the exact crossing positions. The FWHM was then computed as:

$$\text{FWHM} = x_{\text{right}} - x_{\text{left}}$$

where $x_{\text{left}}$ and $x_{\text{right}}$ represent the positions on either side of the central peak where the ILS crossed the half-maximum threshold.

To analyze the rate of change of the ILS, its first derivative was calculated numerically. The derivative was computed using the finite difference method:

$$\frac{d\text{ILS}}{dx} \approx \frac{\Delta\text{ILS}}{\text{step}}$$

where $\Delta\text{ILS}$ is the difference between consecutive points in the ILS data, and step is the sampling interval along the x-axis. This derivative provides insight into the slope and behavior of the ILS, particularly around the peak and sidelobes.

To quantify the relative deviation between modified and measured values of the ILS or other metrics such as $\sigma$ and RMS, we calculated the percentage difference. The formula used is:

$$\text{Percentage Difference} = \frac{M_{\text{modified}} - M_{\text{nominal}}}{M_{\text{nominal}}} \times 100$$

where $M_{\text{modified}}$ is the value of the metric under modified configuration and $M_{\text{nominal}}$ is the value of the metric under nominal configuration.

## 3.10 Demonstrating Sensitivity of Atmospheric Retrievals to Instrument Characterization

### 3.10.1 Selection of Case Study

To demonstrate the practical implications of instrument characterization for atmospheric trace gas retrievals, a focused case study was conducted to quantify how instrumental non-idealities propagate into retrieval results and bias atmospheric inferences when left unaccounted for. The analysis is based on solar absorption measurements acquired on December 14[th], 2012, a date deliberately selected because it follows the major instrument upgrade and coincides with the low-pressure gas cell characterization experiments, thereby ensuring internal consistency between the diagnosed instrumental state and the atmospheric observations. The solar absorption spectrum was recorded at a solar zenith angle (SZA) of $22.6°$, as defined by the solar beam geometry employed in the forward model of the retrieval code ($\mu = \cos(\text{SZA}) = 0.92379$), using the second NDACC narrow bandpass filter. Ethane ($C_2H_6$) was selected as the target species owing to its moderate absorption strength in the mid-infrared and its documented sensitivity to ILS distortions, making it particularly suitable for assessing retrieval impacts arising from non-ideal instrument behavior (e.g. Sun et al., 2018b; Zifarelli et al., 2023). The retrieval was performed using three narrow microwindows (2976.64-2976.96 cm$^{-1}$, 2983.18-2983.54 cm$^{-1}$, and 2986.51-2986.95 cm$^{-1}$) recommended by the NDACC-IRWG, which are optimized for $C_2H_6$ retrievals while maintaining robustness against spectroscopic interference (e.g. Zifarelli et al., 2023). Interfering species within these windows, notably $H_2O$, $O_3$, and $CH_4$, were explicitly accounted for in the forward model to minimize spectroscopic interference and ensure a robust retrieval of the $C_2H_6$ columns, consistent with established NDACC retrieval practice.

### 3.10.2 Forward Model Configuration

The forward model provides the physical link between the atmospheric state vector and the simulated solar absorption spectrum and therefore determines how instrumental characteristics influence atmospheric retrievals (Rodgers, 2000). In this study, spectral simulations were performed using the PROFFWD (PROFile ForWarD) model within the PROFFIT (PROFile FIT) version 9.6 retrieval code (Hase et al., 2004). The model accounts for line-by-line molecular absorption, pressure and temperature broadening, and instrument-dependent spectral effects. Spectroscopic parameters were taken from HITRAN 2009 for $H_2O$, $O_3$, and $CH_4$, while $C_2H_6$ parameters were adopted from the Toon compilation (Rothman et al., 2009). A priori volume-mixing-ratio (VMR) profiles were provided by Whole Atmosphere Community Climate Model version 6 (WACCM6) (Gettelman et al., 2019). Pressure and temperature profiles were obtained from analyzed meteorological fields produced by the National Centers for Environmental Prediction (NCEP) Climate Prediction Center (CPC) and provided through the NASA Goddard Space Flight Center, and were interpolated to the measurement location and time of observation (De Mazière et al., 2018; Kalnay et al., 2018). The atmosphere was discretized into 44 fixed altitude levels extending from surface to 120 km, following the standard PROFFIT vertical grid definition and assuming hydrostatic equilibrium between levels. The nominal forward-model configuration adopted the nominal instrument characterization described above and retained the continuum scaling at unity. In contrast, the modified configuration iteratively fitted these parameters, thereby explicitly propagating empirically characterized non-ideal instrumental effects into the simulated spectra. All other instrumental parameters, including the MOPD, apodization

function, and FOV, were held fixed and identical for both configurations to ensure that observed differences arose solely from the treatment of instrumental non-idealities.

### 3.10.3 Retrieval Diagnostics Used for Evaluation

Retrievals of $C_2H_6$ profiles were performed by fitting simulated solar absorption spectra to the measured spectra using an iterative inversion, in which the Jacobian matrix relates perturbations in the atmospheric state vector and instrumental parameters to changes in spectral radiance, thereby propagating both atmospheric and instrumental sensitivities into the solution (Rodgers, 2000; Hase, 2012). The inversion was carried out with the PROFFIT retrieval code, which addresses the inherently non-linear least-squares problem through linearization about the current state and iterative minimization of the cost function. $C_2H_6$ was retrieved in logarithmic space using a Tikhonov–Phillips regularized inversion, while interfering $H_2O$ profiles were treated via a priori scaling to account for strong spectroscopic overlap within the selected microwindows. When instrument performance is adequately represented by the nominal characterization adopted in the forward model, retrievals are not expected to be significantly influenced by instrumental effects (e.g. von Clarmann et al., 2019). Departures from this nominal characterization, however, introduce systematic spectral distortions through the Jacobian matrix that propagate into the retrieval, leading to underestimation or overestimation of retrieval outputs (e.g. Sun et al., 2018b). Consequently, rigorous instrument characterization and the explicit simulation of corrected ILS, spectral channeling, and baseline distortions are essential to ensure retrieval accuracy and long-term consistency.

### 3.10.4 Observed Impact of Instrument Characterization

The impact of instrument characterization on atmospheric retrievals was evaluated by comparing retrieval outcomes obtained using the nominal and modified instrument configurations for the selected case study. For demonstration purposes, a representative microwindow covering the spectral range 2976.64-2976.96 $cm^{-1}$ was selected to illustrate the sensitivity of the simulated spectra to instrumental effects. Within this microwindow, the analysis was performed using normalized intensity, and residual differences between the measured and simulated solar absorption spectra were calculated as the difference between the observed and simulated spectra after normalization. These residuals were quantified using the same statistical metrics applied to the HBr gas cell analysis in Section 3.9, thereby ensuring methodological consistency between the instrument characterization diagnostics and the atmospheric forward-model evaluation. Systematic reductions in residual amplitude and dispersion under the modified configuration indicate improved spectral consistency when empirically characterized instrumental effects are propagated into the forward model. Retrieval uncertainty for each configuration was assessed by combining the total systematic and total random errors using a quadrature sum. The resulting retrieval error, initially expressed in absolute units (ppm), was subsequently converted to a relative percentage by normalizing with respect to the integrated VMR over the retrieval altitude range. The retrieved total column is reported in absolute physical units for both the nominal and modified configurations, while their relative difference is quantified using the percentage-difference formulation defined in Section 3.9, with the nominal configuration taken as the reference.

## 4   Results and Data Analysis

The figures shown in the Fig. 8 - 11 illustrate the temporal evolution of the FTIR instrument's performance over time, based on data collected from 091208 to 161021. Fig. 8, generated from the background measurement and apodized using the boxcar apodization function, shows changes in interferogram intensity for both forward and backward scans, particularly near the ZPD. This variation highlights the impact of retroreflector misalignment, source brightness fluctuations, and the aging of optical components. Fig. 10 provides a detailed view of the ZPD region, showing variability in peak intensity and sampling over time, particularly in later scans like 150317, reflecting alignment instability. Fig. 9, which includes 512 sampling points with a broken x-axis for clarity, reveals discrepancies between forward and backward scans, suggesting mechanical drift and optical misalignments. Fig. 11 illustrates the evolution of the real and imaginary components of the complex interferogram intensity at the ZPD and its maxima. Over time, both components show variability, with the 3rd maximum showing the greatest reduction in scans such as 150317 and 161021. These observed changes, stemming from alignment and component issues, impact the modulation efficiency and phase accuracy, influencing the overall quality of spectral data and emphasizing the need for regular instrument maintenance and recalibration. Fig. 12 further demonstrates the temporal degradation by comparing transmission spectra from the Globar source, both without (a) and with (b) the HBr absorption cell. The background spectra show a progressive decline in transmission baseline and increasing non-uniformity over time, while the absorption spectra reveal corresponding distortions in line shape and depth, particularly in later scans. These results confirm that instrumental degradations propagate directly into sample measurements, reducing spectral fidelity.

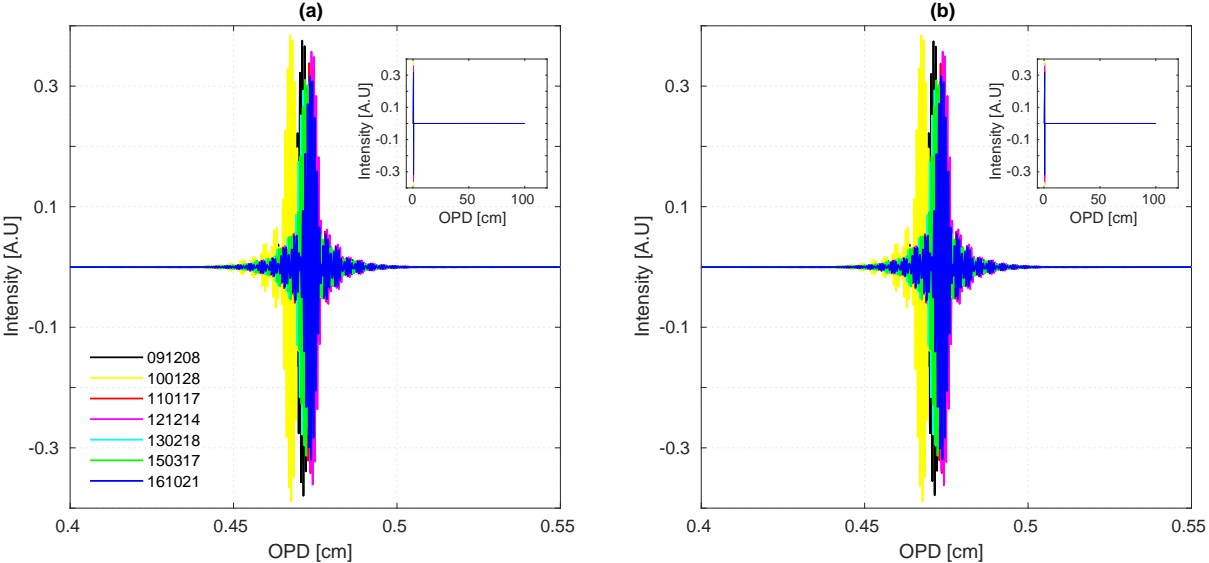

**Figure 8.** Apodized raw interferograms from background measurements using the boxcar apodization function, showing (a) forward scans and (b) backward scans, illustrating the evolution of intensity over time from 091208 to 161021, with noticeable degradation near the ZPD.

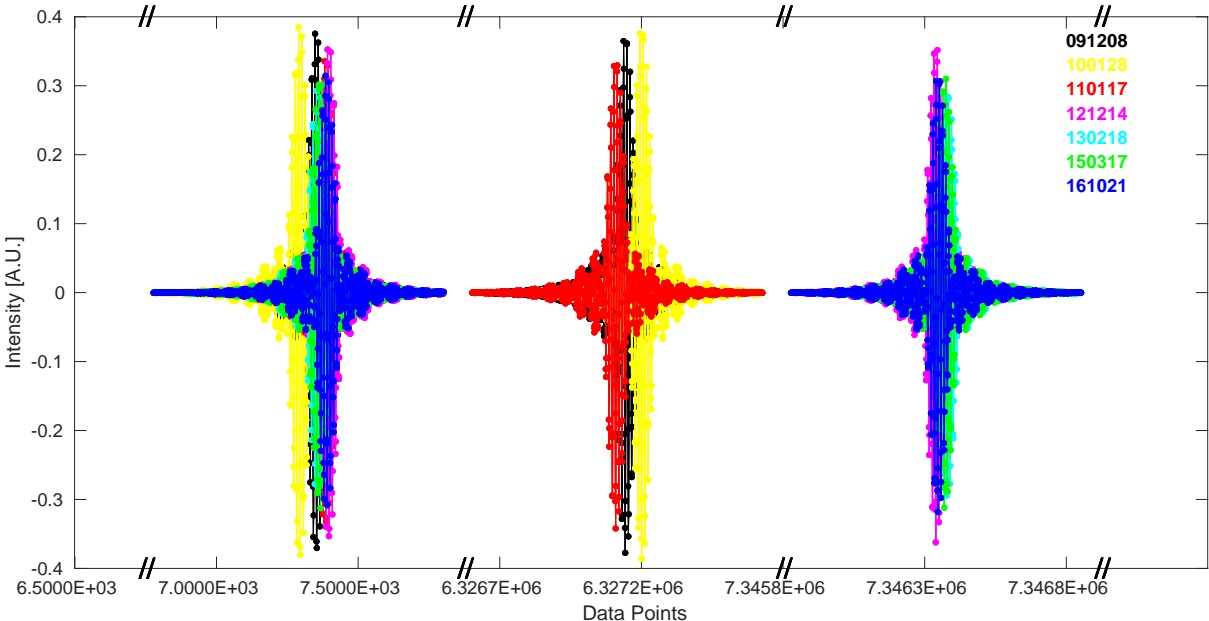

**Figure 9.** Sampling point analysis of apodized raw interferograms with 512 data points plotted for both forward scans and backward scans, with a broken x-axis for better visualization.

The HBr transmission spectra presented in Fig. 13 demonstrate high correlation coefficients ($R^2 > 0.99$) for both the nominal and modified configurations, indicating strong agreement with the observed spectra across all dates. However, the modified configuration consistently achieves slightly higher fidelity in capturing instrumental characteristics, as evidenced by its

improved residual behavior. This is particularly apparent in the residual analysis, where the nominal configuration displays greater variability and higher error metrics across all cases. In contrast, the modified configuration yields systematic reductions in residual range, $\sigma$, and MAE, with improvements as high as >5% on challenging dates such as 150317 and 161021. On 150317, for example, the residual MAE decreases from $0.0795 \times 10^{-2}$ by 5.92%, and the range shifts from $2.13 \times 10^{-2}$ (nominal) by $-2.33\%$ (modified), reflecting the impact of correction. Similarly, the maximum residual difference on 121214

increases from $0.979 \times 10^{-2}$ by 3.03%, underscoring how nominal simulations can underrepresent spectral deviations around peak regions. Earlier dates such as 100128 and 110117 exhibit mixed behavior: while the residual range and maximum values are sometimes comparable or slightly higher for the modified configuration, the MAE shows marked improvement (e.g., $-1.41\%$ on 100128 and $-5.93\%$ on 110117), indicating enhanced modeling accuracy despite small fluctuations in dynamic range. Residuals are most pronounced around absorption peaks—where baseline drift and channeling dominate—while flat-

ter transmission regions show minimal discrepancies, indicating both models capture the continuum well. The trends across dates consistently support the robustness of the modified configuration in compensating for instrumental non-idealities. These systematic improvements, especially under difficult alignment or measurement conditions, highlight the importance of incorporating correction algorithms in spectroscopic simulations to achieve higher fidelity with observed spectra.

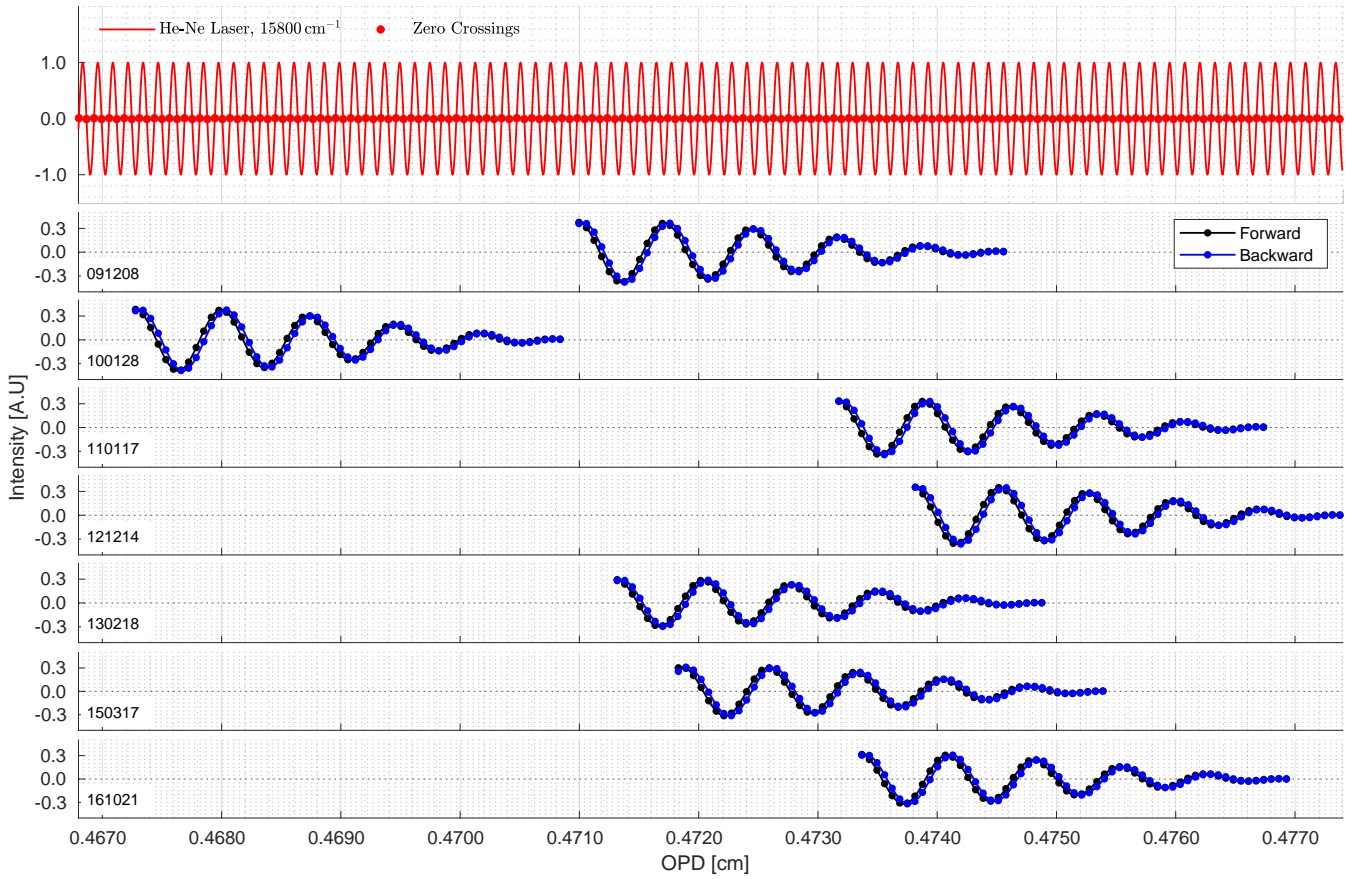

**Figure 10.** Zoomed view of the apodized raw interferograms near the ZPD for background measurements, showing both forward scans and backward scans. The figure includes the simulation of He-Ne laser reference signal at 15800 cm$^{-1}$ and its zero crossings, which serve as a timing reference for sampling.

Given the deviations observed across the full spectral range, the the higher S/N P(2) line is selected as a representative case to illustrate the impact of the modified configuration, as shown in Fig. 14. The adjustment of baseline and channeling parameters in the modified configuration has resulted in improvements in the spectral data quality for the P(2) line of HBr, as demonstrated by the $\sigma$ and RMS values. Under the nominal configuration, the residuals showed slight but systematic deviations, with both $\sigma$ and RMS values within the range $(0.49\text{–}0.67) \times 10^{-3}$ (Fig. 14, middle panels). These values indicated substantial systematic errors due to baseline shifts and channeling effects, which negatively impacted the accuracy of the nominal simulations. In contrast, the modified configuration achieves considerably reduced RMS values in almost all later years, often by 5–12%, and for the 161021 dataset by more than 18%, signifying a notable decrease in the overall discrepancies between the observed and simulated spectra (Fig. 14, bottom panels). This improvement is crucial because it indicates that the modified configuration has successfully corrected the systematic errors, resulting in a more accurate representation of the HBr isotopic lines. For the pre-maintenance dates, the RMS increases by up to 12.8%, indicating that the nominal configuration might have underestimated

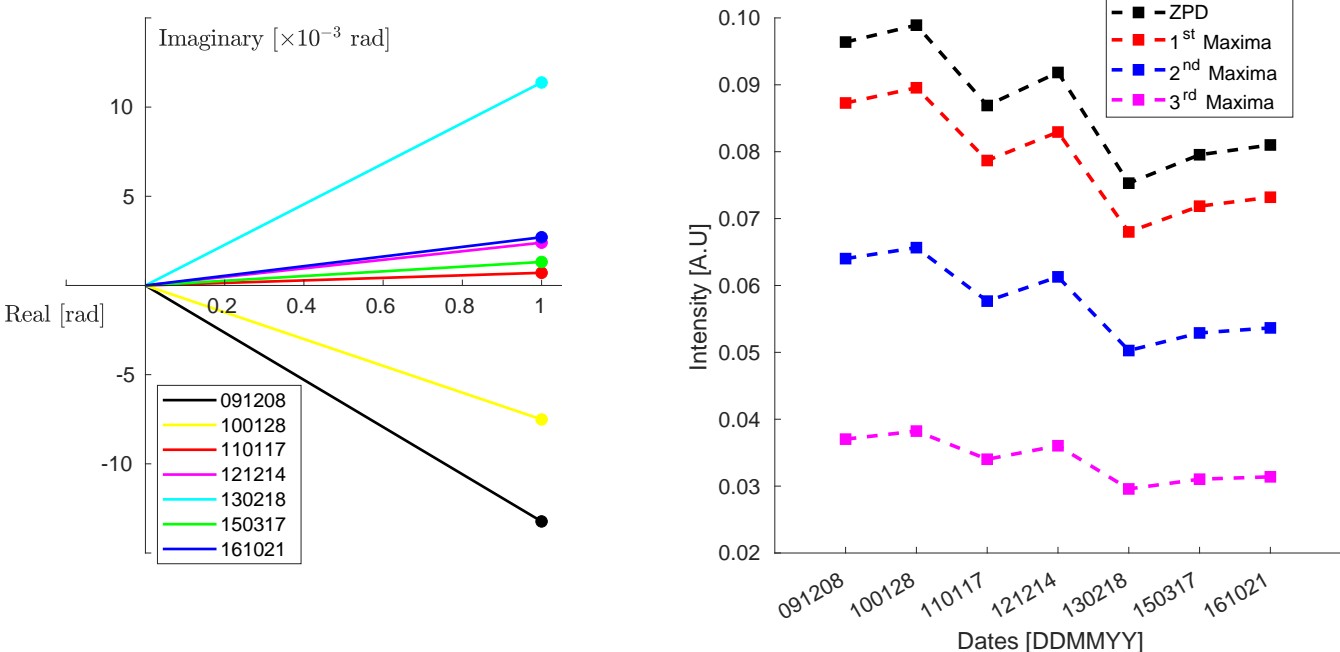

**Figure 11.** Evolution of the complex interferogram intensity at the ZPD and its maxima over time (left: real vs. imaginary components; right: intensity at ZPD, 1st, 2nd, and 3rd maxima).

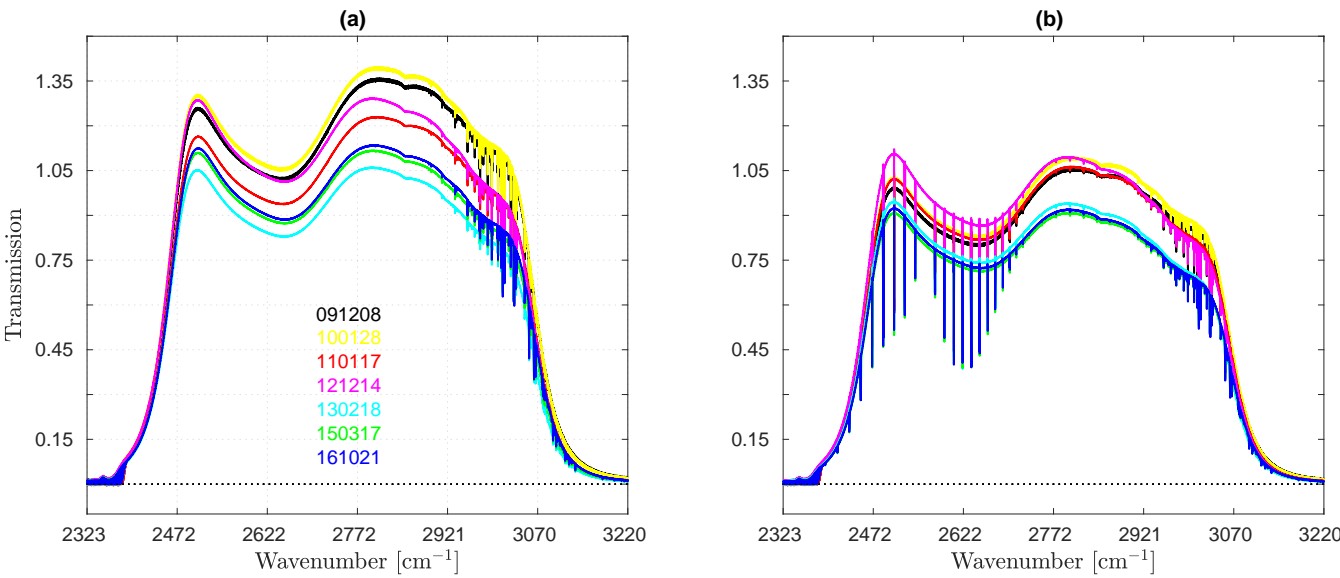

**Figure 12.** Temporal evolution of FTIR transmission spectra with the Globar source over multiple dates, (a) Background spectra recorded without a sample cell, (b) Corresponding spectra with HBr absorption cell in the path.

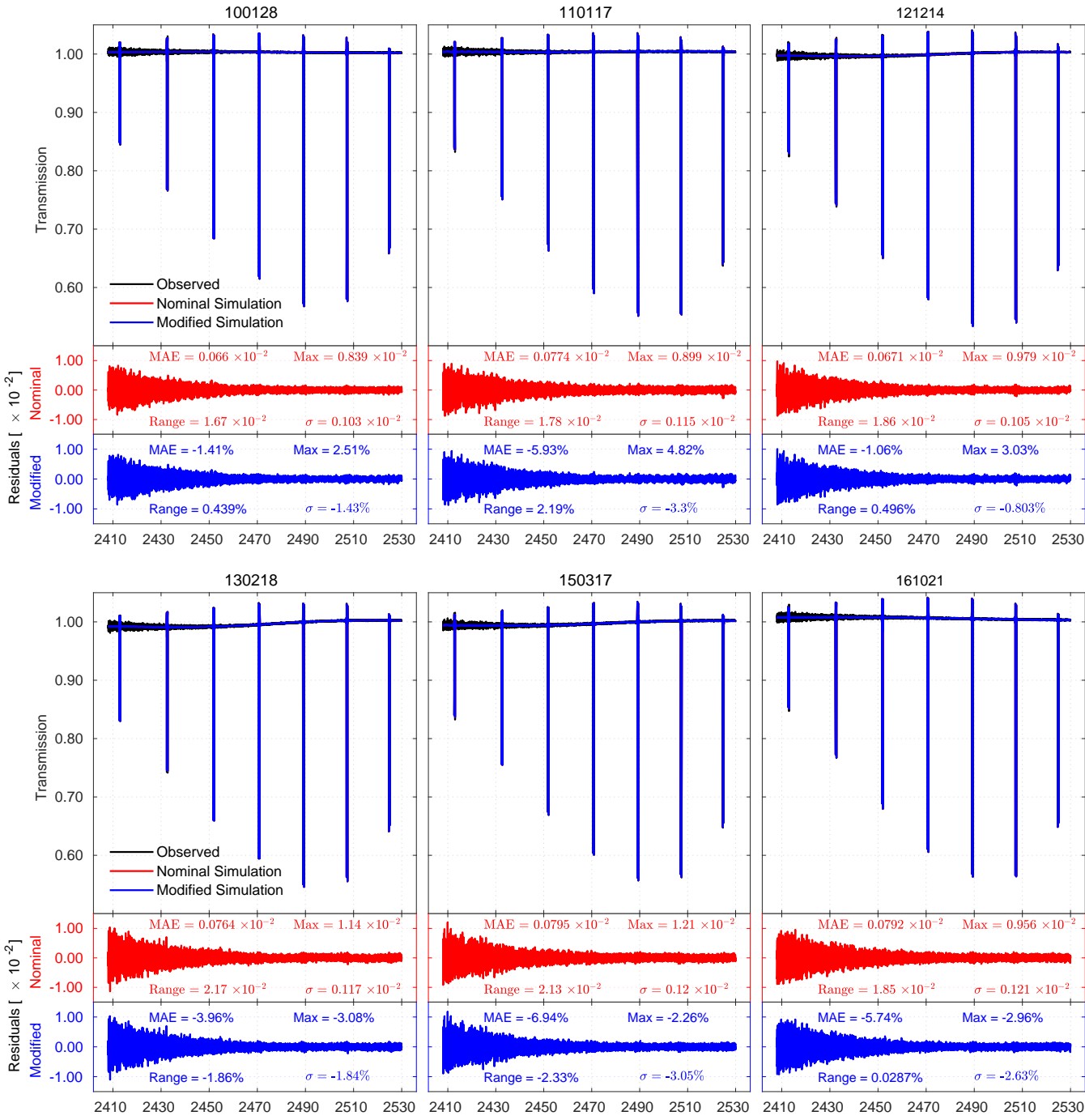

**Figure 13.** Comparison of observed, nominal, and modified HBr transmission spectra with residual analysis across different dates.

the true residual difference between the measured and calculated spectra. Moreover, the S/N values for the observed spectra range from approximately 18.622 to 19.994, with the nominal and modified simulations showing slightly different S/N values (Fig. 14, top panels). The close alignment of these values suggests that the modified configuration has effectively captured the essential features of the observed spectra, with minimal deviation, indicating a stable performance of the spectrometer and a reliable simulation procedure.

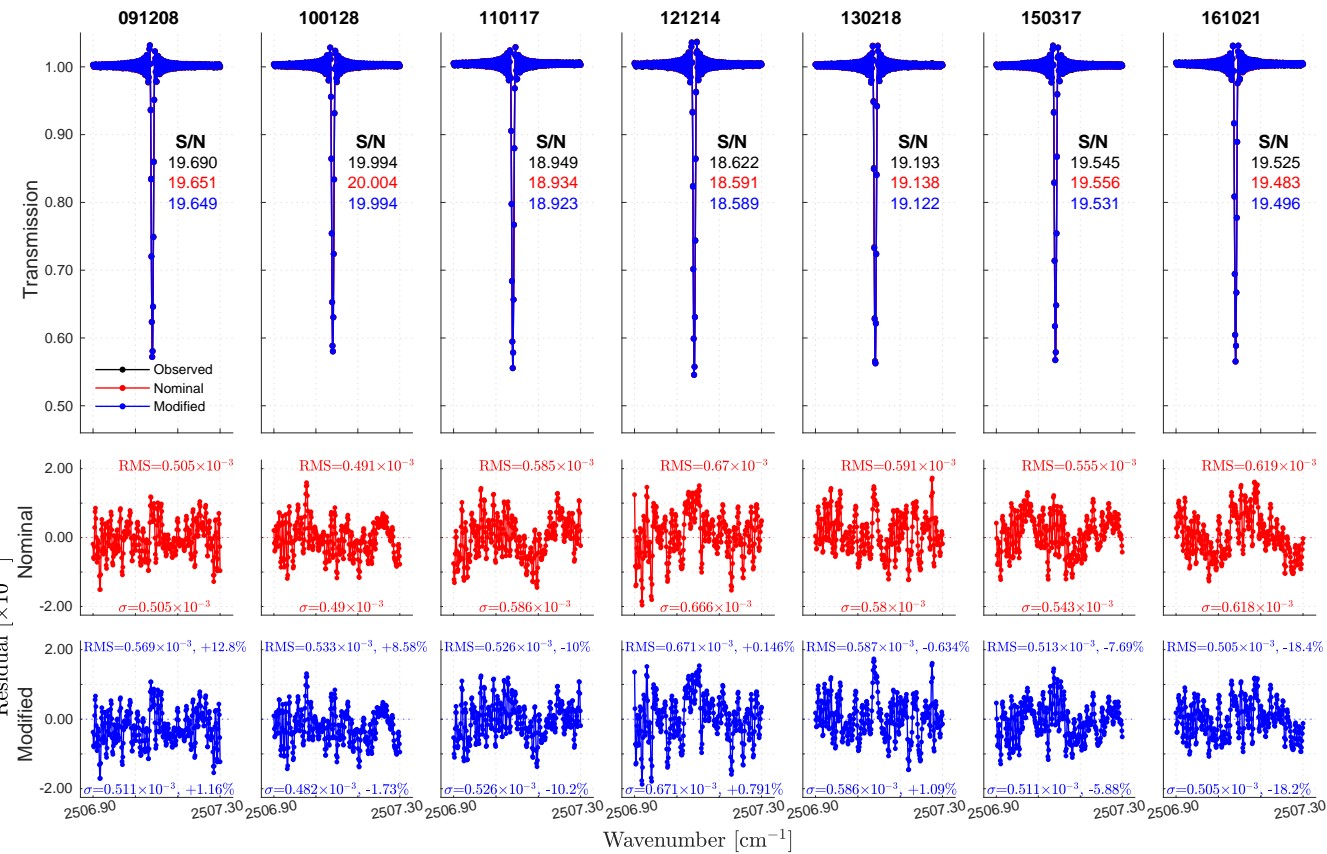

**Figure 14.** The top row shows the P(2) lsidelobe, fitted, and reconstructed spectra across different dates. The middle and bottom rows show residual plots comparing the deviations before and after fitting, respectively.

Fig. 15 and Table 3 illustrate the spectral channeling behavior of the instrument over seven years, revealing both stable and variable channeling frequencies influenced by its optical configuration and mechanical components. The consistent detection of the 2.9750 cm$^{-1}$ frequency across multiple dates strongly suggests it is an inherent characteristic of the instrument, likely arising from its optical design or fixed mechanical elements, such as the retroreflector geometry or field stop. During the first year, this frequency was the sole artifact observed, indicating the instrument was operating near nominal alignment with

minimal spectroscopic artifacts or mechanical instabilities. In subsequent years, an additional frequency around 0.2426 cm$^{-1}$ began to emerge, likely reflecting the onset of misalignment, gradual wear in the retroreflector system, or potential degradation

of the $CaF_2$ beamsplitter over time. A drastic shift in channeling frequency recorded on 121214, where a value of $40.670 \, \text{cm}^{-1}$ was observed, stands out as a significant anomaly. This outlier may be attributed to external influences such as environmental factors (e.g., vibrations or temperature fluctuations), adjustments to the experimental setup (e.g., field stop or aperture changes),

or temporary misalignment or mechanical instability in the retroreflector. The trends observed over the seven years suggest a combination of gradual misalignment and occasional abrupt disruptions, with some recurring frequency patterns tied to the instrument's design and anomalies like the one on 121214 reflecting unpredictable factors. Systematic biases, including minor alignment deviations, imperfections in the $CaF_2$ beamsplitter, or other optical components, likely also contribute to the observed frequencies.

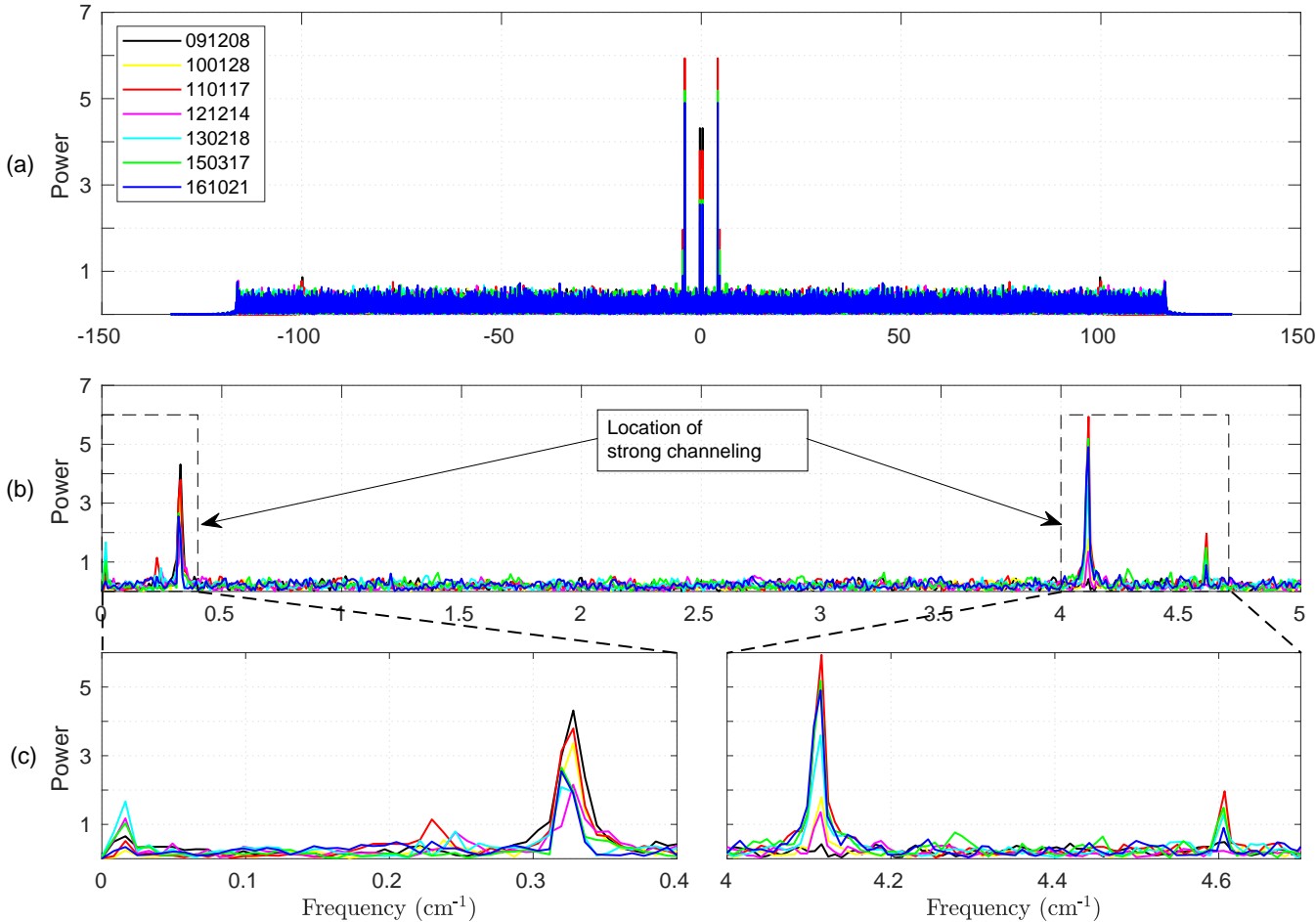

**Figure 15.** Initial guess of the channeling locations based on long-term spectral channeling behavior of the Bruker FTS 120M spectrometer at Addis Ababa, illustrating evolving frequencies: (a) double-sided FFT of the residual difference between measured and nominally simulated spectra, (b) single-sided spectrum emphasizing the low-frequency region (0-5 cm) and (c) magnified views of the boxed regions in (b) around $0 - 0.4$ cm (left) and $4 - 4.7$ cm (right), highlighting the spectral fingerprints of strong channeling.

| Quantity | 091208 | 100128 | 110117 | 121214 | 130218 | 150317 | 161021 |
|---|---|---|---|---|---|---|---|
| channeling frequencies, $f_i$ | 2.9750 | 2.9750 | 0.2425 | 2.9760 | 0.2426 | 0.2426 | 0.2426 |
| | 0.0000 | 0.2425 | 2.9750 | 0.2426 | 40.670 | 3.0500 | 3.0500 |
| | 0.0000 | 0.2167 | 0.2167 | 40.670 | 3.0500 | 0.2167 | 0.2167 |
| | 0.0000 | 0.0000 | 4.2060 | 2.7730 | 0.2167 | 40.670 | 0.0000 |
| $A_{\sin,i} \, (\times 10^{-4})$ | -0.3472 | -1.1670 | 3.0240 | -0.7087 | 1.1890 | 2.0240 | -2.1940 |
| | 0.0000 | 0.0989 | 2.3470 | 0.9358 | -0.5369 | 1.3390 | 0.8579 |
| | 0.0000 | 0.0222 | 0.0548 | -0.3457 | 1.2250 | -0.8114 | 0.0439 |
| | 0.0000 | 0.0000 | 0.7063 | 0.4043 | -0.2030 | -0.2546 | 0.0000 |
| $A_{\cos,i} \, (\times 10^{-4})$ | -2.6460 | -1.7080 | 0.5784 | -1.1320 | -2.2660 | -3.0550 | 2.7160 |
| | 0.0000 | 0.9341 | -0.0663 | -0.1406 | 1.2060 | -0.9982 | -1.3560 |
| | 0.0000 | -0.8834 | 1.2170 | 0.8671 | 0.4502 | -0.4337 | -0.5457 |
| | 0.0000 | 0.0000 | -0.0823 | 0.3251 | -0.7702 | 0.7870 | 0.0000 |
| Total Amplitude, $A_i \, (\times 10^{-4})$ | 2.6690 | 2.0690 | 3.0790 | 1.3360 | 2.5590 | 3.6650 | 3.4910 |
| | 0.0000 | 0.9394 | 2.3480 | 0.9463 | 1.3200 | 1.6700 | 1.6040 |
| | 0.0000 | 0.8837 | 1.2180 | 0.9335 | 1.3050 | 0.9200 | 0.5474 |
| | 0.0000 | 0.0000 | 0.7110 | 0.5188 | 0.7965 | 0.8272 | 0.0000 |

**Table 3.** Final channeling frequency locations after iterative fitting, extracted channeling frequencies and corresponding Fourier coefficients from residual interferograms measured by the Bruker FTS 120M spectrometer over multiple years. For each measurement date, the table lists the identified channeling frequencies $f_i$ (in cm$^{-1}$), along with their associated sine and cosine Fourier amplitudes $A_{\sin,i}$ and $A_{\cos,i}$, respectively. The total amplitude $A_i = \sqrt{A_{\sin,i}^2 + A_{\cos,i}^2}$ quantifies the strength of each channeling component.

Fig. 16 represents the ILS retrieval of the modified configuration, highlighting the instrument's sensitivity in resolving the true ILS, with deviations from a sharp diagonal structure indicating reduced sensitivity and cross-influences caused by instrumental non-idealities, such as retroreflector misalignment. The temporal evolution of the kernel, observed across dates from 091208 to 161021, reveals decreasing sensitivity over time, as evidenced by the broadening of diagonal peaks and increasing off-diagonal contributions. A comparison of different paths, such as ZPD and MOPD, across these dates indicates that

misalignment effects are most pronounced near the ZPD, where the averaging kernel shows broader off-diagonal responses, reflecting maximum spectral smearing and reduced sensitivity due to higher mechanical and optical misalignment in this region. Conversely, the kernel near MOPD exhibits sharper diagonal structures with minimal off-diagonal contributions, indicating the least misalignment and a more stable instrument response at larger optical path differences. Within individual dates, such as 121214 or 150317, this trend is consistent, with the ZPD path demonstrating greater deviations from ideality compared to the

MOPD path. These observations underscore the critical importance of addressing misalignments near the ZPD to improve ILS accuracy and ensure the reliability of high-resolution FTIR spectroscopy measurements.

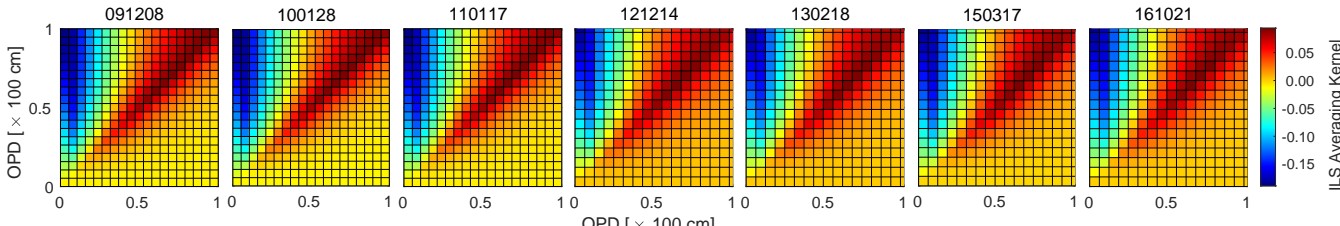

**Figure 16.** The averaging kernel of the Modified ILS function

The Haidinger fringes simulated under the nominal configuration using ALIGN60, as shown in Fig. 17, consistently display patterns near the ZPD across multiple dates, with slight year-to-year variations in brightness. These variations, though minor, may indicate underlying non-idealities in the instrument, such as aging components or subtle calibration shifts, that are not fully captured in the nominal configuration. Notably, the fringe pattern for 121214 exhibits reduced brightness at the center, suggesting transient or systematic deviations from the assumed ideal conditions, potentially due to optical misalignments or environmental factors. This highlights the limitations of the nominal configuration in accounting for all instrumental imperfections. While these simulations provide a valuable baseline, the observations underscore the necessity of addressing non-idealities beyond lateral shear to enhance the accuracy of solar absorption spectra retrievals for CO, $C_2H_6$, HCN, and $C_2H_2$.

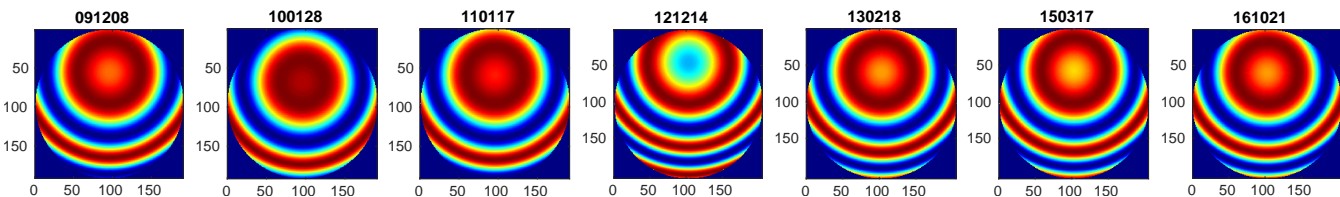

**Figure 17.** Haidinger fringe patterns near ZPD simulated under nominal configuration using ALIGN60 across multiple dates

Both the phase error (PE) and modulation efficiency (ME) show significant improvement in the modified retrieval configuration, demonstrating the effectiveness of addressing instrumental non-idealities, Fig. 18. In the nominal configuration, the ME ranged from $-0.06\%$ to $+10.9\%$, and the PE varied between $-2.110 \times 10^{-2}$ rad and $0.838 \times 10^{-2}$ rad, influenced by lateral shear effects that caused overmodulation due to periodically varying cosine bending misalignments. The instrument performed best in 2011, attributed to a realignment by Bruker Optics personnel during the replacement of the reference HeNe laser (reference: personal communication), which optimized its alignment and performance. However, since 2012, operation by different researchers and uncertainties in measurement techniques likely contributed to alignment evolution, while the initial misalignment in 2009 can be linked to adjustments after shipping and installation. In the modified configuration, the ME range improved to $-0.08\%$ to $+8.40\%$, and the PE range tightened to $-2.200 \times 10^{-2}$ rad to $0.765 \times 10^{-2}$ rad. For example, on 121214, the ME improved from $+10.9\%$ (nominal) to $+8.40\%$ (modified), and the PE decreased from $0.081 \times 10^{-2}$ rad (nominal) to $0.042 \times 10^{-2}$ rad (modified), as shown in the Table 4. Importantly, the ME values in the modified configuration consistently

fell within the NDACC-acceptable limit of 1.1, ensuring reliable and accurate data retrievals. These findings highlight the importance of realignment and parameter adjustments in mitigating non-idealities and maintaining long-term instrument accuracy. The improvements in ME and the averaging kernel of the ILS are directly reflected in changes to key ILS parameters,

| Dates | ME (%) | | PE $\times 10^{-2} rad$ | | FWHM of ILS | | Asymmetry of ILS | |
|---|---|---|---|---|---|---|---|---|
| YYMMDD | Nominal | Modified | Nominal | Modified | Nominal | Modified | Nominal | Modified |
| 091208 | +3.40 | +2.50 | -2.110 | -2.200 | 0.0063 | 0.0064 | 0.1007 | 0.0994 |
| 100128 | +1.50 | +1.00 | -0.936 | -0.926 | 0.0064 | 0.0064 | 0.0999 | 0.0993 |
| 110117 | -0.06 | -0.08 | 0.537 | 0.419 | 0.0064 | 0.0064 | 0.0994 | 0.0993 |
| 121214 | +10.9 | +8.40 | 0.081 | 0.042 | 0.0055 | 0.0055 | 0.0948 | 0.0937 |
| 130218 | +7.50 | +5.60 | 0.838 | 0.765 | 0.0055 | 0.0056 | 0.0949 | 0.0936 |
| 150317 | +6.90 | +5.10 | -0.190 | -0.251 | 0.0055 | 0.0056 | 0.0928 | 0.0918 |
| 161021 | +6.30 | +4.60 | 0.478 | 0.315 | 0.0056 | 0.0056 | 0.0912 | 0.0909 |

**Table 4.** Summary of ME, PE at maximum OPD, and the FWHM and asymmetry of the ILS. The modulation column reflects gain (+) or loss (-) in efficiency. Results are shown for both nominal and modified configurations, with dates formatted as YYMMDD.

including peak height, FWHM, and sidelobe asymmetry, underscoring the connection between instrumental enhancements and spectral fidelity. The ILS shown in the top panel of Fig. 19 reveals two distinct temporal trends: 2009-2011 (former years) and 2012-2017 (later years), likely linked to the upgrade of the MOPD in 2011. This upgrade significantly influenced the ILS parameters, resulting in higher peak heights, narrower FWHM, and reduced sidelobe asymmetry, as shown in Fig. 19 and detailed in Table 4. The earlier years exhibit lower peak heights, averaging $178.80 \pm 1.31$ cm for the nominal configuration and $178.67 \pm 0.96$ cm for the modified configuration, compared to $205.40 \pm 2.11$ cm and $203.88 \pm 1.65$ cm in the later years, reflecting increases of approximately 14.88% and 14.11%, respectively. The FWHM narrows from $0.0064 \, \text{cm}^{-1}$ in the earlier years to $0.0055 \, \text{cm}^{-1}$ in the later years, indicating improved resolution and sharper spectral features with insignificant variation between configuration settings. Additionally, sidelobe asymmetry of the nominal setting decreases from $0.100 \pm 0.0007$ in earlier years to $0.0934 \pm 0.0018$ in later years, with further reductions to $0.0993 \pm 0.0001$ to $0.0925 \pm 0.00137$ of the same years under the modified configuration, demonstrating enhanced symmetry and reduced distortions. The first derivative of the ILS, as shown in the lower panel of Fig.19, reveals percentage differences (shown in bottom panel of Fig.19) between the nominal and modified configurations, with most of the differences falling within $\pm 4\%$. However, in the sidelobe regions, the percentage difference reaches up to $\sim 6\%$, which can be attributed to the greater sensitivity of these regions to changes in asymmetry. This result highlights that the modified configuration effectively reduces distortions caused by non-idealities by improving the symmetry of the ILS, particularly in areas where asymmetry has a pronounced impact on the instrument's response. These advancements, driven by the MOPD upgrade and improved ME, enhance the fidelity of the ILS, ensuring sharper spectral lines, better resolution, and more accurate instrument performance.

While the preceding analysis focused on instrument performance characterization, Fig. 20 and Table 5 demonstrate how this characterization directly translates into atmospheric retrieval results. Fig. 20 illustrates the application of the nominal and

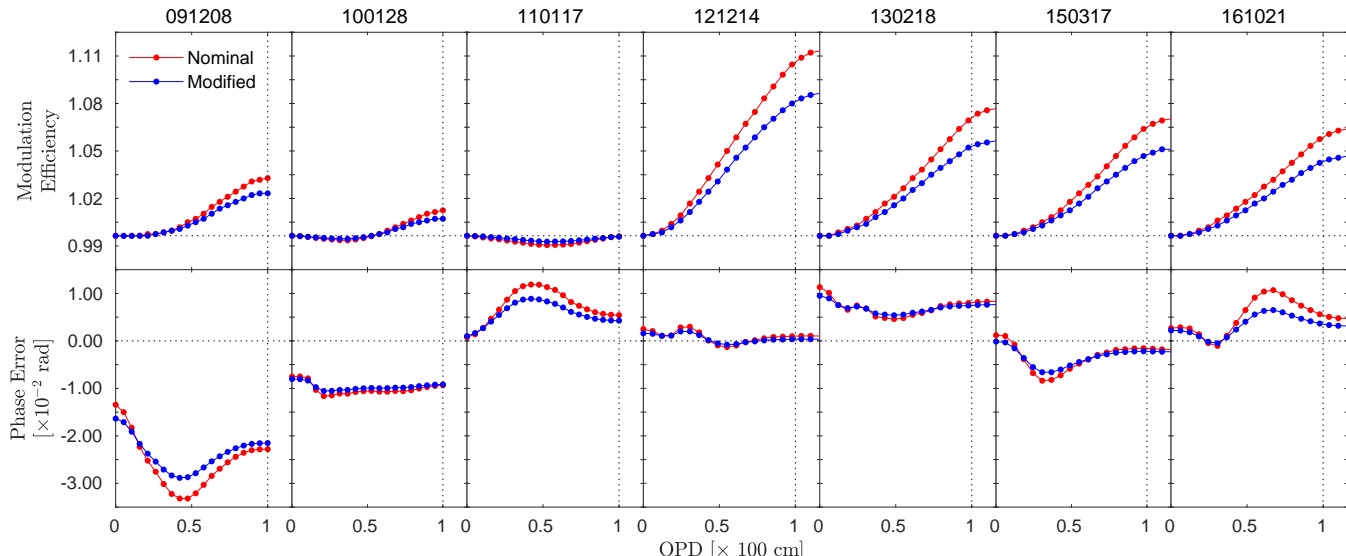

**Figure 18.** Temporal comparison of modulation efficiency and phase error for nominal and modified configurations across multiple dates.

modified instrument configurations to a $C_2H_6$ spectral microwindow spanning 2976.660–2976.950 cm$^{-1}$, based on observations acquired on December 14$^{th}$, 2012. In the upper panel, the observed solar spectrum is compared with forward simulations using the nominal and modified configurations, scaled for visual clarity, while the middle and lower panels show the corresponding residuals with respect to the observation. The residual difference in the nominal case exhibits larger amplitude, wider range, and increased dispersion, indicating that uncorrected non-ideal instrumental effects translate into systematic spectral mismatches. In contrast, the modified configuration produces residuals that are reduced in magnitude and variability, reflecting a more faithful representation of the ILS and associated effects in the forward model. This improvement in spectral consistency is quantitatively reflected in Table 5, where the modified configuration shows a substantial reduction in the RMS of the residuals, alongside clear decreases in systematic, statistical, and total errors. Notably, the retrieved total column increases by approximately 6–7 % relative to the nominal case, demonstrating that neglecting non-ideal instrumental characteristics leads to an underestimation of atmospheric abundance. These results establish a direct and robust link between instrument characterization and retrieval accuracy: accounting for the true instrumental response not only improves spectral fits but also yields more reliable and trustworthy atmospheric retrievals.

## 5   Discussion

Compared to Hase (2012), which primarily addressed improved ILS monitoring and sensitivity using optimized calibration cells, this study focuses on the practical consequences of non-ideal ILS behavior in long-term FTIR operations. By combining LINEFIT-based ILS retrievals with independent diagnostics of baseline distortions and spectral channeling, we examine how multiple instrumental non-idealities jointly influence spectral residuals and performance metrics over time. The study provided

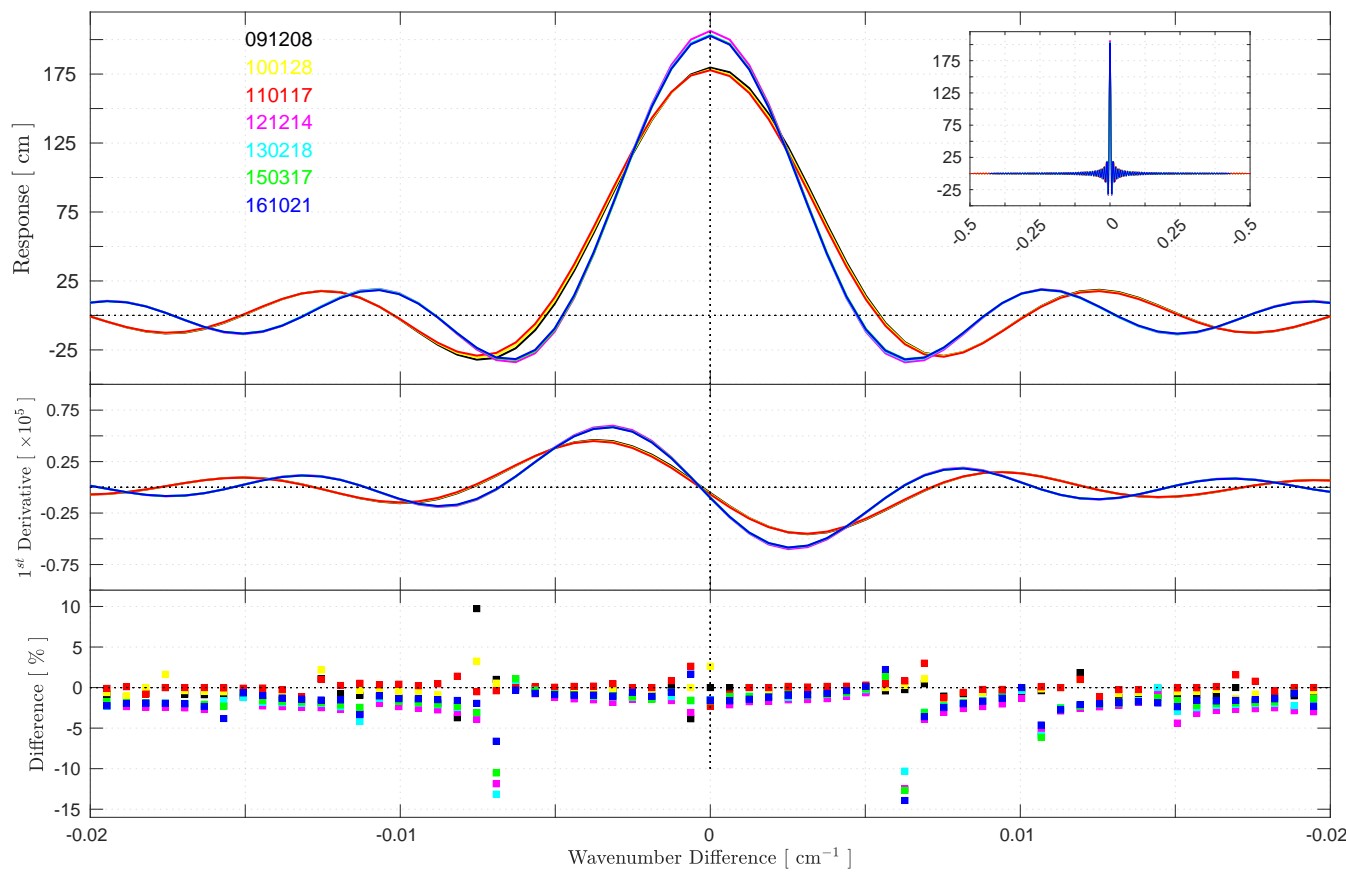

**Figure 19.** Evolution of ILS characteristics, with response (top), first derivative (middle), and percentage differences in the first derivative (bottom).

| Configuration | RMS of residaul ($\times 10^{-3}$) | | | Retrieval Output | | | |
|---|---|---|---|---|---|---|---|
| | MW1 | MW2 | MW3 | Systematic Error (%) | Statistical Error (%) | Total Error (%) | Total column ($\times 10^{19}$ molec m$^{-2}$) |
| **Nominal** | 0.881 | 1.48 | 0.823 | 2.70 | 1.02 | 2.88 | 6.88 |
| **Modified** | 0.463 | 1.37 | 0.672 | 2.38 | 0.39 | 2.41 | 7.35 |
| **$\Delta$ (%)** | $-47.5$ | $-7.21$ | $-18.3$ | $-11.8$ | $-61.8$ | $-16.3$ | $+6.78$ |

**Table 5.** Retrieval results for $C_2H_6$ from solar absorption spectra measured on 14 December 2012, comparing nominal and modified instrument configurations. The table reports the spectral fit RMS for each selected microwindow: MW1 (2976.64-2976.96 cm$^{-1}$), MW2 (2983.18-2983.54 cm$^{-1}$), and MW3 (2986.51-2986.95 cm$^{-1}$), together with the systematic, statistical, and total retrieval uncertainties, and the retrieved total column. Relative differences between the two configurations are expressed as $\Delta$ (%).

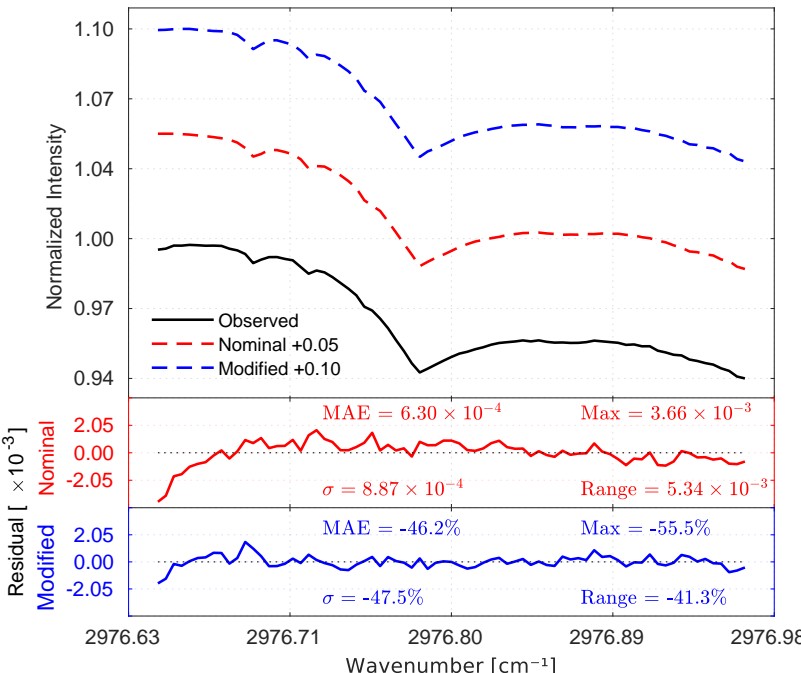

**Figure 20.** Illustration of normalized-intensity spectral fitting and residual analysis for $C_2H_6$ in the 2976.64-2976.96 cm$^{-1}$ microwindow, using data acquired on December 14[th], 2012. Observed and simulated spectra are shown for nominal and modified instrument configurations, with residuals plotted using identical scaling ($\times 10^{-3}$). Residual metrics are reported as absolute values for the nominal configuration and as relative changes (%) for the modified configuration.

a comprehensive investigation into the impact of non-ideal instrumental effects on the performance of high-resolution FTIR spectrometers, with a particular focus on the Bruker 120M. Using diagnostic tools such as ALIGN60 and LINEFIT, artifacts

arising from retroreflector misalignments, baseline drift, and spectral channeling were systematically identified and corrected-issues often overlooked in conventional configurations. These non-idealities were particularly pronounced at the ZPD, where alignment errors caused severe spectral smearing and sensitivity loss. In the nominal configuration, ME deviated by up to +10.9%, while PE variability reached $2.110 \times 10^{-2}$ radians, resulting in broader and asymmetric ILS. Such deviations are consistent with earlier findings by Hase et al. (1999), who highlighted the impact of lateral shear on ME and spectral accuracy.

The modified configuration in this study, designed to address these challenges, achieved substantial improvements, bringing ME within the NDACC-acceptable limit of 1.1 (Sun et al., 2018b) and reducing PE variability to $0.042 \times 10^{-2}$ radians. Improvements in ILS characteristics included sharper peaks, narrower FWHM, and enhanced symmetry, particularly in sidelobe regions. These enhancements align with similar corrections reported by Frey et al. (2019), who demonstrated the importance of precise alignment in reducing ILS distortions and achieving higher spectral fidelity.

The analysis of HBr transmission spectra validated the robustness of the modified configuration. Residuals and MAE were significantly reduced, with S/N varying slightly. Persistent spectral channeling at 2.9044 cm$^{-1}$, an inherent characteristic of

the instrument's design, was effectively corrected. Emerging channeling frequencies, such as 0.24 cm$^{-1}$, were attributed to retroreflector wear and potential degradation of the CaF$_2$ beamsplitter, corroborating findings by Blumenstock et al. (2021), who observed similar artifacts in aging FTIR systems. A transient anomaly at 40.672 cm$^{-1}$, linked to environmental factors such as vibrations or temporary misalignments, was also mitigated. These findings are consistent with Saggin et al. (2007), who emphasized the influence of mechanical instabilities and environmental conditions on spectrometer performance. The ALIGN60 simulations further corroborated these results, revealing Haidinger fringes and alignment degradation over time. Such detailed simulations have been previously utilized by Sun et al. (2018b) to understand alignment dynamics and their effects on spectral quality.

Temporal analysis revealed that alignment errors and component wear disproportionately affected the ZPD region, leading to increased spectral smearing and reduced sensitivity. These results echo the findings of De Mazière et al. (2018), who stressed the need for regular maintenance and recalibration in long-term FTIR operations to prevent performance losses. The instrument upgrade implemented during the study period increased the ILS peak height by >14% in both configurations, while the modified configuration reduced sidelobe asymmetry by >1% and achieved consistent reductions in MAE. These improvements demonstrate the modified configuration's effectiveness in compensating alignment errors, addressing aging-related degradation, and ensuring compatibility with NDACC standards.

The advancements achieved in this study extend beyond the technical improvements of the Bruker FTS 120M spectrometer. By demonstrating the critical importance of addressing non-ideal instrumental effects, this research contributes to the broader field of high-resolution FTIR spectroscopy, particularly for trace gas retrievals and atmospheric monitoring. Previous studies, including Hase et al. (1999) and Frey et al. (2019), have emphasized the need for enhanced alignment techniques and artifact corrections to meet NDACC standards. This study builds on their findings, showcasing the practical application of ALIGN60 and LINEFIT tools to systematically identify and correct instrumental artifacts. The modifications also set a benchmark for operational protocols, advocating for the integration of real-world instrumental deviations into data analysis frameworks. This approach ensures more reliable and reproducible results, advancing the objectives of NDACC and similar global networks.

The case study presented here demonstrates that empirically characterized non-ideal instrumental effects propagate directly into trace-gas retrieval products in a quantitatively significant manner. For the selected C$_2$H$_6$ retrieval on 14 December 2012, inclusion of the modified instrument configuration increases the retrieved total column by approximately 6–7 % relative to the nominal configuration. Previous sensitivity studies have shown that prescribed degradations of the ILS, when applied in isolation, can already induce measurable biases in retrieved C$_2$H$_6$ total columns, with a 10 % ILS degradation leading to changes on the order of 1 %, and that maintaining total-column differences within ±1 % requires limiting ILS degradations to below approximately 9 % (Sun et al., 2018b). In the present analysis, the substantially larger column difference arises from the combined and simultaneous influence of multiple empirically diagnosed non-idealities—including ILS distortion, spectral channeling, and baseline effects—under real operational conditions, rather than from a single prescribed ILS perturbation. The concurrent reductions in the RMS of the residuals and in systematic, statistical, and total retrieval errors further confirm that explicit propagation of the true instrumental response into the forward model is essential for achieving spectrally consistent fits and reliable atmospheric abundance estimates.

Despite these advancements, certain unresolved challenges remain. The persistent spectral channeling frequency at 2.9044 cm$^{-1}$ and the emerging 0.24 cm$^{-1}$ highlight the need for further investigations into mechanical and optical design improvements. Future research should explore automated diagnostic tools, as suggested by Sun et al. (2018b), to streamline the identification and correction of non-idealities. Additionally, extending the analysis to other spectroscopic systems could provide broader insights into the prevalence of such artifacts across various configurations. By addressing these challenges, the field can move toward establishing standardized methodologies for diagnosing and mitigating instrumental artifacts, ultimately enhancing the accuracy and reliability of high-resolution FTIR spectroscopy.

## 6 Conclusions

In this study, we conducted a comprehensive assessment of non-ideal instrumental effects in high-resolution FTIR spectroscopy, focusing on instrument performance characterization over an extended period. Our analysis revealed significant performance degradation over time, attributed to factors such as retroreflector misalignment, source brightness fluctuations, and the aging of optical components. The observed decline in interferogram intensity, particularly near the ZPD, highlights the critical impact of these factors on the instrument's modulation efficiency and phase accuracy. Through the comparison of nominal and modified configurations, we demonstrated that the modified configuration consistently achieved higher correlation coefficients ($R^2 > 0.99$) and exhibited enhanced residual behavior across all dates. Notably, the modified configuration showed systematic improvements in residual range, standard deviation ($\sigma$), and MAE, with reductions as high as >5% on challenging dates such as 150317 and 161021. Even on earlier dates with mixed behavior, such as 100128 and 110117, the MAE showed marked improvement despite small fluctuations in residual range. These trends confirm the robustness of the modified configuration in capturing instrumental characteristics more faithfully, particularly in regions affected by baseline drift and channeling. This underscores the necessity of incorporating correction algorithms in spectroscopic simulations to achieve accurate and reliable spectroscopic data analysis, especially under non-ideal or evolving instrumental conditions.

The analysis of the P(2) line of HBr confirms the significant improvement in instrument performance achieved through the modified configuration. Under the nominal setup, the residual standard deviation ($\sigma$) ranged from $0.49 \times 10^{-3}$ to $0.666 \times 10^{-3}$, and the corresponding RMS errors were between $0.491 \times 10^{-3}$ and $0.67 \times 10^{-3}$, indicating pronounced systematic deviations primarily attributed to baseline distortions and channeling artifacts. Following the implementation of the modified configuration, these values were markedly reduced to $\sigma$ values between $0.482 \times 10^{-3}$ and $0.671 \times 10^{-3}$, and RMS values between $0.503 \times 10^{-3}$ and $0.671 \times 10^{-3}$, reflecting a consistent and quantifiable suppression of instrumental artifacts. Although the improvement in signal-to-noise ratio was marginal, its close agreement between the observed and simulated spectra reinforces the improved spectral fidelity and operational stability introduced by the modified configuration. These results confirm that the modified setup provides a more accurate, consistent, and robust spectral characterization, establishing it as a reliable configuration for high-resolution FTIR measurements and retrieval applications.

Our investigation into spectral channeling behavior over seven years (from 091208 to 161021) revealed both stable and variable channeling frequencies, influenced by the instrument's optical configuration and mechanical components. The consistent

detection of the 2.9750 cm$^{-1}$ frequency across multiple dates strongly suggests it is an inherent characteristic of the instrument, likely arising from fixed optical or mechanical elements such as the retroreflector geometry or field stop. In the initial year, this frequency was the sole artifact observed, indicating near-nominal alignment and minimal mechanical instabilities. However, the subsequent appearance of an additional frequency around 0.2426 cm$^{-1}$ points to gradual misalignment, possible wear in the retroreflector system, or degradation of the CaF$_2$ beamsplitter. A prominent anomaly on 121214, marked by a sudden shift

to 40.670 cm$^{-1}$, underscores the sensitivity of the instrument to abrupt external factors such as environmental disturbances or temporary mechanical failures. These findings, corroborated by improvements in modulation efficiency (ME) and phase error (PE) under the modified configuration - for instance, ME improved from +10.9% to +8.40%, and PE decreased from 0.081 × 10$^{-2}$ rad to 0.042 × 10$^{-2}$ rad on 121214 - reinforce the necessity of mitigating non-idealities. Moreover, the observed narrowing of the FWHM from 0.0064 cm$^{-1}$ in earlier years to 0.0055 cm$^{-1}$ in later years reflects the impact of extending the

maximum optical path difference, emphasizing the importance of instrumental upgrades in enhancing spectral resolution.

Beyond instrument diagnostics, the results of this study demonstrate that non-ideal instrumental effects have a direct and quantifiable impact on atmospheric trace-gas retrievals. Using a representative C$_2$H$_6$ case study, we show that neglecting empirically characterized instrumental deviations leads to systematic spectral mismatches and an underestimation of the retrieved total column by approximately 6–7 %. This finding establishes that non-ideal instrumental effects are not merely second-order

corrections but can constitute a significant source of retrieval bias if left unaccounted for, even under NDACC-compliant operating conditions. Consequently, routine reliance on nominal instrument assumptions is insufficient for high-accuracy FTIR retrievals, particularly in long-term monitoring applications. By explicitly linking detailed instrument characterization to retrieval outcomes within a unified analysis, this study demonstrates that accurate treatment of instrumental non-idealities is an essential prerequisite for reliable atmospheric FTIR measurements.

In conclusion, our findings highlight the critical importance of regular maintenance, realignment, and the implementation of modified configurations to mitigate non-ideal instrumental effects in high-resolution FTIR spectroscopy. These measures are essential to maintain the fidelity of spectral data, ensuring reliable and accurate measurements over time. The systematic approach outlined in this study provides a robust framework for ongoing instrument performance evaluation and optimization, contributing to the advancement of high-resolution spectroscopic techniques.

*Data availability.* The data underlying the results presented in this paper are not publicly available at this time but may be obtained from the corresponding author upon reasonable request.

*Author contributions.* The authors conceptualized the study, performed the simulations and data analysis, interpreted the results, and wrote the manuscript.

*Competing interests.* The authors declare that they have no competing interests.

*Disclaimer.* The findings and interpretations presented in this article are those of the authors and do not necessarily reflect the views of their affiliated institutions. While every effort has been made to ensure the accuracy and reliability of the data and methodologies used, the authors assume no responsibility for any errors or omissions. The diagnostic procedures and performance evaluations discussed herein are based on the specific configuration and operational history of the Bruker FTS 120M spectrometer in Addis Ababa and may not be universally applicable to other FTIR instruments or settings. Readers are encouraged to verify the applicability of these methods to their own systems
and conditions.

*Acknowledgements.* The ALIGN60, LINEFIT14.5, as well as the PROFFIT retrieval code used in this study were kindly provided by Dr. Frank Hase of the Karlsruhe Institute of Technology (KIT), Institute for Meteorology and Climate Research (IMK-ASF), during a research visit to the institute in 2019. During this visit, I also had the opportunity to learn from discussions and seminars held at IMK-ASF, which helped shape the analytical framework of this study. The aforementioned software tools may be requested through the IMK-ASF website:
http://www.imk-asf.kit.edu/english/897.php. This research was conducted as part of my PhD studies at Addis Ababa University, with prior affiliation to Debre Birhan University. The experimental data were collected using the Bruker IFS 120M high-resolution FTIR spectrometer located at Addis Ababa University. I gratefully acknowledge the Department of Physics, Addis Ababa University, for providing access to the instrument and laboratory facilities essential for this work.

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
