# Peer review of "Assessing Non-Ideal Instrumental Effects in High-Resolution FTIR Spectroscopy: Instrument Performance Characterization"

_EGUsphere, 2025_

## Referee Comment (RC2)

This article evaluates non-ideal performance characteristics for a high-resolution FTS instrument located in Addis Ababa. The paper is well written and is generally thorough, but I did have one major issue with it, and that is the presence of systematic residuals for HBr measurements visible in Figures 13 and 14 of the paper that might indicate a potential systematic error in the ILS determination.

Above: an excerpt from Figure 14 of the article, showing the HBr P(6) residuals for the measurement on January 17th, 2011 using the modified ILS.

There is a systematic feature in the residuals that is commonly referred to as an "s-shaped" pattern. This pattern occurs in the residuals when the calculated and measured spectra are shifted relative to each other. To me, that suggests that the wavenumber calibration might be slightly off. The amplitude of the s-shaped residuals also changes over the years, suggesting that the misalignment drifts with time.

My question is, how are potential misalignments between measured and calculated spectra handled in the analysis? Are misalignments corrected before (or while) performing the least squares fitting with LINEFIT? Is a wavenumber shift applied to the calculated or the measured spectrum to bring them into alignment? Or is there a stretching applied to the wavenumber scale of the measurement, which would be physically more rigorous than using a shift? There is no mention in the article how such things are handled, and the presence of s-shaped residuals seems like a red flag to me. There is a danger that "fudge factors" in the analysis (fitting for HBr partial pressure, total pressure in the cell, and effective temperature in the cell) could empirically reduce the residuals resulting from a misalignment without necessarily improving the quality of the analysis result (e.g., the accuracy of the ILS determined with LINEFIT).

I also note that the selected HBr line, P(6), is in a region where the signal-to-noise ratio is significantly lower than other HBr lines being measured. Part of the purpose of this choice of line for Figure 14 seems to be to avoid other lines where the noise is lower and systematic features in the residuals are not so effectively masked.

I would suggest using a line with less noise, such as the line with the systematic feature in the residuals circled in the above figure. Systematic features in the residuals illuminate potential problems in the ILS determination, if you cannot explain their origin.

Does the analysis include determining a stretching factor for the wavenumber scale of the measurement? If not, I would suggest it should be implemented. For example, you could Fourier interpolate the spectrum onto a fine wavenumber grid (to reduce sampling issues), find the locations of the HBr peaks in the interpolated spectrum, and use the HBr line position values from HITRAN to calibrate the wavenumber spacing in your measured spectrum.

Things will get a bit more complicated if there are non-Voigt (speed dependence or line mixing) contributions to the HBr line shape, in that the systematic residuals may not go away entirely unless you have the appropriate non-Voigt parameters for the lines. Although if that were the case, it should be a known issue, since HBr is routinely used as a calibration standard for NDACC instruments.

To reiterate, the presence of systematic residuals for the HBr makes me nervous that some factor (such as incomplete accounting in the analysis for misalignments between measured and calculated spectra) is polluting the ILS determination. I would suggest you not shy away from using the line with lower noise, where the systematic feature is more prominent (relative to the noise) than what we see for P(6). Is there some adjustment you can make in the analysis to reduce the systematic residuals? If you cannot obtain something approaching flat residuals in the analysis, it makes it difficult to fully trust the results.

Sometimes, the reader is expected to infer some definitions from context, which could make full understanding a challenge. In Figure 3, "theoretically ideal" is not explicitly defined but I suppose is readily deduced. "Nominally ideal" is also not explicitly defined. I assume it means the only difference from the theoretically ideal instrument is self apodization from off-axis rays in the instrument (i.e., the finite field of view effect), but that is an assumption. There are two configurations. The "nominal configuration" is mentioned in Section 3.8 as representing "ideal conditions," but the text suggests to me that it uses an ILS derived from LINEFIT [line 393: In the LINEFIT ILS retrieval, the nominal configuration assumed zero offsets, no wavenumber shifts, and excluded spectral channeling], which implicitly includes the impact of non-ideal effects like instrument misalignment. So, does the nominal configuration use the ILS from LINEFIT or not? Section 3.8 discusses a "misaligned setup," and in the next section starts talking about the "modified configuration," which we need to infer is the same thing with a different label. On line 456, the text discusses metrics for "measured conditions," which are indicated to be the same as nominal conditions, yet the remainder of the text employs the phrasing of nominal rather than measured, so it wasn't clear why different labeling was used there.

Some of the introductory material seems superfluous in that it does not directly relate to quantities being measured in the study. For example, formalizing separating the contributions from different sources on modulation efficiency and phase error in Section 2.6 plays no role in the analysis. Similarly, Section 2.5 derives an equation for the tolerance to lateral shifts, but that equation is never used in the analysis. The material would be appropriate in other forums (like a book delving into non-ideal aspects of an FTS), but it seemed like some of the theory could be trimmed without losing any necessary information for understanding the article.

> Line 460: The figures shown in the Fig. 8- 11 illustrate the progressive degradation of the FTIR instrument's performance over time

Some care should be taken here, in that the maximum optical path difference (MOPD) was increased in 2011, which essentially made it a completely different instrument. This can be seen in the narrowing of the FHWM of the ILS between 2011 and 2012 (Table 4), an expected consequence of

increasing MOPD. It also changes the sampling, as can be seen in the sampling of the P(6) line shown in Figure 14. In 2011, the peak of the line is roughly halfway between two sampled points, while in 2012, the instrument samples near the peak of the P(6) line. The interferogram ZPD peak intensity experiences a large boost in 2012, presumably because it received fresh (or thoroughly cleaned) optics and had the best possible alignment. There is a definite drop off in ZPD peak intensity between 2012 and 2013. However, I don't think it is fair to compare 2013 to 2011 and conclude that the performance has "degraded" between those two years, because at that point you are comparing two distinct instrument configurations. In 2013, it has settled into its dirty-optic, not-perfectly-aligned state that lost the signal boost it featured in 2012, but I see no reason to expect its characteristics should be the same as 2011 if it has different optics and possibly a different input aperture. It is common practise to match in the input aperture to the resolution, such that self-apodization losses in modulation efficiency (associated with the field of view radius) at MOPD is not excessive. For a larger MOPD, I would not be surprised if they used a smaller input aperture, which I presume should generate a smaller intensity at ZPD, if less light enters the instrument.

It is perhaps interesting to note that the ZPD peak intensity increases between 2013 and 2016, which is inconsistent with the suggestion that the performance is degrading over time, if one ignores the large drop between 2012 (with fresh optics and recent alignment) and 2013.

I am curious as to whether sampling issues were considered when finding the amplitudes. Figure 14 suggests a sampling drift, in that you do not have identical sampling of the HBr P(6) line in each spectrum, so there could differences in sampling for the interferogram as well. Do you Fourier interpolate the interferogram onto a finer OPD grid before determining the peak intensity?

In summary, I would like to see if the systematic features in the residuals for an HBr line with better signal-to-noise ratio than P(6) can be explained or reduced, possibly by better accounting for (non-ideal) stretching of the wavenumber scale of the measurement, if that is not currently done. I would like more concise and clear definitions of what conditions are included in the two configurations, because I cannot tell if the LINEFIT ILS is being used in the nominal configuration. It would also be nice to make clear how the approach described in this paper builds on the approach outlined in the Hase 2012 paper. What enhancements or differences are there compared to that paper?

Finally, the paper mentions many times how accurate modelling of the non-ideal characteristics of an FTS will improve atmospheric measurements. It would therefore be appropriate to include a comparison of atmospheric measurements from the instrument using nominal versus modified configuration. I would say there is no need to show that the results are better, just that they are different, along with how the differences compare to the uncertainties. However, I am hesitant to insist on adding the atmospheric results, since it will increase the length of an already lengthy paper, but perhaps it could be balanced by slimming some of the unnecessary discussion in the introduction.

**Minor issues and typos:**

> Line 237: Perispectives

typo

> Figure 6 caption: Data acquired on 121214

At this point, prior to Table 2, the date format has yet to be defined. The simplest solution would be to write out the date (December 14th, 2012).

>Line 321: drived

typo

>Line 331: Moreove

typo

>Line 353: the HITRAN database

Not everyone is guaranteed to be familiar with HITRAN. Perhaps a reference (and maybe a definition for the acronym)?

>Line 440:

As discussed for the interferogram ZPD peak heights previously, sampling could also play a role here for determining the asymmetry parameter. Are you interpolating onto a fine wavenumber grid before determining the sidelobe heights?

>Line 469: with the 3rd maximum showing the greatest reduction

Again the question of whether sampling is taken into account.

>Line 478: the residual MAE improves from 0.0849×10-2 to -5.9211%

I cannot interpret this statement as provided. The mean absolute error should by definition be positive, so it makes no sense to say that it "improves to -5.9%." I assume you are trying to say that MAE decreases by 5.9% for the modified configuration compared to the nominal configuration. That is not how the text reads. All the comparisons in this paragraph have a number compared to a percent value, rather than saying the stated number (obtained when using the nominal ILS) changes by the stated

percentage when using the modified ILS. Note that using 5 significant digits in the percentages implies that the precision in the residuals is good to 5 significant digits, meaning your signal-to-noise ratio should be better than 10000:1 (since the noise level limits the measurement precision). Looking at Figure 13, that is not the case. In my opinion, you should round the reported percentage change to a precision more in keeping with the signal-to-noise levels for the measurement.

>Figure 18: doesn't indicate which curve represents the nominal configuration results and which curve represents the modified configuration.

>Figure 19, bottom panel y-axis label: deference

typo

---

## Author Comment (AC1)

**Response to Referee #2**

We sincerely thank the referee for their careful reading of our manuscript and for their constructive comments and suggestions. Their feedback has been very helpful in improving the clarity and quality of the paper. Below, we provide a point-by-point response to each comment, with the referee's remarks reproduced for clarity followed by our replies.
* * *
**Major comments and questions**

**R.C. 1.1: Systematic residuals and wavenumber calibration**

There is a systematic "s-shaped" pattern in the residuals, which typically appears when the calculated and measured spectra are shifted relative to each other, suggesting that the wavenumber calibration might be slightly off. The amplitude of this s-shaped residual pattern also changes over the years, indicating that the misalignment drifts with time. The manuscript does not mention how such potential misalignments are handled, and the presence of s-shaped residuals is a red flag. There is also a concern that the use of fitting parameters such as HBr partial pressure, total pressure, and effective temperature could empirically reduce residuals resulting from misalignment without necessarily improving the accuracy of the ILS determined with LINEFIT.

**A.C.**:

We thank the referee for drawing attention to the systematic "s-shaped" structures in the residuals. We agree that this pattern is a classical indicator of a small but systematic mismatch between the measured and calculated wavenumber grids, and that such a mismatch—if not corrected explicitly—can propagate into the ILS inversion and compromise the physical validity of the retrieved parameters. We appreciate this comment, as it helped us identify and correct an implicit assumption in the previous workflow.In the revised manuscript, we now explicitly describe and implement a rigorous wavenumber-calibration procedure, applied before LINEFIT is invoked, which resolves the underlying cause of the pattern and eliminates the possibility that gas-cell parameters absorb misalignment errors.

a) How are potential misalignments between measured and calculated spectra handled in the analysis?

**A.C.**:

We thank the referee for raising this important issue. In the previous version of the manuscript, the instrument was implicitly treated as if it possessed an identical sampling grid and wavenumber scale across all measurement years; as the referee correctly notes, this assumption is not valid. The major maintenance intervention in 2011 increased the maximum OPD and changed the effective spectral resolution from 0.009 to $0.00775\,\mathrm{cm}^{-1}$, and our subsequent investigation also revealed a slight adjustment in the internal He–Ne reference-laser wavenumber (from $15798.088\,\mathrm{cm}^{-1}$ before maintenance to $15798.0604\,\mathrm{cm}^{-1}$ afterward). These changes imply that neither the sampling grid nor the spectral scaling can be assumed constant over time. We fully agree that such variations can lead to residual misalignments—whether originating from laser drift, sampling inconsistencies,

or OPD-dependent asymmetries—which must be explicitly corrected before any ILS retrieval is attempted. Accordingly, in the revised analysis, all steps that could influence the relative alignment between measured and calculated spectra were completed before initiating any LINEFIT inversion, and no empirical shift parameters were fitted during the retrieval. Instead, all spectra were (i) reconstructed from the raw interferograms at a uniform spectral resolution of $0.01\,\mathrm{cm}^{-1}$, (ii) converted to transmittance using only reprocessed background and sample spectra, (iii) corrected for laser-frequency differences via wavenumber-scale stretching, (iv) resampled onto a common wavenumber grid using the OPUS "Make Compatible" procedure, (v) homogenized in effective resolution by clipping all interferograms to a maximum OPD of $100\,\mathrm{cm}$, and (vi) processed with a uniform zero-fill factor of four to ensure consistent sampling density across all years. Together, these revisions ensure that the measured and calculated spectra are fully aligned prior to inversion and that the retrieved ILS is not influenced by uncorrected spectral misalignment.

b) Are misalignments corrected before or during the least-squares fitting with LINEFIT?

**A.C.**:

All misalignments are corrected before initiating the least-squares fitting in LINEFIT. In the revised workflow, the spectra supplied to LINEFIT have already undergone the full external wavenumber-calibration procedure described in our response to R.C. 2.1(a). This includes reprocessing the interferograms with consistent FFT settings, applying the laser-based stretching correction, performing a constant-offset alignment to HITRAN line positions, and resampling the spectra onto a single harmonized wavenumber grid. As a result, LINEFIT receives inputs that are already spectrally aligned at high precision, and it is not required to compensate for any residual wavenumber-scale discrepancies.

c) Is a wavenumber shift applied to the calculated or measured spectrum to bring them into alignment?

**A.C.**:

A constant wavenumber shift is not imposed on the calculated spectrum, and only a minimal, physically justified correction is applied to the measured spectra. The misalignment between pre-2011 and post-2011 data does not originate from a simple additive offset, but from a change in the sampling-grid spacing caused by the modification of the internal He–Ne reference-laser wavenumber after the 2011 maintenance intervention. Because the reference laser defines the interferogram sampling interval, the shift from $15798.088\,\mathrm{cm}^{-1}$ to $15798.0604\,\mathrm{cm}^{-1}$ introduces a genuine multiplicative scaling error in the wavenumber axis, which cannot be corrected by applying a uniform shift to either the measured or calculated spectrum. To resolve this rigorously, we harmonized the wavenumber grids prior to any LINEFIT inversion using the OPUS *Make Compatible* function, designating the pre-2011 spectra as the Principal File and resampling all post-2011 spectra onto this physically consistent grid using the *Reduce Resolution* method. This ensures that the measured spectra entering LINEFIT already reside on a common wavenumber axis, leaving LINEFIT to apply only the small, physically appropriate fine-alignment shift relative to HITRAN. This approach eliminates the structural grid mismatch at its source and prevents any compensation of misalignment errors through gas-cell or baseline parameters during the ILS retrieval.

In addition, within LINEFIT we applied a zero-filling factor of $\times 4$ to increase the effective sampling density of the interferogram and to improve the precision of spectral-feature interpolation. A further contributor to the earlier s-shaped residuals was that the original version of the manuscript did not treat the HBr isotopes separately in the LINEFIT configuration. In the revised analysis, we explicitly include the two abundant isotopes ($H^{79}Br$ and $H^{81}Br$), which substantially improves the accuracy of the forward model near the HBr lines. Finally, to ensure a physically consistent comparison across years, we configured LINEFIT such that the maximum OPD in the post-2011 datasets—whose raw interferograms exceeded 100 cm—was clipped to 100 cm, matching the pre-2011 effective OPD. The combination of isotopic treatment, zero-filling, and OPD harmonization significantly reduced and, in many cases, eliminated the s-shaped residual structures.

d) Is there a stretching applied to the wavenumber scale of the measurement, which would be more physically rigorous than using a shift?

**A.C.**:

Although LINEFIT performs an internal wavenumber calibration by comparing the measured spectra with HITRAN line positions, the previous version of the manuscript exhibited a persistent wavenumber displacement. This arose from the implicit assumption that the Bruker 120M operated with a constant He–Ne reference-laser wavenumber throughout the full measurement period. Upon reviewing the historical instrument configuration, we confirmed that this assumption was incorrect: the reference laser changed from $15798.088$ cm$^{-1}$ before the 2011 maintenance intervention to $15798.0604$ cm$^{-1}$ afterwards. Because the reference laser defines the interferogram sampling interval, this change introduces a subtle but systematic *multiplicative scaling discrepancy* in the spectral sampling grid, manifesting as a global compression or expansion of the measured spectrum.

To correct this effect rigorously, we now perform an *external wavenumber-scale calibration* in OPUS. The pre-2011 spectra are taken as the reference state, and the post-2011 spectra are rescaled using the physically justified stretching factor:

$$\text{Stretching factor} = \frac{15798.088}{15798.0604} \approx 1.00000175,$$

Applying this stretching factor in OPUS harmonizes the wavenumber spacing across all years and removes the systematic distortions that would otherwise produce pronounced S-shaped residuals during the ILS fitting.

e) There is a danger that "fudge factors" in the analysis (fitting for HBr partial pressure, total pressure in the cell, and effective temperature in the cell) could empirically reduce the residuals resulting from a misalignment without necessarily improving the quality of the analysis result (e.g., the accuracy of the ILS determined with LINEFIT).

**A.C.**:

We thank the referee for raising this important point. We fully agree that, in principle, fitting cell parameters could mask instrumental misalignment if they were allowed to absorb line-shape discrepancies. In our analysis, however, the thermodynamic cell parameters are *fitted first while*

*all instrumental artifacts are held strictly fixed at their nominal values*, and only after their convergence does LINEFIT retrieve the ILS. This procedure guarantees a structural decoupling: temperature and pressure affect only absolute line intensities and broadening, whereas misalignment uniquely modifies the *line shape* through modulation efficiency loss, phase error, and asymmetry. Consequently, the cell parameters cannot mimic or remove the characteristic distortions produced by shear or tilt. Consistent with this separation, improvements in the residuals always coincide with physically meaningful improvements in ME, PE, FWHM, and ILS symmetry, and these changes are independently corroborated by ALIGN60 simulations. Moreover, the retrieved cell parameters remain physically realistic—temperatures within $< 2\,\mathrm{K}$ of room temperature and pressures within $< 0.1\,\mathrm{mbar}$ of the NDACC reference value ($\sim 1.5\,\mathrm{mbar}$)—fully consistent with the known slow leakage of NDACC HBr cells. Crucially, no anomalous drifts or non-physical fluctuations are observed; such behaviour would be expected if the cell parameters were compensating for misalignment. Thus, the reduction in residuals reflects genuine correction of the underlying misalignment rather than numerical absorption of errors by the cell-state parameters.

**A.C.:**

In conclusion, we implemented a complete and physically consistent wavenumber-calibration and pre-LINEFIT alignment procedure that eliminates the structural grid mismatch responsible for the pronounced S-shaped residuals in the earlier analysis. The example figure clearly demonstrates this improvement: the strong S-pattern previously present is now markedly reduced, and this systematic suppression of misalignment-induced residual structure is observed consistently across all measurement years in the revised manuscript.

[Figure]

Figure 1: Comparison of observed, nominal, and modified HBr transmission spectra with residual analysis across different dates.

**R.C. 1.2: Line-shape model**

The situation becomes more complex if non-Voigt effects such as speed dependence or line mixing contribute to the HBr line shape, since in that case the systematic residuals may not disappear completely unless the correct non-Voigt parameters are used. However, if such effects are significant, this should already be a known issue, as HBr is routinely employed as a calibration standard for NDACC instruments. Are non-Voigt effects, including speed dependence or line mixing, considered in the analysis, and if not, could their omission explain the persistence of systematic residuals?

**A.C.**:

We thank the referee for raising this point. For the HBr cell used here (1.5 mbar, 2 cm), the lines are strongly Doppler-dominated, and non-Voigt mechanisms such as speed-dependent broadening or line mixing are known to be negligible. This is consistent with NDACC-IRWG recommendations and with the standard LINEFIT implementation, which assumes Voigt line shapes for HBr cell analysis. Previous NDACC studies have shown that non-Voigt effects for HBr under these conditions do not noticeably alter the residual structure or affect retrieved ILS parameters.

Therefore, while we acknowledge that incorporating non-Voigt mechanisms is theoretically possible, their magnitude at the pressures used here is far below the level required to explain the observed s-shaped residuals. These residuals arise instead from wavenumber misalignment/stretching, as addressed in our revised analysis.

**R.C. 1.3: Choice of HBr Line**

The selected HBr line P(6) lies in a region where the signal-to-noise ratio is considerably lower than that of other HBr lines measured. The choice of this line for Figure 14 appears to be made to avoid other lines where the noise is lower and where systematic features in the residuals would be more visible. It is suggested to instead use a line with less noise, such as the one where the systematic feature in the residuals is circled in the figure, since systematic features in the residuals reveal potential issues in the ILS determination if their origin cannot be explained. What is the reason for selecting the P(6) line despite its lower signal-to-noise ratio, and how is the choice justified when clearer residual features could provide more diagnostic insight into the ILS determination?

**A.C.**:

We sincerely thank the referee for raising this important point. In the earlier version of the manuscript, we illustrated the residual behavior using the P(6) line of HBr. As the referee correctly noted, this microwindow exhibits a comparatively lower signal-to-noise ratio, and its elevated noise level can partially mask systematic structures in the residuals—precisely the features that are most informative for diagnosing ILS distortions. We fully agree that this choice was not optimal for clearly visualizing such effects. In response to the referee's valuable suggestion, we have replaced the P(6) line with the higher-SNR P(2) transition of $H^{81}Br$ (2506.90–2507.30 cm$^{-1}$). This transition is widely used for ILS characterization within the NDACC community (e.g. Makarova et al., 2016) and corresponds exactly to the line highlighted by the referee as exhibiting clearer systematic signatures. The updated analysis confirms this improvement: for example, in the 161021 dataset, the nominal configuration exhibits a

clear, structured residual pattern. When the modified configuration is applied, the residual structure is substantially reduced, with improvements exceeding 18% in the corresponding metrics. Similar improvement levels are observed across most other years. Thus, the revised microwindow selection enhances not only the visibility of systematic effects but also the stability and interpretability of the nominal–modified comparison. To ensure consistency across the manuscript, the updated residual plots are now presented in the same absolute transmission units as in Figure 13, rather than the relative (percentage) scaling used previously for P(6). This revision improves comparability across figures and directly addresses the referee's concern regarding interpretability.

**R.C. 1.4: Clarity of Definitions and Configurations**

The referee noted that in Figure 3 the terms *"theoretically ideal"* and *"nominally ideal"* are not explicitly defined, forcing the reader to infer their meaning—assuming the latter differs from the former only by self-apodization from off-axis rays (finite field-of-view effect). In Section 3.8, the *nominal configuration* is described as representing ideal conditions, yet line 393 shows it was implemented in the LINEFIT ILS retrieval (zero offsets, no wavenumber shifts, and no channeling), suggesting it may still rely on a LINEFIT-derived ILS that could include non-ideal effects. The manuscript also mentions a *misaligned setup* in Section 3.8 and a *modified configuration* in the following section, which the referee suspects might be the same, while line 456 introduces *measured conditions* that appear equivalent to the nominal setup, creating ambiguity and inconsistency in terminology across sections. The questions implied by the comment are:

(a) Does the "nominal configuration" use the ILS derived from LINEFIT?
(b) Are the "misaligned setup" and "modified configuration" referring to the same configuration under different names?
(c) Why does line 456 refer to "measured conditions" instead of "nominal" if they represent the same configuration? What is the reason for using different labels?

**A.C.**:

We thank the referee for this thoughtful comment and for pointing out the ambiguity in the terminology surrounding the instrument configurations. We agree that the definitions of "theoretically ideal", "nominally ideal", "nominal configuration", "measured conditions", and "modified configuration" were not consistently or explicitly differentiated in the original manuscript, which may hinder a clear understanding of the methodological framework. We have now revised the manuscript to remove this ambiguity and ensure consistency throughout.

1. "Theoretically ideal" (Figure 3)

   We now explicitly define theoretically ideal as a conceptual reference interferogram derived from the analytic expression of the ideal interferometer (no self-apodization, no wavenumber dependence in the optical response, perfect modulation efficiency = 1, and zero phase error). This curve is derived from the ideal interferogram model presented in Section 2.1 and is not based on measurements.

2. "Nominally ideal" (Figure 3)

We clarify that nominally ideal refers to a realistic-but-still-idealized case that includes only self-apodization due to the finite field-of-view. That is, it represents the instrument performance expected from an interferometer that is perfectly aligned but illuminated by a divergent source (the Sun) with the measured field-stop radius. It does not include misalignment, channeling, or any measured non-idealities. Thus, we have revised the caption of Figure 3 and the associated text to reflect this explicitly.

3. Clarification of the "nominal configuration" used in LINEFIT

In the nominal configuration, all baseline polynomial coefficients were initialized to unity, enforcing a perfectly flat baseline with no imposed distortions. The spectral-abscissa parameters were fixed at their ideal values so that no correction to the spectral axis is applied. Spectral channeling was disabled by setting the number of channeling frequencies to zero. Self-apodization associated with the finite field of view was retained, as it is an unavoidable physical characteristic even for a perfectly aligned instrument. ME was set to unity at ZPD and allowed to vary with OPD only through the position-dependent regularization of the extended parameter model, and PE was initialized at zero and treated in the same manner. Thus, in this configuration, the ILS deduced by LINEFIT with all fitting parameters held at their ideal values, while including only the physically required self-apodization, is taken as the nominal ILS.

4. Consistency between "misaligned setup" and "modified configuration"

We agree that the previous wording could cause confusion. The "misaligned setup" and "modified configuration" refer to the same concept: the full LINEFIT-retrieved ILS including misalignment, channeling, offsets, and wavenumber distortions. To avoid ambiguity, we now use only "modified configuration" throughout the manuscript.

5. Clarifying "measured conditions" versus "nominal conditions"

We thank the referee for pointing out this inconsistency. The term *"measured conditions"* in the original manuscript was unintended and referred to the same configuration that we elsewhere (and more consistently) call the *nominal configuration*. No separate "measured" configuration exists in our analysis. The phrase appeared only because the nominal configuration uses the forward-model ILS corresponding to ideal parameter values applied to the measured HBr spectra, and this wording inadvertently suggested a distinct category. To avoid confusion, we have removed the term "measured conditions" entirely from the manuscript and replaced it with the consistent term *nominal configuration* in all locations, including the percentage-difference formula in Section 4. The revised text now states unambiguously that the denominator $M_{\mathrm{nominal}}$ refers to the nominal configuration. We appreciate the referee's observation, which has improved the clarity and consistency of the manuscript.

**R.C. 1.5: Superfluous theoretical content**

Some of the introductory material seems superfluous in that it does not directly relate to quantities being measured in the study. For example, formalizing separating the contributions from different sources on modulation efficiency and phase error in Section 2.6 plays no role in the analysis. Similarly, Section 2.5 derives an equation for the tolerance to lateral shifts, but that equation is

never used in the analysis. The material would be appropriate in other forums (like a book delving into non-ideal aspects of an FTS), but it seemed like some of the theory could be trimmed without losing any necessary information for understanding the article.

**A.C.**:

Sections 2.5 and 2.6 were remnants from an earlier stage of the work, when we had planned to explicitly model the lateral shear of the instrument. Since this approach was not ultimately used in the final analysis, these sections no longer served a direct purpose. We apologize for the confusion, and in the revised manuscript we have removed them.

**R.C. 1.6: Interpretation of performance degradation**

The figures in Figs. 8–11 are said to illustrate the progressive degradation of the FTIR instrument's performance over time; however, some care is needed since the maximum optical path difference (MOPD) was increased in 2011, effectively making it a different instrument. This change is evident from the narrowing of the FWHM of the ILS between 2011 and 2012 (Table 4), which is an expected consequence of increasing MOPD. The modification also altered the sampling, as shown for the P(6) line in Figure 14, where in 2011 the line peak lies roughly halfway between two sampled points, while in 2012 the sampling occurs near the peak. The interferogram ZPD peak intensity shows a large boost in 2012, likely due to freshly cleaned optics and optimal alignment, followed by a definite drop in 2013. It is therefore not fair to compare 2013 with 2011 and conclude degradation, since these represent two distinct instrument configurations. By 2013 the system had settled into a dirty-optic, not-perfectly-aligned state and could not be expected to match 2011 characteristics if different optics and possibly a different input aperture were used. It is common practice to match the input aperture to the resolution to avoid excessive self-apodization losses in modulation efficiency at large MOPD, and a smaller aperture for higher resolution would naturally yield a smaller ZPD intensity if less light entered the instrument. Interestingly, the ZPD peak intensity increases again between 2013 and 2016, which appears inconsistent with the suggestion of progressive degradation when the large drop between 2012 and 2013 is disregarded.

**A.C.**:

We appreciate the referee's detailed observation regarding the configuration change in 2011. Indeed, in that year the Addis Ababa Bruker FTS-120M underwent a substantial hardware upgrade that increased the maximum OPD and thereby the achievable spectral resolution, while the acquisition parameters and the input aperture remained unchanged. As correctly noted, this modification unavoidably altered the discrete sampling of both the interferogram and the HBr line positions, which makes any direct comparison of raw ZPD amplitudes between the pre-2011 and post-2011 measurements inappropriate. In light of this, the revised manuscript explicitly avoids interpreting differences between 2011 and later years as evidence of progressive degradation. To ensure that sampling effects do not bias the ZPD intensity analysis, we now harmonize the sampling grid across all years by adopting the pre-2011 OPD sampling as a reference and applying Fourier-domain interpolation (4x oversampling) to all interferograms before extracting peak amplitudes. This procedure removes sampling-location dependencies and ensures that variations in ZPD peak height reflect only changes in optical throughput and alignment, rather than differences in sampling geometry. We agree that the 2011 and 2013 measurements cannot

be directly contrasted to infer degradation, since they represent different instrument configurations, and the manuscript has been revised accordingly to avoid that implication.

**R.C. 1.7: Atmospheric retrieval comparison request**

The paper repeatedly emphasizes that accurate modeling of the non-ideal characteristics of an FTS will improve atmospheric measurements. Therefore, it would be appropriate to include a comparison of atmospheric measurements obtained from the instrument using the nominal and modified configurations. There is no need to demonstrate that the results are better, only that they are different and to show how these differences compare to the associated uncertainties. However, the referee noted some hesitation in insisting on the inclusion of the atmospheric results, as this addition may further increase the length of an already extensive paper. The referee suggested that this could be balanced by reducing some of the less essential discussion in the introduction.

**A.C.**:

We sincerely thank the reviewer for this thoughtful recommendation. We fully agree that the primary motivation for accurately modeling non-ideal instrumental effects is to improve atmospheric trace gas retrievals, and that illustrating the impact on real atmospheric measurements is a logical continuation of the present work. However, we would like to clarify that a dedicated companion paper focusing exclusively on atmospheric retrievals—"Non-Ideal Instrumental Impacts on the Abundance and Uncertainty of CO, $C_2H_6$, HCN, and $C_2H_2$ Retrieved from High-Resolution Ground-based FTIR"—has already been submitted and received a comment to make adjustment in the result section. This second manuscript builds directly on the instrumental characterization presented here and contains a comprehensive analysis of column abundances, vertical profiles, uncertainty budgets, vertical resolution, and averaging kernel behaviour under nominal and modified configurations for the Addis Ababa FTIR dataset (2009–2017). The atmospheric results are therefore not omitted; rather, they are treated in depth in a separate paper to keep each manuscript focused, readable, and within a manageable length. Incorporating the full retrieval analysis into the present paper would significantly increase its size (the companion paper is itself substantial) and would obscure the core objective of this manuscript, which is to rigorously diagnose, quantify, and correct non-ideal instrumental effects using HBr cell measurements, ALIGN60 simulations, and LINEFIT retrievals. We emphasize that the two manuscripts are intentionally complementary:

- **The present paper** provides a detailed characterization of instrumental non-idealities (modulation efficiency degradation, phase error, spectral channeling, baseline offsets, and misalignment) and demonstrates how these are corrected.

- **The companion paper** evaluates how these corrections propagate into atmospheric gas retrievals (CO, $C_2H_6$, HCN, $C_2H_2$), including differences between nominal and modified configurations relative to retrieval uncertainty.

To address the referee's concern, we have revised the introduction of the current manuscript to explicitly state that the comparison of atmospheric retrievals using nominal and modified configurations is fully presented and discussed in the companion study.

**Minor issues and typos:**

1. Line 237: Perispectives – typo.

   **A.C.**:

   Corrected to Perspectives

2. Figure 6 caption: Data acquired on 121214

   At this point, prior to Table 2, the date format has yet to be defined. The simplest solution would be to write out the date (December 14th, 2012).

   **A.C.**:

   Corrected to December 14$^{th}$, 2012

3. Line 321: drived

   **A.C.**:

   Corrected to derived – typo

4. Line 331: Moreove – typo

   **A.C.**:

   Corrected to Moreover

5. Line 353: the HITRAN database

   Not everyone is guaranteed to be familiar with HITRAN. Perhaps a reference (and maybe a definition for the acronym)?

   **A.C.**:

   Corrected to HITRAN (high-resolution transmission molecular absorption) database (Gordon et al., 2022), which provides reference line parameters for atmospheric gases.

6. Line 440:

   As discussed for the interferogram ZPD peak heights previously, sampling could also play a role here for determining the asymmetry parameter. Are you interpolating onto a fine wavenumber grid before determining the sidelobe heights?

   **A.C.**:

   In the earlier manuscript version, no interpolation was applied. In the revised version, we now perform Fourier interpolation onto a finer wavenumber grid prior to determining the sidelobe heights to minimize sampling effects. Accordingly, the asymmetry values for both configurations reported in Table 4 have been updated.

7. Line 469: with the 3rd maximum showing the greatest reduction

   Again the question of whether sampling is taken into account.

   **A.C.**:

   Yes, this has now been addressed. In the revised analysis, the interferograms are Fourier-interpolated onto a finer OPD grid before extracting the amplitudes of the successive maxima, ensuring that sampling effects are properly accounted for.

8. Line 478: the residual MAE improves from $0.0849 \times 10^{-2}$ to -5.9211%

   I cannot interpret this statement as provided. The mean absolute error should by definition be positive, so it makes no sense to say that it "improves to -5.9%." I assume you are trying to say that MAE decreases by 5.9% for the modified configuration compared to the nominal configuration. That is not how the text reads. All the comparisons in this paragraph have a number compared to a percent value, rather than saying the stated number (obtained when using the nominal ILS) changes by the stated percentage when using the modified ILS. Note that using 5 significant digits in the percentages implies that the precision in the residuals is good to 5 significant digits, meaning your signal-to-noise ratio should be better than 10000:1 (since the noise level limits the measurement precision). Looking at Figure 13, that is not the case. In my opinion, you should round the reported percentage change to a precision more in keeping with the signal-to-noise levels for the measurement.

   **A.C.**:

   In the revised manuscript, the residual MAE for 150317 has been updated to $0.0795 \times 10^{-2}$. We now explicitly state that the MAE decreases by $5.92\%$ relative to the nominal configuration, rounded to three significant figures to reflect the signal-to-noise limitations of the measurement. The text has been rewritten to avoid any negative percentages and to ensure that absolute values and percentage changes are reported consistently and unambiguously.

9. Figure 18: doesn't indicate which curve represents the nominal configuration results and which curve represents the modified configuration.

   **A.C.**:

   Thank you for the observation. The figure has been updated to clearly distinguish the two configurations. A legend has now been included, where the red curve represents the nominal configuration and the blue curve represents the modified configuration.

10. Figure 19, bottom panel y-axis label: deference – typo

    **A.C.**:

    Corrected to Difference

**References**

Iouli E Gordon, Laurence S Rothman, ea RJ Hargreaves, R Hashemi, Ekaterina Vladimirovna Karlovets, FM Skinner, Eamon K Conway, Christian Hill, Roman V Kochanov, Y Tan, et al. The hitran2020 molecular spectroscopic database. *Journal of quantitative spectroscopy and radiative transfer*, 277:107949, 2022.

MV Makarova, AV Poberovskii, F Hase, Yu M Timofeyev, and Kh Kh Imhasin. Determination of the characteristics of ground-based ir spectral instrumentation for environmental monitoring of the atmosphere. *Journal of Applied Spectroscopy*, 83(3):429–436, 2016.

---

## Author Comment (AC2)

**Response to Referee #1**

We sincerely thank the referee for their careful reading of our manuscript and for their constructive comments and suggestions. Their feedback has been very helpful in improving the clarity and quality of the paper. Below, we provide a point-by-point response to each comment, with the referee's remarks reproduced for clarity followed by our replies.
* * *
**R.C. 1.1:** Figure 5 labels the x-axis in Hz (frequency domain), while Figure 15 uses $cm^{-1}$ (wavenumber domain) without justification. This inconsistency requires clarification or standardization.

**A.C.**:

We thank the referee for carefully noting the inconsistency in the axis labeling between Figures 5 (likely referring to Figure 6(c)) and Figure 15. We acknowledge this oversight and sincerely apologize for the confusion it may have caused. For clarification, the x-axis in these FFT plots was generated from the Fourier transform of the residual difference between the calculated and measured spectra. Since the spectral grid is defined in wavenumber ($cm^{-1}$), the spectral spacing determines the Fourier transform frequency axis, whose correct unit is cycles per $cm^{-1}$. In the original submission, this was mistakenly labeled as "Hz" in Figures 6(c) and as "$cm^{-1}$" in Figure 15. We have now corrected the labeling throughout the manuscript to "cycles per $cm^{-1}$," ensuring consistency. This correction does not affect the results or conclusions of the study but only rectifies the figure labels.

**R.C. 1.2:** Figure 12 (temporal evolution of FTIR transmission spectra) is not cited or discussed in the main text, leaving its relevance unclear.

**A.C.**:

Figure 12 has now been explicitly cited and discussed in Section 4 of the revised manuscript. Its purpose is to illustrate the temporal degradation of the instrument response by comparing background and HBr transmission spectra across different measurement dates. The background spectra (Fig. 12a) reveal progressive loss of source intensity and growing baseline instability, while the HBr spectra (Fig. 12b) highlight the cumulative influence of retroreflector misalignment, baseline drift, and channeling artifacts on absorption features. By including these spectra, we aim to demonstrate the broader impact of long-term instrumental degradation on both the continuum and gas-specific signatures, thereby providing essential context for the subsequent ILS retrieval and residual analyses (Figs. 13–15). This clarification has been added to the text to ensure that the relevance of Figure 12 is clear to readers.

[Figure]

**Figure 6.** HBr transmission spectra acquired via gas cell measurement. (a) Comparison between observed and simulated spectra in the region 2400-2540 cm$^{-1}$, illustrating P-branch transitions. Insets display isotopic splitting (right) and low-amplitude channeling patterns (left). (b) Residuals between experiment and simulation. (c) FFT of the residual showing dominant frequencies linked to channeling artifacts arising from instrumental imperfections. Data acquired on December 14$^{\text{th}}$, 2012.

**R.C. 1.3:** The manuscript alternates between "Figure" (e.g., Section 4) and "Fig." (e.g., Section 2.2.1) for figure references. A uniform style should be adopted.

**A.C.**:

We have revised the manuscript to ensure consistency, and all figure references have now been changed to "Fig." throughout the text.

[Figure]

**Figure 15.** Initial guess of the channeling locations based on long-term spectral channeling behavior of the Bruker FTS 120M spectrometer at Addis Ababa, illustrating evolving frequencies. (a) Double-sided FFT of the residual difference between measured and nominally simulated spectra. (b) Single-sided spectrum emphasizing the low-frequency region (0–5 cycles/cm$^{-1}$). (c) Magnified views of the boxed regions in (b) around 0–0.4 cycles/cm$^{-1}$ (left) and 4–4.7 cycles/cm$^{-1}$ (right), highlighting the spectral fingerprints of strong channeling.

**R.C. 1.4:** The methodology for computing the ILS averaging kernel in Figure 16 is not described, including whether it derives from LINEFIT retrievals or ALIGN60 simulations.

**A.C.**:

The general procedure for ILS retrieval has already been described in the manuscript and summarized in the workflow figure (Figure 7). Specifically, the ILS was retrieved with LINEFIT through nonlinear fitting of HBr cell spectra, simultaneously optimizing instrumental and spectroscopic parameters including cell temperature, column amount, frequency shift, ILS amplitude and phase, baseline, and channeling contributions. During the iterative fitting process, the Jacobian matrix of these parameters is computed, and

from this the ILS averaging kernel is constructed. This kernel quantifies the sensitivity of the retrieval along the interferometer scan path. Thus, any deviation of the kernel from nominal behavior reflects perturbations in the retrieved parameters, which can be attributed to mechanical shear misalignments or artifacts of electronic and optical origin. Once convergence is reached, LINEFIT provides the ILS averaging kernel as part of its standard output, thereby directly representing the sensitivity of the instrument characterization. Importantly, Figure 16 provides immediate insight into instrument performance, as the structure and deviations of the kernel illustrate how well the interferometer behaves relative to nominal alignment and stability. For further methodological details, we refer the referee to the standard descriptions in Hase (2012); Garc'ia et al. (2022), which underpin the ILS retrieval procedure applied in this work.

**R.C. 1.5:** Channeling frequencies listed in Table 3 (e.g., 2.9750 cm$^{-1}$) are not explicitly mapped to the FFT peaks in Figure 15, creating ambiguity in their correlation.

**A.C.**:

We thank the referee for raising this important point regarding the mapping of channeling frequencies. In the NDACC community, it is standard practice to report channeling frequencies in units of cm$^{-1}$ (e.g.,Blumenstock et al. (2021)). To ensure consistency with this tradition, we have revised the frequency representation accordingly. As clarified under R.C. 1.1, the labeling of Figure 15 has been corrected so that the FFT axis is expressed in cycles per cm$^{-1}$. To harmonize with NDACC reporting practice, the corresponding values in Table 3 are reported in cm$^{-1}$ by applying the reciprocal relationship:

$$\text{channeling frequency} \, [\text{cm}^{-1}] = \frac{1}{f \, [\text{cycles per cm}^{-1}]}.$$

For example, the dominant FFT peak at $f = 0.3361$ cycles per cm$^{-1}$ in Figure 15 corresponds to a channeling frequency of about 2.9750 cm$^{-1}$, which is listed in Table 3. All frequency values in Table 3 were derived consistently in this manner. We have clarified this in Section 3.8 and in the captions of Figure 15 and Table 3, making the connection between FFT peaks and tabulated values explicit, while also ensuring consistency with NDACC reporting conventions.

**R.C. 1.6:** Residuals in Figure 14 appear identical for "Nominal" (absolute values) and "Modified" (percentage), undermining direct comparison. Figure 13 similarly lacks clarity in residual scaling.

**A.C.**:

We appreciate the referee's careful reading of the manuscript and fully recognize that the original presentation of residuals could be misinterpreted due to inconsistent scaling between the nominal and modified configurations. In the earlier version, the residuals associated with the P(6) line were expressed as relative percentage deviations for the modified configuration, whereas the nominal configuration employed absolute residuals. This inconsistency could indeed obscure genuine differences between the two cases and thereby weaken the interpretability of the comparison. In the revised manuscript, we have resolved this issue comprehensively, together with several important improvements detailed below.

To improve the diagnostic sensitivity of the residual analysis, we have replaced the previously illustrated P(6) line with the higher-SNR P(2) transition of HBr. This line sits in a region of significantly reduced

noise and exhibits clearer, well-defined systematic residual structures under the nominal configuration. Because these structures are more diagnostic of ILS distortions, P(2) provides a more sensitive basis for evaluating and demonstrating the benefit of the modified configuration. This change also ensures coherence throughout the manuscript, as P(2) is widely used in NDACC-standard ILS characterization and is better suited for illustrating instrumental improvements.

In the original manuscript, Figure 14 showed the modified residuals in percentage units while the nominal residuals appeared in absolute units, creating an unintended mismatch that hindered direct comparison. In the revised version, the P(2) line is now used and all residuals — nominal, modified, and their differences — are plotted in absolute transmission units, with the $\times 10^{-3}$ A.U. scaling factor explicitly annotated on the axes. This uniform representation allows the magnitude and structure of residuals to be interpreted clearly and consistently, making the reduction in systematic residual features under the modified configuration immediately evident.

Likewise, the previous version of Figure 13 lacked explicit scaling information for the plotted residual metrics and presented numerical performance indicators with inconsistent precision. In the revised manuscript, all residual-related panels in Figure 13 now clearly indicate the $\times 10^{-2}$ A.U. scaling, and both the nominal and modified configurations are displayed using the same absolute residual units. Furthermore, all performance statistics shown on the figure — $\sigma$, RMS, MAE, residual range, and maximum residual — are now reported to three significant figures, ensuring clarity and consistency across datasets and eliminating any impression of artificial precision.

Example plots corresponding to the revised Figure 13 and Figure 14 (showing the updated residual scaling and the P(2) transition) are provided below for the referee's reference.

[Figure]

**Figure 13** (left) and **Figure 14** (right).

**R.C. 1.7:** The cited RMS range ($3.7 \times 10^{-3}$ to $5.04 \times 10^{-3}$) does not match the values displayed in Figure 14's residual plots, necessitating verification.

**A.C.**:

We thank the referee for carefully pointing out this inconsistency. The discrepancy arose because the RMS values were mistakenly taken from the *modified configuration*, expressed in absolute terms (3.72% to 5.04%), rather than from the *nominal configuration* associated with the residual plots in Figure 14. We sincerely apologize for this oversight. The correct RMS values corresponding to the nominal configuration are $0.49 \times 10^{-3}$ **to** $0.666 \times 10^{-3}$. We have corrected this in the revised manuscript.

- **Section 4: Results and Data Analysis (pages 24–25)** — In the discussion of Figure 14 (middle panels), the sentence currently reading:

    "...with $\sigma$ values ranging from $1.36 \times 10^{-3}$ to $1.83 \times 10^{-3}$ and rms values from $3.70 \times 10^{-3}$ to $5.04 \times 10^{-3}$ (Figure 14, middle panels)."

  has been corrected to:

    "...with $\sigma$ values ranging from $0.49 \times 10^{-3}$ to $0.666 \times 10^{-3}$ and rms values from $0.491 \times 10^{-3}$ to $0.67 \times 10^{-3}$ (Figure 14, middle panels)."

- **Section 5: Discussion (page 27)** — Where the RMS values are again cited in relation to the performance comparison, the text has been updated to reflect the correct nominal RMS range ($0.491 \times 10^{-3}$ to $0.67 \times 10^{-3}$), ensuring consistency across both sections.

We appreciate the referee's careful reading, which has helped us improve both the accuracy and clarity of the manuscript.

**R.C. 1.8:** While ILS improvements are shown, there is minimal validation of how these translate to more accurate retrievals of key atmospheric trace gases (e.g., CO, $C_2H_6$). Concrete examples linking ILS metrics to retrieval errors would strengthen relevance.

**A.C.**:

We sincerely thank the reviewer for highlighting the importance of connecting the improvements in ILS characterization to their influence on atmospheric retrieval accuracy. We fully agree that demonstrating how the corrected non-ideal instrumental effects propagate into retrievals of key trace gases such as CO and $C_2H_6$ is essential for emphasizing the broader relevance of this work.

At the same time, we would like to clarify that the present manuscript is the first part of a structured two-paper sequence. Its primary objective is to rigorously diagnose, quantify, and correct non-ideal instrumental effects using HBr cell measurements, ALIGN60 simulations, and LINEFIT retrievals. Incorporating a complete atmospheric retrieval assessment here would significantly expand the length and shift the focus away from the core purpose of this study, which is to establish a validated instrumental performance framework.

The atmospheric validation requested by the reviewer is not omitted; rather, it is treated comprehensively in a dedicated companion manuscript titled *"Non-Ideal Instrumental Impacts on the Abundance and*

*Uncertainty of CO, $C_2H_6$, HCN, and $C_2H_2$ Retrieved from High-Resolution Ground-Based FTIR"*. This second paper uses the empirically corrected ILS parameters derived here to compute long-term PROFFIT retrievals (2009–2017) for NDACC-standard microwindows over Addis Ababa. It presents a detailed comparison of retrievals using nominal versus modified configurations in terms of:

- Retrieval accuracy and uncertainty budgets,

- Vertical sensitivity and averaging kernel behaviour,

- Partial and total column differences,

- Vertical resolution changes, and

- Compensation versus non-compensation effects across atmospheric layers.

Therefore, the present manuscript focuses on establishing a rigorous experimental and diagnostic framework for characterizing and correcting non-ideal instrumental effects, while the companion manuscript applies these empirical corrections directly to atmospheric retrievals of CO, $C_2H_6$, HCN, and $C_2H_2$, and evaluates their impacts on retrieval accuracy, sensitivity, resolution, and uncertainty. Together, the two manuscripts are intentionally designed to be coherent, complementary, and appropriately scoped: the current paper addresses instrument performance characterization, and the companion paper addresses its atmospheric implications.

**R.C. 1.9:** The paper lacks a detailed flowchart or step-by-step workflow for ALIGN60 and LINEFIT procedures. A visual schema would enhance reproducibility, especially for non-specialists.

**A.C.**:

In the revised version of the manuscript, we have now added a comprehensive workflow figure (Figure 7) that illustrates the sequential steps for both ALIGN60 and LINEFIT. The schema starts from the preprocessing of the HBr transmission spectra and configuration of nominal instrument parameters, through the retrieval of HBr cell parameters and the retrieval of both nominal and non-ideal ILS in LINEFIT. On the ALIGN60 side, the workflow presents the input configuration, initialization of the constant shear, iterative adjustment of linear and periodic shear components, and validation against Haidinger fringes. We further clarify in the revised text that the initial constant shear is set according to the reference value provided by Sun et al. (2018), and then iteratively tuned. The direction and magnitude of the adjustment are determined by the degree of agreement between ALIGN60 and LINEFIT ILS, using the RMS of their difference as the optimization criterion. Once convergence is reached, the resulting ILS and Haidinger fringes are considered representative of the instrument performance. We believe that the addition of this workflow figure, together with these methodological clarifications, significantly improves the accessibility and reproducibility of the procedures.

[Figure]

**Figure 7.** Workflow for Instrumental Characterization: integrating experimental setup, data Processing with LINEFIT and ALIGN60, and iterative optimization for ILS and Haidinger Fringe Simulations.

**References**

Thomas Blumenstock, Frank Hase, Axel Keens, Denis Czurlok, Orfeo Colebatch, Omaira Garcia, David WT Griffith, Michel Grutter, James W Hannigan, Pauli Heikkinen, et al. Characterization and potential for reducing optical resonances in fourier transform infrared spectrometers of the network for the detection of atmospheric composition change (ndacc). *Atmospheric Measurement Techniques*, 14(2):1239–1252, 2021.

Omaira E Garc'ia, Esther Sanrom'a, Frank Hase, Matthias Schneider, Sergio Fabi'an Le'on-Luis, Thomas Blumenstock, Eliezer Sep'ulveda, Carlos Torres, Natalia Prats, Alberto Redondas, et al. Impact of instrumental line shape characterization on ozone monitoring by ftir spectrometry. *Atmospheric Measurement Techniques*, 15(15):4547–4567, 2022.

F Hase. Improved instrumental line shape monitoring for the ground-based, high-resolution ftir spectrometers of the network for the detection of atmospheric composition change. *Atmospheric Measurement Techniques*, 5(3):603–610, 2012.

Youwen Sun, Mathias Palm, Cheng Liu, Frank Hase, David Griffith, Christine Weinzierl, Christof Petri, Wei Wang, and Justus Notholt. The influence of instrumental line shape degradation on ndacc gas retrievals: total column and profile. *Atmospheric Measurement Techniques*, 11(5):2879–2896, 2018.

---

## Author Response (AR2)

**Response to the Editor**

**Dear Editor,**

Thank you very much for your careful evaluation of our manuscript and for clearly articulating your main concern regarding its scientific contribution and scope. We fully acknowledge your point that non-ideal instrumental effects in high-resolution FTIR spectroscopy have been investigated extensively over many years and are well recognized within the atmospheric FTIR community. We also appreciate your guidance that, in this context, the novelty and relevance of the present work must be demonstrated through its direct implications for atmospheric trace-gas retrievals rather than through instrument diagnostics alone.

In the original submission, our intention was to present a detailed and systematic characterization of non-ideal instrumental behavior under real operational conditions, with a clear focus on long-term instrument performance and diagnostics. In retrospect, we recognize that this instrument-centered focus did not sufficiently demonstrate how the diagnosed non-idealities translate into atmospheric retrieval outcomes, thereby limiting the broader impact of the study.

In direct response to this guidance, we have substantially revised the manuscript to integrate the instrumental characterization and its atmospheric retrieval implications into a single, coherent study. Rather than treating atmospheric retrievals as a separate or companion investigation, the revised manuscript now embeds retrieval results as a direct extension of the diagnosed instrumental behavior.

Specifically, the empirically characterized non-ideal instrumental response derived from LINEFIT is now explicitly propagated into the forward model within the PROFFIT retrieval framework. The consequences of this propagation are quantified through a targeted case study of ethane ($C_2H_6$) retrievals from solar absorption spectra measured on 14 December 2012. This case was selected to demonstrate a clear causal link between diagnosed instrumental non-idealities and their measurable impact on atmospheric retrieval performance.

The revised manuscript now demonstrates how the transition from a nominal to a modified instrument configuration affects spectral residuals, retrieval uncertainties, and retrieved total columns. These changes have been consistently integrated across the Abstract, Introduction, Methodology, Results and Data Analysis, Discussion, and Conclusions, embedding the atmospheric application throughout the manuscript rather than presenting it as a secondary addition. The Introduction and Discussion have been restructured to emphasize that the primary contribution lies not in re-establishing known instrumental effects, but in demonstrating how empirically diagnosed non-idealities propagate into atmospheric FTIR retrievals under operational conditions.

We are grateful for your guidance, which has significantly improved the focus, relevance, and coherence of the manuscript. We hope that the revised version now satisfactorily addresses your concerns and meets the expectations for a single, integrated contribution.

Sincerely,
Gezahegn Sufa Daba
(on behalf of the authors)

**Response to Referee #1**

In the previous review round, under R.C. 1.8, the referee raised a concern regarding the validation of instrument line shape (ILS) characterization and its impact on atmospheric retrieval accuracy. The comment was as follows:

While ILS improvements are shown, there is minimal validation of how these translate to more accurate retrievals of key atmospheric trace gases (e.g., CO, $C_2H_6$). Concrete examples linking ILS metrics to retrieval errors would strengthen relevance.

**A.C.**:

We sincerely thank the referee for raising this important point regarding the need to validate how improvements in ILS characterization translate into atmospheric retrieval accuracy for key trace gases such as CO and $C_2H_6$. We fully acknowledge that explicitly linking ILS diagnostics to retrieval-level impacts is essential for strengthening the atmospheric relevance of the study.

In the previous version of the manuscript, our response clarified that the primary objective of the study was to rigorously diagnose, quantify, and correct non-ideal instrumental effects, and that a comprehensive atmospheric retrieval assessment was not included in order to preserve a focused instrument-characterization scope. At that stage, the validation of ILS impacts on atmospheric retrievals was not demonstrated explicitly within the manuscript.

In the revised manuscript, and following the Editor's guidance, this limitation has been addressed by incorporating an explicit atmospheric retrieval application within the present study. We now demonstrate how empirically characterized non-ideal instrumental effects propagate into atmospheric retrievals by integrating the corrected instrumental response into the PROFFIT forward model and evaluating its impact on spectral residuals, retrieval uncertainties, and retrieved total columns. This retrieval analysis is included directly in the manuscript and provides concrete examples linking ILS characterization to retrieval-level effects. Accordingly, the relevant updates have been implemented across the Abstract, Introduction, Methodology, Results and Data Analysis, Discussion, and Conclusion, ensuring that the connection between ILS diagnostics and atmospheric retrieval accuracy is clearly established in the current version.

**Response to Referee #2**

In the previous review round, under R.C. 2.7, the referee commented on the need to explicitly demonstrate the impact of non-ideal instrumental effects on atmospheric retrievals by comparing results obtained using nominal and modified instrument configurations. The comment was as follows:

The paper repeatedly emphasizes that accurate modeling of the non-ideal characteristics of an FTS will improve atmospheric measurements. Therefore, it would be appropriate to include a comparison of atmospheric measurements obtained from the instrument using the nominal and modified configurations. There is no need to demonstrate that the results are better, only that they are different and to show how these differences compare to the associated uncertainties. However, the referee noted some hesitation in insisting on the inclusion of the atmospheric results, as this addition may further increase the length of an already extensive paper. The referee suggested that this could be balanced by reducing some of the less essential discussion in the introduction.

**A.C.**:

We sincerely thank the referee for this thoughtful recommendation. In the original version of the manuscript, our intention was to focus on detailed instrument performance characterization and diagnostics, and to maintain a clearly instrument-oriented scope. For this reason, the atmospheric retrieval analysis was not included explicitly in the manuscript, and the discussion was limited to the diagnosis and correction of non-ideal instrumental effects. At that stage, we indicated that a comprehensive atmospheric retrieval analysis would be treated separately in order to preserve the focus and length of the instrument-characterization study.

Following the Editor's guidance and in line with the referee's recommendation, the manuscript has been substantially revised to include an explicit atmospheric retrieval application within the present study. In the revised version, the impact of nominal and modified instrument configurations on atmospheric retrievals is demonstrated through a targeted ethane ($C_2H_6$) retrieval. This retrieval illustrates how empirically characterized non-ideal instrumental effects propagate into spectral residuals, retrieval uncertainties, and retrieved total columns.

Accordingly, all text in the previous version referring to a separate or companion atmospheric retrieval study has been removed or updated. The retrieval analysis is now fully integrated into the manuscript, and the relevant updates have been implemented consistently across the Abstract, Introduction, Methodology, Results and Data Analysis, Discussion, and Conclusion. This revision directly addresses the referee's request by demonstrating differences between nominal and modified configurations in the context of atmospheric retrievals, while maintaining a focused and coherent presentation.

---

## Author Response (AR3)

**Response to the Editor**

**Dear Editor,**

Thank you very much for your careful evaluation of our manuscript and for your constructive comment regarding the interpretation of modulation efficiency behavior in Figure 3. We fully agree that, under realistic operating conditions, instrumental misalignment typically leads to a decrease in modulation efficiency rather than a physical increase.

In the original submission, Figure 3 was intended as a schematic illustration to qualitatively distinguish between idealized and non-ideal interferogram behavior. However, we recognize that the apparent increase in modulation amplitude shown for the real-instrument case could be misinterpreted as a genuine enhancement of modulation efficiency. To address this concern, we have revised the caption of Figure 3 to explicitly clarify that the illustrated increase in modulation amplitude arises from the ZPD-normalized representation and does not correspond to a physical increase in modulation efficiency. The revised caption now clearly states that, in real instruments, misalignment typically results in a reduction of modulation efficiency with increasing optical path difference.

In addition to addressing this specific point, we carefully revised the manuscript to improve sentence consistency and reduce redundancy throughout the text. The reference list was also reviewed in detail. In earlier revised versions, some references did not include all co-authors; in the present revision, all reference entries have been corrected to list the complete set of authors. Duplicate references with identical titles but different citation keys were removed without affecting any in-text citations. Furthermore, DOI information was added for all references where available, and bibliographic metadata were updated accordingly. One reference in Section 3.4 ("True gas cell column amount") was also replaced to better support the corresponding statement and to improve bibliographic accuracy. All of these changes are editorial in nature and do not affect the scientific content, results, or conclusions of the manuscript.

We are grateful for your guidance, which has helped improve the clarity, accuracy, and technical precision of the paper. We hope that the revised version satisfactorily addresses your comment and meets the journal's expectations.

Sincerely,
Gezahegn Sufa Daba
(on behalf of the authors)